# FRAG: Filtering Noise Using Snippet-Level Query Relevance

## ABSTRACT

Retrieval-Augmented Generation (RAG) augments large language models (LLMs) with external retrievals. Typically, expanding the retrieval window can improve RAG performance by retrieving more relevant content. However, it risks increased noise, which distracts the model's attention and degrades accuracy. To mitigate this, we propose *Fine-Grained RAG (FRAG)*, which *identifies key snippets from query* and *extracts relevant information while filtering noise from retrievals using snippet-level query relevance*. Yet, a new challenge arises in addressing complex RAG queries, which require knowledge pieces with implicit multi-hop logical relationships. Failure to identify these relationships may lead to loss of inference-based knowledge during filtering, degrading performance. To address this, we propose *Self-Recognition*, which extracts inference-based knowledge by *leveraging historically extracted knowledge as contextual references*. While FRAG notably improves performance, it incurs additional latency. To alleviate this, we present FRAG-ip, a fine-tuned framework which markedly accelerates FRAG by approximately $10\times$. Extensive experiments show that FRAG significantly boosts RAG, yielding average accuracy gains of 4.94%/13.44% on simple/complex tasks.

## 1 INTRODUCTION

Although the performance of LLM has seen breakthrough improvements, the issues of hallucination and outdated knowledge remain persistent challenges for LLM (Xu et al., 2024b). The RAG method (Lewis et al., 2020; Guu et al., 2020), by retrieving query-relevant knowledge from external knowledge bases as the generation context, effectively addresses these issues and significantly enhances LLM performance in knowledge-intensive tasks (Ram et al., 2023).

RAG typically employs a retriever to retrieve $N$ passages ($N$ denotes the retrieval window size) and combines them into a retrieval document as the generation context (Gao et al., 2023). Expanding the retrieval window is a common approach to enhance Retrieval-Augmented Language Model (RALM) performance by including more relevant knowledge (Lewis et al., 2020). However, this also introduces noise due to the text or vector similarity methods that retrievers rely on (Gao et al., 2023), which may degrade RALM accuracy (Yu et al., 2024b;c; Zhu et al., 2024), as *excessive noise distracts RALM's attention and ultimately leads to incorrect responses* (Theorem 2.2).

Existing works aim to reduce noise. SelfRAG mitigates it via multiple generations in shorter contexts (Asai et al., 2024), but neglects logical relationships across passages, limiting performance on complex RAG tasks (Zhang et al., 2024). RECOMP uses a compressor to compress the retrievals and filter noise (Xu et al., 2024a), yet fails to retain sufficient relevant information, resulting in suboptimal performance. RankRAG reranks the retrievals to select the top-$k$ relevant ones (Yu et al., 2024c). Yet, it faces a trade-off: a small $k$ may lose information while a large $k$ retains noise.

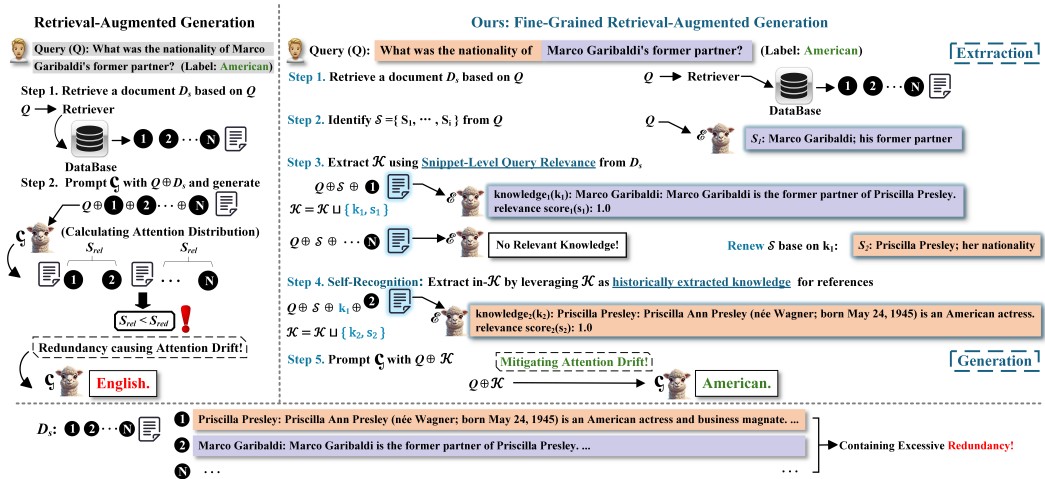

Figure 1: **Overview of FRAG.** $\mathcal{K}$ represents the extracted relevant knowledge. $\mathcal{S}$ denotes key sequence snippets of the query. $\mathcal{E}$ and $\mathcal{G}$ are the extractor and generator models, respectively. $S_{\text{rel}}$ and $S_{\text{red}}$ represent the total attention weights distributed to relevant information and noise in the generation context. Sentences highlighted with background color and color correspond to distinct snippets in $\mathcal{S}$ or associated relevant information.

To mitigate the negative impact of noise, we propose *Fine-Grained RAG (FRAG)*, which uses an extractor model to construct the generation context by performing fine-grained extraction of relevant knowledge and filtering noise from initial retrievals. *Firstly*, we observe that certain sequence snippets in the query can serve as key indicators for extracting relevant knowledge, which we refer to as "*key sequence snippets*". To this end, we identify key sequence snippets from the query, incorporate them into the extraction process, and filter noise using snippet-level query relevance (Theorem 3.2). To alleviate the attention distraction caused by noise, we decompose the extraction into steps, each targeting a single, shorter-context passage (Appendix B.1).

Nevertheless, new challenges arise. Complex RAG tasks, especially multi-hop reasoning, require multiple pieces of knowledge with implicit multi-hop relationships (Mavi et al., 2022). Some of the relevant knowledge, e.g., *inference-based knowledge*, can only be extracted if corresponding prior knowledge is available (Li & Peng, 2023). However, decomposing extraction into multiple steps may omit prerequisite knowledge and corresponding relationships, hindering inference-based knowledge extraction. To address this, *secondly*, we propose *Self-Recognition* (Section 3.2), which leverages historically extracted knowledge as reference in extraction to restore missing logical relationships.

While FRAG markedly enhances RALM performance, it incurs high computational cost due to frequent LLM calls for extraction. To address this, we propose FRAG-ip, a wrapper framework that employs dual-stage fine-tuning and markedly accelerates FRAG by approximately $10\times$ (Section 3.3).

Extensive experiments validate FRAG effectiveness in extracting relevant knowledge and filtering noise from initial retrievals. On the test dataset, FRAG retains over *90%* of the golden passages and filters over *80%* noise (Table 3). Compared to concurrent baselines, FRAG achieves *state-of-the-art* performance, yielding average accuracy gains of *4.94%/13.44%* on simple/complex tasks over naive RAG for the best-performing test model (Table 2). Our contributions are as follows:

- We demonstrate the negative impact of noise on RAG performance (Theorem 2.2), motivating the promising direction of extracting relevant information and filtering noise from the initial retrieved documents to enhance RAG accuracy.
- We propose FRAG, which introduces snippet-level query relevance to effectively filter noise (Theorem 3.2) and incorporates Self-Recognition to enhance inference-based knowledge extraction in complex tasks. To mitigate the computational cost, we present a wrapper-based fine-tuning framework that improves efficiency by approximately $10\times$.
- Extensive experiments show that FRAG markedly boosts RALM performance, demonstrating state-of-the-art performance. For the best-performing model, FRAG yields average accuracy gains of 4.94%/13.44% on simple/complex tasks compared to naive RAG (Table 2).

## 2 FOUNDATIONS: HOW NOISE PRODUCES A NEGATIVE IMPACT

**Preliminary.** Given the embedding matrix of an input sequence for RAG generation: $X = X_Q \oplus X_d$, the sequence consists of a query $X_Q$ and a retrieval document $X_d$. Here, $X_d = X_{\text{rel}} \oplus X_{\text{red}}$, and: $X \in \mathbb{R}^{n \times d}$, $X_Q \in \mathbb{R}^{n_q \times d}$, $X_{\text{rel}} \in \mathbb{R}^{n_{\text{rel}} \times d}$, and $X_{\text{red}} \in \mathbb{R}^{n_{\text{red}} \times d}$, where $X_{\text{rel}}$ represents the portion of the retrieved document $D_s$ that is relevant to $Q$, while $X_{\text{red}}$ stands for the irrelevant portion (i.e., noise). $n, n_q, n_{\text{rel}}$ and $n_{\text{red}}$ denote the token counts of $X, X_Q, X_{\text{rel}}$ and $X_{\text{red}}$, respectively. $d$ signifies the dimension of the embedding vectors. "$\oplus$" indicates the concatenation of different vectors along the vertical (row-wise) dimension in natural language order. $\mathfrak{T}$ refers to the first output token.

For simplicity, we consider: the RALM lacks parameterized knowledge to answer $Q$, relying solely on $X_{\text{rel}}$ for the required information; the RALM follows the correlation paradigm where the average cosine similarity satisfies $\tilde{S}_{(X_Q, X_{\text{rel}})} > \tilde{S}_{(X_Q, X_{\text{red}})}$; the RALM's attention focuses on $X_Q$ and its correlation with other input sequence parts. Additionally:

**Assumption 2.1.** The Word Mover's Distance (WMD) between $X_{\text{rel}}$ and $X_{\text{red}}$ satisfies the inequality (Kusner et al., 2015): $WMD_{(X_{\text{rel}}, X_{\text{red}})} \gg \tau, \tau \in \mathbb{R}^+$.

This indicates that the semantic distinction between $X_{\text{rel}}$ and $X_{\text{red}}$ is sufficiently pronounced, allowing the computation of RALM's attention distribution across different segments of $X_d$ (i.e., generating based on $X_{\text{rel}}$ or $X_{\text{red}}$) to be approximated by the attention matrix $\mathbf{A}_{\mathfrak{T}}$ computation for $\mathfrak{T}$.

**Theorem 2.2.** *In the input sequence of the retrieved document $X_d = X_{rel} \oplus X_{red}$, as the token count of irrelevant content $X_{red}$, i.e., $n_{red}$, increases, there exists $\tilde{n} \in \mathbb{N}^+$ such that when $n_{red} > \tilde{n}$, we have: $\mathfrak{A}_{rel} < \mathfrak{A}_{red}$, where $\mathfrak{A}_{rel}, \mathfrak{A}_{red}$ denote the total attention distribution weights allocated to $X_{rel}, X_{red}$ based on $X_Q$ (the proof is provided in Appendix C.1).*

Namely, *noise negatively impacts RALM generation by distracting the model's attention, causing it to shift toward irrelevant content and fail to generate correct answers based on relevant knowledge.* This theorem aligns with Zhu et al. (2024), who showed from the information bottleneck perspective that noise may degrade RALM accuracy by interfering with generation.

*Remark* 2.3. *The negative impact motivates us to extract relevant knowledge from the initial retrieval document, thereby filtering out noise.* Through reducing attention drift during RAG generation, FRAG mitigates noise-induced adverse effects and improves RALM accuracy.

## 3 FRAG: FILTERING NOISE USING SNIPPET-LEVEL QUERY RELEVANCE

FRAG finely extracts a set of relevant sequences ($\mathcal{R} = [x_1, x_2, \ldots, x_m]$) from the initial retrieved document ($D_s = [d_1, d_2, \ldots, d_N]$) based on their relevance to the query $Q$, using an extractor model $\mathcal{E}$. To enhance the extraction and ensure minimum information loss while filtering noise, FRAG identifies several key sequence snippets $\mathcal{S}$ from $Q$, and extracts the relevant information using *snippet-level query relevance*. To further filter noise in $\mathcal{R}$, each sequence in $\mathcal{R}$ is then evaluated for its relevance score to the query, and those sequences with scores above 0 are selected as the basic knowledge $\mathcal{K}$ that is relevant to $Q$, and those above a given relevance threshold $\mathcal{T}_G$ are selected as the basic knowledge for generation $\mathcal{K}'$. Furthermore, to reduce the negative effects of noise, FRAG conducts extraction within several steps, each focusing on *extracting information from a single, shorter-context retrieval passage*, which is proved to enhance extractor model's attention focusing on the relevant knowledge (see Appendix C.2 for proof). Additionally, to extract inference-based knowledge based on the corresponding prior information, we introduce *Self-Recognition*, a method for logical recognition. We provide an overview of FRAG extraction framework (Section 3.3), and furthermore, introduce FRAG-ip for better computational efficiency (Section 3.3).

### 3.1 HOW SNIPPET-LEVEL QUERY RELEVANCE BENEFITS THE NOISE FILTERING

**Definition.** Real-world user queries often include *lengthy background descriptions*. We define "*key sequence snippets*": certain sequence snippets of $Q$ that serve as key indicators for extracting relevant knowledge; "*non-key sequence snippets*": the remaining irrelevant snippets. The input sequence is denoted as $X_Q = X_{Q+} \oplus X_{Q-}$, where $X_{Q+}$ and $X_{Q-}$ are the embedding matrices of key and non-key sequence snippets. $X'_{Q+}$ denotes the embedding matrix of the key sequence snippets decomposed from $Q$. $X_{\text{rel}, \subset}$ signifies the portion of extracted passage relevant to $X_{Q+}$, while $X_{\text{red}, \subset}$ the irrelevant.

**Lemma 3.1.** *The presence of non-key sequence snippets $X_{Q^-}$ can diminish the LLM's attention on relevant knowledge during extraction. Namely, we have (proof is provided in Appendix C.3):*

$$\frac{\mathfrak{A}_{rel}^+}{\mathfrak{A}_{red}^+} \geq \frac{\mathfrak{A}_{rel}^+ + \mathfrak{A}_{rel}^-}{\mathfrak{A}_{red}^+ + \mathfrak{A}_{red}^-}. \tag{1}$$

Here, $\mathfrak{A}_{rel}^+, \mathfrak{A}_{rel}^-$ and $\mathfrak{A}_{red}^+, \mathfrak{A}_{red}^-$ signify the total attention distribution weights allocated to $X_{rel,\subset}$ and $X_{red,\subset}$, respectively, corresponding to $X_{Q^+}, X_{Q^-}$. *Thus, we have:*

**Theorem 3.2.** *By identifying key sequence snippets from $Q$, and explicitly appending the snippets to $Q$ as query-augmented information during extraction, i.e., $X'_{Q^+}$—which indicates leveraging snippet-level query relevance—it mitigates the attention reduction caused by the non-key sequence snippets $X_{Q^-}$ (Lemma 3.1), enhancing the extraction of relevant information. Namely, we have:*

$$\frac{\mathfrak{A}_{rel}^+ + \mathfrak{A}_{rel}^- + \mathfrak{A}_{rel}'^+}{\mathfrak{A}_{red}^+ + \mathfrak{A}_{red}^- + \mathfrak{A}_{red}'^+} \geq \frac{\mathfrak{A}_{rel}^+ + \mathfrak{A}_{rel}^-}{\mathfrak{A}_{red}^+ + \mathfrak{A}_{red}^-}. \tag{2}$$

Here, $\mathfrak{A}_{rel}'^+$ and $\mathfrak{A}_{red}'^+$ denote the total attention distribution weights allocated to $X_{rel,\subset}$ and $X_{red,\subset}$, respectively, corresponding to $X'_{Q^+}$. Proof is provided in Appendix C.4.

*Remark* 3.3. As established in Theorem 3.2, FRAG identifies key sequence snippets from the query and explicitly appends them to the query as query-augmented information during extraction, thereby *enhancing the extraction of relevant information from the retrievals using snippet-level query relevance* and *filtering noise with minimum information loss*.

### 3.2 SELF-RECOGNITION: ENHANCING INFERENCE-BASED KNOWLEDGE EXTRACTION

While multi-step extraction increases the likelihood of retrieving relevant information, it poses a challenge for complex tasks like multi-hop reasoning, which require knowledge pieces with multi-hop relationships. In such cases, it is necessary for an inference-based sequence in one passage to be preceded by corresponding prior relevant sequences from other passages to correctly identify its relevance Li & Peng (2023). Yet, lacking access to these passages leads to the absence of multi-hop links and hinders correct identification of the inference-based sequences within the extracted passage.

To address this, we introduce *Self-Recognition*, which incorporates historically extracted knowledge into subsequent extraction as contextual references. This facilitates the extraction of inference-based knowledge exhibiting multi-hop contextual relationships within retrieved passages, formalized as:

$$\mathbb{1}_{k_{\text{In-B}}^i} = Q \wedge \mathbb{1}_{k_{\text{P-B}}^{(1,\dots l)}} \wedge d_i, \tag{3}$$

where $\mathbb{1}_k$ denotes the presence or generation of extracted knowledge; $k_{\text{In-B}}^i$ and $k_{\text{P-B}}^{(1,\dots l)}$ the inference-based knowledge within $d_i$ and its $l$ pieces of prior knowledge; "$\wedge$" means logical necessity.

### 3.3 OVERVIEW OF FRAG EXTRACTION FRAMEWORK

To extract the relevant sequences $\mathcal{R} = [x_1,\dots,x_m]$ from the retrieval document $D_s = [d_1,\dots,d_N]$, where $d_i$ represents the $i$-th retrieval passage, and to construct the basic knowledge $\mathcal{K} = [k_1,\dots,k_t]$, from which the most relevant sequences are selected to form the generation context $\mathcal{K}'$ based on a relevance threshold for generation ($\mathcal{T}_G$), FRAG employs *an open-sourced extractor model $\mathcal{E}$* and constructs *an LLM-based Extraction Framework*. The framework comprises six modules: *Extractor $\mathfrak{E}$, Validator $\mathfrak{V}$, Prefixer $\mathfrak{P}$, Deduplicator $\mathfrak{D}$, Filter $\mathfrak{F}$, and Assessor $\mathfrak{S}$*. Figure 2 illustrates the workflow of the FRAG extraction framework, while

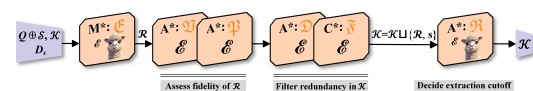

Figure 2: **Workflow of FRAG extraction framework.** $\mathfrak{E}, \mathfrak{V}, \mathfrak{P}, \mathfrak{D}, \mathfrak{F}, \mathfrak{R}$ represent *Extractor, Validator, Prefixer, Deduplicator, Filter, and Verifier*, respectively. $M^*, C^*, A^*$ stand for *main module, core module* and *auxiliary module*, respectively. $\mathcal{E}$ signifies the extractor model. $\mathcal{R}, s$ denote the extracted relevant sequences and their corresponding relevance scores, respectively.

Table 1 outlines the function of each module. FRAG extraction framework leverages *Chain-of-Thought (CoT)* reasoning Wei et al. (2022) and *few-shot prompt engineering* Brown et al. (2020) to guide the LLM in reasoning for better extraction. Algorithm 1 illustrates the FRAG extraction.

**Construction of FRAG extraction framework.** The six modules are driven by $\mathcal{E}$, and operate sequentially to extract the relevant knowledge $\mathcal{K}$ from $D_s$ and filter noise. $\mathfrak{E}$, main module of FRAG, identifies key sequence snippets $\mathcal{S}$ from $Q$, and extracts relevant sequences $\mathcal{R}$ from $D_s$ using snippet-level query relevance:

$$x_i = (Q \oplus S) \otimes d_i = \begin{cases} d_i^{\text{rel}}, & \text{if } d_i^{\text{rel}} \text{ in } d_i, \\ \varnothing, & \text{otherwise;} \end{cases} \quad (4)$$

where "$\otimes$" signifies the semantic relevance matching operation and $d_i^{\text{rel}}$ denotes the portion (if present) of $di$ relevant to $Q$. It incorporates Self-Recognition, which integrates the historically extracted knowledge as references in subsequent extraction. $\mathfrak{V}, \mathfrak{D}$ verify the consistency of $\mathcal{R}$ with $D_s$ and $\mathcal{K}$, respectively: the former validates correctness, while the latter detects noise. Any sequence that is either incorrect or noisy will be discarded. $\mathfrak{F}$ assesses the relevance score $s$ between $\mathcal{R}$ and $Q$, filtering noise in $\mathcal{K}$:

$$s_i = (Q \oplus S) \odot x_i = Q \odot x_i + S \odot x_i, \quad (5)$$

Table 1: **FRAG work modules. con**, **dup**, **ans** and **stop** represent *consistent, duplicate, QueryAnswered* and *stop* flag, respectively. $\mathcal{K}_{\text{prior}}$ denotes the basic knowledge extracted in the last extraction round. $(\mathcal{K})$ means $\mathcal{K}$ is an optional input.

| Module | Input→Output | Function |
|---|---|---|
| $\mathfrak{E}$ | $Q \to \mathcal{S}$ $Q \oplus \mathcal{S}, (\mathcal{K}), d_i \to x_i$ | Identify $\mathcal{S}$ from $Q$. Extract $x_i$ from $d_i$. |
| $\mathfrak{V}$ | $x_i, d_i \to$ **con** | Discard $x_i$ if $x_i$ is inconsistent with $d_i$ (**con** is *false*). |
| $\mathfrak{P}$ | $x_i, d_i \to x_i$ | Prepend the topics or subjects of $d_i$ to $x_i$. |
| $\mathfrak{D}$ | $x_i, \mathcal{K} \to$ **dup** | Discard $x_i$ when $x_i$ duplicates knowledge in $\mathcal{K}$ (**dup** is *true*). |
| $\mathfrak{F}$ | $x_i, Q, (\mathcal{K}) \to s_i$ | Assess relevance $s_i$ of $\{x_i, Q\}$. |
| $\mathfrak{S}$ | $Q, \mathcal{K} \to$ **ans** $\mathcal{K}, \mathcal{K}_{\text{prior}} \to$ **stop** | Stop extraction if $Q$ is answerable or no more $x_i$ is extracted (**ans**/**stop** is *true*). |

where "$\odot$" denotes the semantic relevance computation between sequences. $\mathcal{K}$ also acts as optional references, enabling accurate scoring of inference-based knowledge. $s_i$ is evaluated on a scale from 0 to 1, and extracted sequences with $s_i > 0$ are included in basic knowledge $\mathcal{K}$:

$$\mathcal{K} = \mathcal{K} \cup \begin{cases} \{(x_i, s_i)\}, & \text{if } s_i > 0, \\ \varnothing, & \text{otherwise;} \end{cases} \quad (6)$$

and the extracted sequences are sorted by their relevance in positive order:

$$\mathcal{K} = [(x_u, s_u), \ldots, (x_v, s_v)], \quad s_u \geq s_{u+1} \geq \cdots \geq s_v > 0. \quad (7)$$

$\mathfrak{S}$ decides when the extraction process should be terminated, thereby minimizing the extraction time. An extraction round is defined as a process in which all passages from the initial retrieval document are processed once by FRAG extraction framework to extract relevant knowledge. The workflow iterates until no further relevant knowledge can be extracted or the maximum extraction rounds, $L_{\text{Max-Rounds}}$, is reached. More details of FRAG implementation are provided in Appendix B.

---

**Algorithm 1** FRAG Extraction

---

**Require:** Query $Q$; $Q$'s key sequence snippets $\mathcal{S}$; Retrieved documents $D_s = [d_1, \ldots, d_N]$; FRAG extraction framework $\mathcal{F}$; Basic knowledge $\mathcal{K}$; Extracted relevant sequence $x_i$; Relevance score $s_i$ of $\{Q, x_i\}$; Max extraction rounds $L_{\text{Max-Rounds}}$; Check $\mathcal{K} \leq \mathcal{K}_{\text{prior}}$: verify if $\mathcal{K}$ does not exceed $\mathcal{K}_{\text{prior}}$.

   **Initialize ans** $= false$; **stop** $= false$; $r = 1$;

   $\mathcal{S} \xleftarrow{\mathcal{F}} Q$       ▷ Decompose $\mathcal{S}$ from $Q$ (Section 3.1)

   **while** {**ans** is $false$ **and stop** is $false$ **and** $r \leq L_{\text{Max-Rounds}}$} **do**

      $\mathcal{K}_{\text{prior}} \leftarrow \mathcal{K}$

      **for** $i = 1, \ldots, n$ **do**

         **if** $\mathcal{K}$ is $\varnothing$ **then**

           $(x_i, s_i) \xleftarrow{\mathcal{F}} \{Q \oplus \mathcal{S}, d_i\}$    ▷ Extract $x_i$ given $Q \oplus \mathcal{S}$ & Assess $s_i$ (Section 3.1)

         **else**

           $(x_i, s_i) \xleftarrow{\mathcal{F}} \{Q \oplus \mathcal{S}, d_i, \mathcal{K}\}$     ▷ Self-Recognition (Section 3.2)

         $\mathcal{K} \leftarrow \mathcal{K} \cup \{(x_i, s_i)\}$, **with** $s_i > 0$

      $\mathcal{S} \xleftarrow{\mathcal{F}} Q \oplus \mathcal{K}$; $r \leftarrow r + 1$           ▷ Reidentify $\mathcal{S}$ if necessary

      **Assess** {**ans**: $true$ **or** $false$} $\xleftarrow{\mathcal{F}} \{Q, \mathcal{K}\}$    ▷ Assess if $Q$ is answerable based on $\mathcal{K}$

      **if** $\mathcal{K} \leq \mathcal{K}_{\text{prior}}$ **then**

         **stop** $\leftarrow true$            ▷ No more basic knowledge extracted

   Return $\mathcal{K}$

---

**FRAG limitation in computational efficiency.** Although FRAG significantly improves RALM performance by effectively extracting relevant information and filtering noise from the initial documents, it requires a large number of LLM calls for extraction, leading to high computational cost. The extraction process can take $15.07s$ per query on an A100 GPU on HotPotQA (Table A.6).

To reduce computational cost, we propose FRAG-ip, a wrapper framework that performs a *dual-stage fine-tuning process* on the Qwen2-7B-Instruct model to accelerate the extraction. In the first stage, the extractor model $\mathcal{E}$ is fine-tuned to (1) *identify key sequence snippets $\mathcal{S}$ from the query $Q$*, and (2) *extract relevant sequences $\mathcal{R}$ and filter noise from $t$ retrieved passages in a fine-grained manner using snippet-level query relevance*, while maintaining consistency with the source passages (to support $\mathfrak{E}$). The post-training of the extraction module further incorporates the Self-Recognition method, leveraging historically extracted knowledge as contextual references to enhance the extraction of inference-based knowledge. In the second stage, $\mathcal{E}$ is fine-tuned to (1) *assess the relevance score of a given sequence with respect to $\mathcal{S}$ and $Q$* (to support $\mathfrak{F}$), and (2) *determine whether a query is answerable given a set of relevant or irrelevant knowledge* (to support $\mathfrak{A}$).

Furthermore, with improved extraction performance and more relevant, consistent extracted sequences, $\mathfrak{V}, \mathfrak{D}$ are transformed to be purely driven by an program executor $\mathcal{P}$. *These optimizations significantly reduce the extraction runtime by approximately by $10\times$ (Table A.6).* More details, including the fine-tuning process, results of the experiments utilizing FRAG-ip and a comparison of computational cost between FRAG-ip and FRAG are provided in Appendix B.4.3.

## 4 EXPERIMENTS

### 4.1 DATASETS AND EVALUATION METRICS

We evaluate FRAG on three simple single-hop datasets — PopQA (Mallen et al., 2023), PubHealth (Akhtar et al., 2022), and ARC-Challenge (Clark et al., 2018) — and on four complex datasets, including three multi-hop datasets (HotpotQA (Yang et al., 2018), 2WikiMultihopQA (Ho et al., 2020), and MuSiQue (Trivedi et al., 2022)) and one long-context task (LongBench-v2 (Bai et al., 2024)). For single-hop and long-context datasets, we use accuracy (*acc*) as the evaluation metric; For multi-hop datasets, we evaluate using Exact Match (*em*) (Rajpurkar et al., 2016). However, since *em* may yield inaccurate assessments by simply matching answer text with the ground truth (Wang et al., 2023a), we use *LLM-em* (Yu et al., 2024a), which leverages LLMs to assess the correctness of RALM-generated answers based on ground-truth, thereby providing more reliable evaluation. Experimental details, including the conduction of LLM-em evaluation, are present in Appendix D.

### 4.2 BASELINES

**Baselines without retrievals.** We evaluate the most advanced models, DeepSeek-v3 (Liu et al., 2024a) and GPT-4o (Achiam et al., 2023), as well as advanced open-source models, including Llama3-8B-Instruct (Meta AI, 2024), Qwen2-7B-Instruct (Yang et al., 2024), and the advanced fine-tuned question answering models for RAG, ChatQA-1.5-8B/70B (Liu et al., 2024c). We also compare FRAG to other advanced models simulated by GPT-4, Alpaca7B/13B (Dubois et al., 2023).

**Baselines with retrievals.** We test the baselines using the initial retrieval passages from our method for generation. We evaluate Llama3-8B-Instruct, Qwen2-7B-Instruct and ChatQA-1.5-8B/70B using *naive RAG* across all datasets. We also include the following representative RAG baselines: RankGPT (Sun et al., 2023), which relies on GPT-3.5 to rerank the retrievals and select the top-k ($k = 5$ for test) passages for generation; SelfRAG (Asai et al., 2024), which employs a self-reflection mechanism to critique RALM responses and select the best one. Results of Alpaca 7B/13B were reported in SelfRAG method; RECOMP (Xu et al., 2024a), which relies on a compressor to condense the retrieval passages into short summaries; Rewrite-Retrieve-Read (Ma et al., 2023), which inserts a query rewriting step to enhance retrieving; RQ-RAG (Chan et al., 2024), which fine-tunes a model to rewrite, decompose, and disambiguate queries; ActiveRAG (Xu et al., 2024c), which uses four agent-driven strategies to integrate external evidence with LLM memory; SAIL (Luo et al., 2023), a method that instruction-tuning a model using the *top* retrieved documents; RA-DIT (Lin et al., 2024), a method that fine-tunes both the retriever and the generator; REPLUG (Shi et al., 2024), which augments RALMs with a tunable retrieval model; Results of LLAMA-65B REPLUG were reported in the RA-DIT method; LongRAG (Zhao et al., 2024), which utilizes a hybrid retriever

and a long-context retrieval chunks refinement method; HippoRAG (Jimenez Gutierrez et al., 2024), which mimics hippocampal memory via an LLM-built knowledge graph and Personalized PageRank for efficient single-step multi-hop retrieval; Search-o1 (Li et al., 2025), which enhances RAG via a reasoning model and agentic search with document refinement for multi-step knowledge integration.

### 4.3 IMPLEMENTATION

**Extraction settings.** We separately employ the advanced instruction fine-tuned models, Llama3-8B-Instruct and Qwen2-7B-Instruct, as extractor models to extract relevant knowledge $\mathcal{K}$ from the initial retrieval documents. $L_{\text{Max-Rounds}}$ is set to a default of 1 for simple tasks and 3 for complex ones.

**Generation and evaluation settings.** We separately use Llama3-8B-Instruct and Qwen2-7B-Instruct as the generator models. Additionally, we utilize ChatQA-1.5-8B/70B as generators using the basic knowledge extracted by Qwen2-7B-Instruct to test the robustness of FRAG when transferring extracted knowledge from one extractor model to another generator. We test four set values as $\mathcal{T}_G$ during the experiments: 0.2/0.45/0.7/0.95 to analyze the impact of different $\mathcal{T}_G$ values on RALMs' performance. We use GPT-4o as the evaluator to assess LLM-em scores across the multi-hop datasets.

Table 2: **Overall experiment results. Bold** numbers indicate the best result across all models, underlined numbers denote the second-best, and **gray-colored bold** numbers signify the best result among retrieval-based models. "-" indicates values not reported in the original papers or inapplicable. "*" signifies that a detailed analysis is provided in Appendix E.1 for the results. "†" suggests potential data leakage to achieve such performance without retrieving any documents (Zhou et al., 2023).

| | Single-hop | | | Multi-hop | | | | | | Long-Context |
|---|---|---|---|---|---|---|---|---|---|---|
| | **Pop** | **Pub** | **ARC** | **HotpotQA** | | **2Wiki** | | **MuSiQue** | | **LongBench-v2** |
| **LMs** | (acc) | (acc) | (acc) | (llm-em) | (em) | (llm-em) | (em) | (llm-em) | (em) | (acc) |
| | | | | *Baselines without retrieval* | | | | | | |
| DeepSeek-v3 | 31.17 | 71.02 | **95.74**† | - | 31.9 | - | 43.9 | - | 9.2 | **33.6** |
| GPT-4o | 43.6 | 56.43 | **95.74**† | - | 36.7 | - | 47.4 | - | 14.5 | 32.41 |
| Alpaca-7B | 23.6 | 49.8 | 45 | - | - | - | - | - | - | - |
| Alpaca-13B | 24.4 | 55.5 | 54.9 | - | - | - | - | - | - | - |
| ChatQA-1.5-8B | 27.66 | 62.21 | 58.52 | - | 15.2 | - | 36 | - | 0.7 | 24.06 |
| Llama3-8B-Instruct | 26.16 | 67.88 | 74.11 | - | 0.8 | - | 0 | - | 0 | 1.79 |
| Qwen2-7B-Instruct | 27.38 | 62.92 | 78.36 | - | 0 | - | 0 | - | 0 | 20.87 |
| ChatQA-1.5-70B | 33.17 | 66.97 | 86.46† | - | 19.5 | - | 35.3 | - | 2.3 | 32.41 |
| | | | | *Baselines with retrieval* | | | | | | |
| Alpaca-7B | 46.7 | 40.2 | 48 | - | - | - | - | - | - | - |
| Alpaca-13B | 46.1 | 51.1 | 57.6 | - | - | - | - | - | - | - |
| SAIL-7B | - | 69.2 | 48.4 | - | - | - | - | - | - | - |
| SelfRAG-7B | 54.9 | 72.4 | 67.3 | - | 12.9 | - | 16.8 | - | 1.2 | 22.86 |
| SelfRAG-13B | 55.8 | 74.5 | 73.1 | - | 13.2 | - | 6.2 | - | 1.5 | 2.58 |
| RECOMP-20B | - | - | - | - | 30.4 | - | - | - | - | - |
| RA-DIT-65B | - | - | 60.5 | - | 40.7 | - | - | - | - | - |
| REPLUG-LLAMA-65B | - | - | - | - | 41.1 | - | - | - | - | - |
| RankGPT-7B | 55.68 | 78.52 | 82.28 | - | 42.9 | - | 35.6 | - | 15.9 | 24.45 |
| Rewrite-Retrieve-Read | - | - | - | - | 34.38 | - | - | - | - | - |
| RQ-RAG-7B | 57.1 | - | 68.3 | - | 0* | - | 0* | - | 0* | 27.04 |
| LongRAG-6B | - | - | - | - | 40.5 | - | 37.5 | - | 17.5 | - |
| ActiveRAG-8B | 46.46 | 32.22 | 46.34 | - | 23.6 | - | 29 | - | 7.5 | 17.89 |
| HippoRAG | - | - | - | - | 41.8 | - | 46.6 | - | 19.2 | - |
| IRCoT+HippoRAG | - | - | - | - | 45.7 | - | 47.7 | - | 21.9 | - |
| Search-o1-32B | - | - | - | - | 45.2 | - | **58.0** | - | 16.6 | - |
| ChatQA-1.5-8B | 53.75 | 67.17 | 37.99 | 54.6 | 9.9 | 43.7 | 33.8 | 25.7 | 1.3 | 13.12 |
| Llama3-8B-Instruct | 59.39 | 72.14 | 75.64 | 74.9 | 38.8 | 54.4 | 41.8 | 28.4 | 14.9 | 22.27 |
| Qwen2-7B-Instruct | 51.82 | 75.08 | 76.83 | 57.4 | 34.9 | 50.2 | 42.2 | 23.5 | 9.2 | 22.66 |
| ChatQA-1.5-70B | 58.97 | 72.14 | 74.11 | 72.3 | 26.8 | 54.2 | 21.9 | 38.7 | 8 | 24.06 |
| | | | | *ours* | | | | | | |
| FRAG-ChatQA-1.5-8B | 54.11 | 73.66 | 46.42 | 67.4 | 27.9 | 52.2 | 44.2 | 35.3 | 12.4 | 21.47 |
| FRAG-Llama3-8B-Instruct | **59.83** | 75.99 | 77.34 | 75.4 | 41.7 | 45.2* | 32.1* | 37.3 | 22.9 | 24.45 |
| FRAG-Qwen2-7B-Instruct | 57.97 | **79.84** | 82.03 | 72.7 | **47.7** | **60.9** | 51.5 | **44.1** | **25.6** | **29.82** |
| FRAG-ChatQA-1.5-70B | 59.69 | 76.49 | **82.79** | **77.8** | 38.4 | 58.5 | 28.4 | 42.4 | 19.3 | 29.03 |

### 4.4 MAIN RESULTS

**FRAG performance.** It can be demonstrated that the same model consistently performs better when using FRAG compared to naive RAG across most tested datasets (except for Llama3 on

2WikiMultihopQA, for which analysis is provided in Appendix E.1), as shown in Table 2 and Figure 3. In most cases, the naive RAG method generally outperforms non-retrieval-based method. This confirms Theorem 2.2 that *introducing excessive noise can indeed reduce the accuracy of RALM*. Moreover, *by effectively extracting relevant knowledge and filtering noise, the negative effects of noise can be mitigated, thereby improving the performance of RALM*.

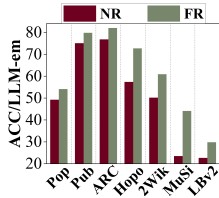 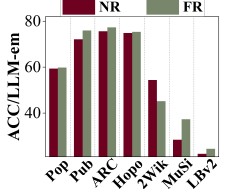 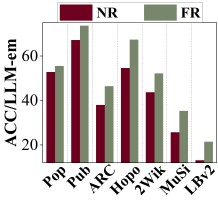 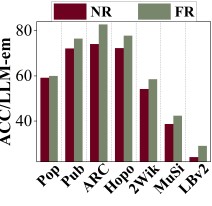

(a) Qwen2-7B-Instruct  (b) Llama3-8B-Instruct  (c) ChatQA-1.5-8B  (d) ChatQA-1.5-70B

Figure 3: Performance comparison of four tested models using FRAG (**FR**) versus naive RAG (**NR**).

Additionally, *FRAG achieves the best overall performance among retrieval-based baselines*, outperforming the reranking method RankGPT, query reformulation baselines such as RQ-RAG, and advanced methods including SelfRAG and ActiveRAG across all tasks. On most multi-hop datasets, FRAG also surpasses strong reasoning-model-based method, Search-o1, and advanced baselines such as HippoRAG, LongRAG, RA-DIT, REPLUG, and RECOMP. Furthermore, FRAG outperforms state-of-the-art fine-tuned QA models ChatQA-1.5-8B/70B when they are used with naive RAG. Notably, *FRAG-Qwen2-*

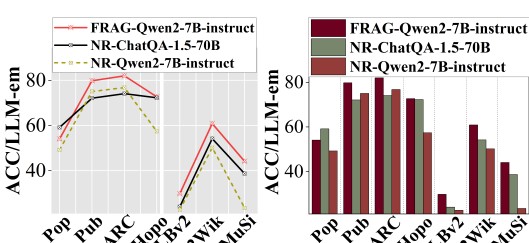

Figure 4: Performance comparison of FRAG-Qwen2-7B-Instruct versus NR-ChatQA-1.5-70B.

*7B-Instruct achieves significant improvements across all datasets,* with an average accuracy gain of *4.94%* on single-hop datasets and *13.44%* on multi-hop and long-context datasets, compared to using naive RAG. These results demonstrate the advantages of FRAG. Moreover, *FRAG-Qwen2-7B-Instruct achieves performance that nearly surpasses ChatQA-1.5-70B using naive RAG,* despite a much smaller model size, as shown in Figure 4. This suggests that small-sized models can still demonstrate excellent RAG performance when generating with contexts that have been filtered to remove excessive noise, without the need to simply increase the model size to improve performance. The fact that *ChatQA-1.5-8/70B, when augmented with retrievals extracted by Qwen2, consistently perform better than with naive RAG* further demonstrates FRAG effectiveness and robustness.

**FRAG efficiency in extracting relevant knowledge and filtering noise.** The HotpotQA dataset contains annotations for golden passages within the retrieved documents (Yang et al., 2018). In the experiments using the FRAG method to extract relevant knowledge, we evaluate the practical effectiveness of our method using the following four metrics: (a) *rete*: retention ratio of the golden passages after extraction, i.e., the number of golden passages extracted divided by the total number of golden passages (relevant knowledge extracted from the same golden passage is regarded as a single extracted golden passage). (b)

Table 3: FRAG efficiency in retaining relevant knowledge and filtering noise. "↑"/"↓" indicate an increase/decrease relative to the corresponding proportion in the initially retrieved documents.

| LMs | HotpotQA | | | |
|---|---|---|---|---|
| | ($rete$) | ($p_{ori}$) | ($p_{ext}$) | ($\eta$) |
| FRAG-Qwen2-7B -Instruct | 90.15 | 16.10 | **42.21** (26.11↑) | **82.87**↓ |
| FRAG-Llama3-8B -Instruct | 87.65 | 16.10 | **39.13** (23.03↑) | **83.95**↓ |

$p_{ori}$: proportion of tokens from golden passages within the initial retrieved documents. (c) $p_{ext}$: proportion of tokens from golden passages in the extracted documents. (d) $\eta$: filtering rate of tokens from noise in the extracted documents compared to the initial retrieved documents.

Table 3 demonstrates that FRAG effectively extracts and retains relevant information from initial retrieval documents, achieving a retention ratio of 90.15% with Qwen2 and 87.65% with Llama3, while significantly filtering noise by 82.87% and 83.95%, respectively. This notably increases the

proportion of relevant information in the generation context by at least 26.11% on Qwen2 and 23.03% on Llama3 (notably, as noise exists even within golden passages, the actual enhancements surpass these values). The above analysis indicates that FRAG significantly enhances RALM performance due to two key factors: first, *it effectively identifies and extracts relevant information from the initial retrieval documents*; second, *it efficiently filters noise*, thus *substantially increasing the proportion of relevant information in the generation context*. This process effectively *mitigates attention drift caused by noise during RAG generation*, thereby enhancing RALMs' capability to generate accurate answers based on the relevant knowledge within the generation context.

**FRAG advantages in improving spatial efficiency for RAG generation.** FRAG achieves an average reduction in input tokens of up to 73.23% for Qwen2 across all datasets during the RAG generation phase compared to naive RAG, and 78.54% for Llama3. Under optimal conditions, this reduction reaches 78.33% for Qwen2 and 85% for Llama3, as illustrated in Figure 5. The reduction in input tokens within the generation context across all datasets under varying $\mathcal{T}_G$ values highlights FRAG's significant enhancement of the spatial efficiency in the RAG approach. Furthermore, it demonstrates that *RALM performance can be significantly improved by effectively filtering noise from the generation context*, as supported by the combined results from Table 2, Table 3, and Figure 5.

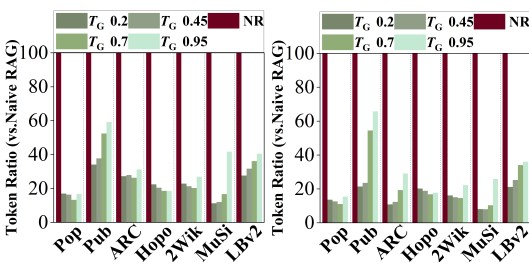

(a) Qwen2-7B-Instruct  (b) Llama3-8B-Instruct

Figure 5: Comparison of input token counts between FRAG and naive RAG (NR).

## 4.5 ABLATION STUDIES

**Contribution of different components.** To verify whether conducting extraction using snippet-level query relevance increases the probability of extracting relevant knowledge from the initial retrieved documents, and assess the role of Self-Recognition in identifying logical relationships between different relevant sequences in the retrieved documents and recognizing inference-based knowledge based on historically extracted knowledge, we conduct ablation experiments using Qwen2-7B-Instruct as both the extractor and generator model on HotpotQA.

The results in Table 4 indicate that the extraction method using snippet-level query relevance indeed improves the retention rate of golden passages and enhances model performance. For

Table 4: Results of the ablation studies. **Bold** numbers indicate the best performance, and "↓" indicates a decrease compared to the best.

| LMs | HotpotQA | | | |
|---|---|---|---|---|
| | (llm-em) | (*rete*) | ($p_{\text{ext}}$) | ($\eta$) |
| FRAG-Qwen2-7B -Instruct | **72.7** | **90.15** | 42.21 | 82.87 |
| **w/o** adding $\mathcal{S}$ | 66.5 (6.2↓) | 75.10 | **46.69** | 92.01 |
| **w/o** Self-Recognition | 60.0 (12.7↓) | 78.10 | 38.73 | 88.81 |
| **w/o** adding $\mathcal{S}$ and Self-Recognition | 58.6 (14.1↓) | 62.15 | 46.03 | **95.18** |

multi-hop datasets, the introduction of Self-Recognition significantly improves both the retention rate of golden passages and proportion of relevant information in the generation context, thereby yielding a greater impact on accuracy. These findings validate the correctness and effectiveness of FRAG.

## 5 CONCLUSION

This work presents FRAG, an innovative LLM-based extraction framework that filters noise in retrieved documents, increases the proportion of relevant information in the RAG generation context, and enhances RALM accuracy. FRAG utilizes six modules to extract relevant knowledge using snippet-level query relevance and incorporates Self-Recognition to leverage historically extracted knowledge as references and enhance inference-based knowledge extraction. Extensive experiments demonstrate that FRAG markedly boosts RAG performance. Additionally, we propose an improved wrapper framework FRAG-ip to reduce the computational cost.

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

CONTENTS

## A    RELATED WORK

**Retrieval-Augmented Generation.** Retrieval-Augmented Generation (RAG) enhances the generation process by retrieving relevant knowledge from external knowledge bases as context for answering queries (Lewis et al., 2020; Guu et al., 2020), demonstrating outstanding performance in knowledge-intensive tasks (Ram et al., 2023). With advancements in retrievers and large models used as generators, a diverse range of RAG techniques have emerged to improve generation quality and accuracy. SelfRAG fine-tunes the generator model to adaptively retrieve the necessary knowledge and leverages a self-reflection mechanism to generate and critique responses, ultimately yielding the best answers (Asai et al., 2024). ChatQA is a model series trained with high-quality question-answer pairs specifically for RAG and QA tasks, with its most advanced version outperforming GPT-4 on RAG and QA tasks (Liu et al., 2024c). Meanwhile, RAG methods continue to expand into more complex application scenarios. ToG proposes an integrating paradigm "LLM $\otimes$ KG", using LLMs over retrieved knowledge graphs to solve multi-hop problems (Sun et al., 2024). Graph RAG constructs global knowledge graph communities from unstructured text knowledge bases to address tasks like "query-focused summarization" (QFS) (Edge et al., 2024).

**The Negative Impact of Retrieved Noise on RAG Tasks.** A common belief is that increasing the retrieval window to introduce more relevant knowledge enhances RAG performance, as generative models effectively aggregate and synthesize evidence from multiple passages (Izacard & Grave, 2021). However, empirical engineering practices have shown that continually expanding the retrieval window does not guarantee sustained performance improvements for RAG; instead, it may degrade performance as the context length increases (Yu et al., 2024b;c). This issue arises because retrievers typically use text or vector similarity to retrieve contexts related to the query (Gao et al., 2023), and these retrieved contexts may not always be relevant to the query (Cuconasu et al., 2024), leading to noise. As the retrieval window expands, a substantial amount of noisy information is included, which in turn dilutes the model's focus on truly relevant information (Yu et al., 2024b;c). This results in an "attention drift" towards irrelevant content, thereby undermining the performance of RALM.

**Concurrent Research on Reducing the Negative Impact of Retrieved Noise.** Numerous research efforts have been implemented to mitigate the detrimental effects of noise. SelfRAG mitigates the negative impact of noise by generating responses from a single, shorter retrieved document, either once or multiple times, rather than relying on multiple retrieved passages simultaneously for a single generation (Asai et al., 2024). However, this approach severs logical connections between different retrieved passages, resulting in suboptimal performance in complex RAG tasks (Zhang et al., 2024). RECOMP reduces the negative effects of noise by incorporating additional compressors to summarize the retrieval documents (Xu et al., 2024a). However, this method lacks effective practical principles for identifying relevant information in the retrieved documents, leading to the loss of relevant knowledge during the compression process, which hinders it from achieving optimal performance (Xu et al., 2024a). RA-Dit employs a dual-stage fine-tuning approach involving both the retriever and the generator to acquire more relevant knowledge and improve the RALM's utilization of relevant information during generation, thereby achieving higher-quality outputs (Lin et al., 2024). Ori Yoran et al. fine-tuned models based on a mix of relevant and irrelevant contexts to enhance robustness against noisy content, thereby reducing the model's focus on irrelevant information during generation (Yoran et al., 2024). FILCO utilizes fine-tuned context filtering models to identify useful contexts based on lexical and information-theoretic approaches to effectively filter out noise (Wang et al., 2023b). RankRAG reduces the input of irrelevant content during generation by adding a small fraction of ranking data into the training blend, enabling the model to obtain the top-k most relevant passages (Yu et al., 2024c). However, methods relying on fine-tuning often have performance limitations that are dependent on the quality and comprehensiveness of the datasets used for fine-tuning or training, which can result in reduced generalization and versatility of the model.

Fine-tuning (Sun et al., 2019) is a widely adopted technique in adapting pre-trained models to downstream tasks, where model parameters are updated using task-specific labeled data. A well fine-tuned models often exhibit improved performance or computational efficiency on specific tasks (Naveed et al., 2023). Recent studies have proposed various fine-tuning strategies, including full-parameter tuning, adapter-based methods, and low-rank adaptation (e.g., LoRA (Hu et al., 2022)), to balance performance and efficiency.

# B  FRAG DETAILS

## B.1  EXTRACTION IN SHORTER CONTEXT MATTERS

The typical retrievers employed in RAG generally split the retrieval knowledge base into fixed-length chunks (usually 256/512 tokens) (Gao et al., 2023) prior to retrieval. Therefore, the retriever returns $N$ logically independent and semantically complete paragraphs, determined by the retrieval window size. Furthermore, existing retrievers typically rely on text similarity or vector similarity for knowledge retrieval (Gao et al., 2023), which results in the retrieved passages not always being relevant to the query. Specifically, relevant information is concentrated in $t$ paragraphs (where $t \leq N$). For these $t$ retrieved passages, derivations show that the average proportion of relevant information in a single retrieval passage is higher than in a long-context document composed of $N$ passages (Derivation is presented in Appendix C.2).

Established on Theorem 2.1, this suggests that *using multiple extraction steps, each extracting knowledge from a single, shorter-context passage, facilitates the extractor model focusing on relevant information*, thereby improving the extraction of such information. However, for other $N - t$ noisy passages that contain no relevant information, the extractor model might mistakenly extract irrelevant knowledge. To mitigate this, we implemented two measures: (i) Since noise has inherently low relevance to the query, we require the extractor model to first assess whether the passage to be extracted is relevant to the query before extraction, forming a complete reasoning chain to guide the model's correct extraction behavior; (ii) We utilize the module Filter $\mathfrak{F}$ (Section 3.3, overview of FRAG extraction framework) to evaluate the relevance score of $\mathcal{R}$ to the query, where we standardize the scoring process to exclude sequences with low or no relevance.

## B.2  DETAILS OF FRAG EXTRACTION FRAMEWORK

### B.2.1  MODULE: EXTRACTOR $\mathfrak{E}$

$\mathfrak{E}$ identifies and extracts key sequence snippets ($\mathcal{S}$) from the query Q, following the formulation:

$$\mathcal{S} = Q \cap A^{-1}, \tag{A.8}$$

where $A^{-1}$ denotes the essential information required to derive the query answer. Key sequence snippets $\mathcal{S}$ refer to *the certain sequence snippets of a query which serves as key indicators or fundamental information* for retrieving the query answer. In contrast, non-key snippets typically consist of lengthy, noisy background or instructional content that may distract the model during information extraction or answer generation (Section 3.1). For example, consider the following instance from LongBench-v2:

> **Question:** You are given a grammar book of Kalamang language, now translate the following Kalamang sentence into English: Faisal emun me mindi don bolonet me ma he kademor.

The essential snippet in this case is *"Faisal emun me mindi don bolonet me ma he kademor"*, as each word in this sentence needs to be interpreted to produce the correct English translation. FRAG identifies $\mathcal{S}$ for the purpose of enhancing the extractor model $\mathcal{E}$'s attention on these key elements.

Incorporating Self-Recognition in $\mathfrak{E}$. In Complex tasks such as the multi-hop, the query requires knowledge pieces with multi-hop relationships. To better recognize the multi-hop contextual links during extraction, FRAG incorporates the Self-Recognition method. This method (1) updates the key snippets $\mathcal{S}$ and (2) extracts inference-based knowledge by leveraging historically extracted knowledge as contextual references, formally defined as:

$$\mathcal{S}' = (Q \oplus \mathcal{K}) \cap A^{-1}, \tag{A.9}$$

$$k_{\text{in-B}}^i = (Q \oplus \mathcal{S} \oplus k_{\text{P-B}}^{(1,\dots,l)}) \otimes d_i, \tag{A.10}$$

where $\mathcal{S}'$ denotes the updated key snippets, reidentified from Q based on historically extracted knowledge. An illustrative example is shown in Figure 1. $\mathfrak{E}$ then performs snippet-level extraction using CoT and few-shot prompt engineering. The prompt instances is provided in Appendix I.1.

### B.2.2  MODULE: VALIDATOR $\mathfrak{V}$

Since hallucinations (Xu et al., 2024b) in LLMs can negatively impacts the extraction, it is crucial to validate whether the extracted sequences $\mathcal{R}$ remain consistent with the original retrievals to avoid factual errors. To this end, $\mathfrak{V}$ adopts a hybrid validation strategy that combines: (1) *string-level match* using a program executor $\mathcal{P}$; (2) *semantic-level match* using $\mathcal{E}$. Formally, this is defined as:

$$x_i = \begin{cases} x_i & \text{if } x_i \in_s d_i; \\ \text{None}, & \text{otherwise}. \end{cases} \tag{A.11}$$

where $\in_s$ represents string-level or semantic-level consistent.

### B.2.3  MODULE: PREFIXER $\mathfrak{P}$

$\mathfrak{P}$ is introduced to ensure the *semantic completeness* of $\mathcal{R}$ by *0*, as determined by $\mathcal{E}$. This design is motivated by the observation that $\mathcal{R}$ may omit critical contextual information present in the original retrievals. An example illustrative from PopQA is provided below:

> **Rawly extracted** $x_i$: *He was an English Conservative Party politician.*
>
> $\Downarrow$ *(Prefixer)*
>
> **Prefixed** $x_i$: *Henry Feilden (Conservative politician): He was an English Conservative politician.*

### B.2.4  MODULE: DEDUPLICATOR $\mathfrak{D}$

Similar to $\mathfrak{V}$, $\mathfrak{D}$ verifies whether each sequence in $\mathcal{R}$ is string- or semantically consistent with $\mathcal{K}$, which contains the historically extracted relevant knowledge. However, $\mathfrak{D}$ discards sequences in $\mathcal{R}$ that are already consistent with $\mathcal{K}$, thus further filtering noise in $\mathcal{K}$. Formally, this is defined as:

$$x_i = \begin{cases} \text{None}, & \text{if } x_i \in_s \mathcal{K}; \\ x_i, & \text{otherwise}. \end{cases} \tag{A.12}$$

### B.2.5  MODULE: FILTER $\mathfrak{F}$

$\mathfrak{F}$ assesses a relevance score of each sequence in $\mathcal{R}$ to Q, and discards those sequences with low relevance (default threshold: 0) to further reduce noise in $\mathcal{K}$. $\mathcal{S}$ is adding to the assessing process to enhance the snippet level query relevance. Scoring is guided by chain-of-thought (CoT) reasoning and few-shot prompt engineering, and is performed by $\mathcal{E}$. Self-Recognition is incorporated in $\mathfrak{F}$ to provide multi-hop contextual links, thereby assessing the relevance of inference-based knowledge more accurately.

$$s^i_{\text{in-B}} = (Q \oplus \mathcal{S} \oplus k^{(1,\dots l)}_{\text{P-B}}) \odot x_i = (Q \oplus k^{(1,\dots l)}_{\text{P-B}}) \odot x_i + (\mathcal{S} \oplus k^{(1,\dots l)}_{\text{P-B}}) \odot x_i, \tag{A.13}$$

Relevance scores are normalized to the range [0,1] and discretized into five levels: 1, 0.75, 0.5, 0.25, and 0, representing decreasing degrees of relevance. The sequences with $s_i > 0$ are included in basic knowledge $\mathcal{K}$ (Section 3.3). The prompt instances is provided in Appendix I.1.

### B.2.6  MODULE: ASSESSOR $\mathfrak{S}$

$\mathfrak{S}$ serves as an early-stop controller that determines when to terminate the extraction. The terminating condition depends on the following criteria: (1) whether the extracted knowledge is sufficient enough to solve the query, or (2) whether there is no further relevant knowledge can be extracted – namely, no more knowledge in the latest extraction round is extracted, or (3) whether the maximum extraction rounds $L_{\text{Max-Rounds}}$ has been reached. The stopping signal is formally defined as:

$$\tilde{s} = \begin{cases} 1, & \text{if } \tilde{a} = 1 \text{ or } \mathcal{K} \geq \mathcal{K}_{\text{prior}} \text{ or } r \geq L_{\text{Max-Rounds}}, \\ 0, & \text{otherwise}. \end{cases} \tag{A.14}$$

where $\tilde{s}$, $\tilde{a}$ denote the *stop* and *Answerable* flag, respectively; $r$ represents the current extraction round. And:

$$\tilde{a} = \mathbb{I}\left[Q \xrightarrow{\mathcal{K}} A\right] \tag{A.15}$$

where $\mathbb{I}[\cdot]$ denotes the indicator function that outputs 1 if the condition holds, and 0 otherwise; A signifies the query answer.

### B.3 DETAILS OF FRAG GENERATION

The basic knowledge $\mathcal{K}$ extracted by FRAG extraction framework contains knowledge varying from high relevance to low relevance. To evaluate the performance of RALMs across knowledge with varying relevance, we set a relevance threshold for generation $\mathcal{T}_G$ for the generation process. The knowledge with a relevance not lower than $\mathcal{T}_G$ is selected as the basic knowledge for generation ($\mathcal{K}'$) to construct the generation context to answer the query.

$\mathcal{T}_G$ has four threshold levels: 0.95, 0.7, 0.45, and 0.2. The reason for the slight difference between the $\mathcal{T}_G$ threshold settings and the scoring levels which $\mathfrak{F}$ set is that arithmetic performed by large models is not always reliable, and there are occasional arithmetic errors in calculating the final relevance scores, leading to inaccurate scoring. To mitigate the adverse effects of such occurrences, we slightly lowered the $\mathcal{T}_G$ threshold during the design process. Before generation, $\mathcal{K}'$ is also sorted in descending order of relevance score.

$$\mathcal{K}' = [(k'_u, s'_u), \ldots, (k'_v, s'_v)], \quad s'_u \geq s'_{u+1} \geq \cdots \geq s'_v \geq \mathcal{T}_G, \tag{A.16}$$

It should be noted that when there is no relevant knowledge in $\mathcal{K}'$, the initial retrieved documents will be used in the generation process. The final output is produced by the generator model $\mathcal{G}$. As is shown in the following formula:

$$\{Q, \mathcal{K}_G\} \xrightarrow{\mathcal{G}} A \tag{A.17}$$

where:

$$\mathcal{K}_G = \begin{cases} \mathcal{K}', & \text{if } \mathcal{K}' \neq \emptyset \\ D_s, & \text{otherwise.} \end{cases} \tag{A.18}$$

### B.4 FRAG-IP: A WRAPPER FINE-TUNED FRAMEWORK FOR OPTIMIZING COMPUTATIONAL EFFICIENCY

#### B.4.1 OPTIMIZING STRATEGY OF FRAG-IP

**Analysis of FRAG computational efficiency.** Given an initially retrieved document $D_s = [d_1, \ldots, d_N]$ containing $n$ passages, FRAG requires $(n+1) \sim [(n \times 5 + 2) \times L_{\text{Max-Rounds}} - 1]$ LLM calls to conduct extraction. The upper bound reflects the worst-case scenario; however, in most cases, only a small fraction of passages contain relevant information, which is commonly observed in practice. In addition, FRAG encounters data concurrency challenges due to inconsistent synchronized scheduling across modules during multi-query extraction. Moreover, the extraction process for a multi-hop query involves step-wise context dependencies, which constrain the potential for parallel extraction in such tasks. Those result in substantial computational overhead.

To address this challenge, firstly, we propose a wrapper framework which featuring *an asynchronous extraction pipeline*, orchestrated by a centralized hub scheduler. This design enables large-scale, asynchronous scheduling of multi-query extraction tasks in single GPU (for instance, FRAG can handle batch sizes of 256–512 queries on a single RTX 4090 GPU with this approach, in contrast to the batch size of 1 in the serial strategy). The wrapper framework not only significantly resolves data concurrency bottlenecks but also boosts extraction throughput by approximately 4–8×.

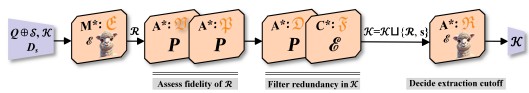

Figure A.6: **Workflow of FRAG-ip extraction framework.** $\mathcal{P}$ signifies the program executor. Modules $\mathfrak{V}, \mathfrak{P}, \mathfrak{D}$ are transformed to be purely driven by $\mathcal{P}$.

**FRAG-ip, a framework for significantly enhancing computational efficiency.** Established on the wrapper framework, secondly, we further propose *FRAG-ip*, a wrapper fine-tuned framework which applies a dual-stage fine-tuning strategy to the extractor model (the detailed dual-stage fine-tuning strategy is present in Section 3.3). This approach enables more effective and consistent recognition and extraction of relevant sequences from a batch of $t$ retrieved passages. Moreover, the modules $\mathfrak{V}, \mathfrak{D}$ to be transformed to be purely driven by $\mathcal{P}$. Notably, $\mathfrak{P}$ can be purely driven by $\mathcal{P}$ as well, since instances in typical datasets contain a structured "title" field. These improvements notable simplify the extraction process. Figure A.6 illustrates the workflow of the improved FRAG extraction framework, and Algorithm A.2 illustrates the FRAG-ip extraction.

The fine-tuned extractor model $\mathcal{E}_T$ extracts relevant and consistent sequences from a batch of $t$ retrieved passages and significantly reduces the LLM calls required for extraction to $(\lceil \frac{n}{t} \rceil + 1) \sim [(\lceil \frac{n}{t} \rceil \times 2 + 2) \times L_{\text{Max-Rounds}} - 1]$. This further decreases the extraction runtime by about $10\times$.

---

**Algorithm A.2** Improved FRAG Extraction

---

**Require:** Query $Q$; $Q$'s key sequence snippets $\mathcal{S}$; Retrieved documents $D_s = [d_1, \ldots, d_N]$; FRAG extraction framework $\mathcal{F}$; Basic knowledge $\mathcal{K}$; Batch size of passages to be extracted per step $t$; Relevant sequences set $\{x_i, \ldots, x_{i+m}\}$ extracted a batch of $t$ passages; Relevance score of $\{Q, x_i\}$, $s_i$; Max extraction rounds limit $L_{\text{Max-Rounds}}$; The $i$-th retrieved passage interval $\tilde{d}_{i \times t}$ (obtained by dividing $D_s$ into $t$ equal parts); Constraint: $\mathcal{K}$ does not exceed $\mathcal{K}_{\text{prior}}$, i.e., $\mathcal{K} \leq \mathcal{K}_{\text{prior}}$.

    **Initialize ans** $= false$; **stop** $= false$; $r = 1$;

    $\mathcal{S} \overset{\mathcal{F}}{\leftarrow} Q$          ▷ Decompose $\mathcal{S}$ from $Q$ (Section 3.1)

    **while** {**ans** is $false$ **and stop** is $false$ **and** $r \leq L_{\text{Max-Rounds}}$} **do**

        $\mathcal{K}_{\text{prior}} \leftarrow \mathcal{K}$

        **for** $i = 1, \ldots, \lceil n/t \rceil$ **do**

            **if** $\mathcal{K}$ is $\varnothing$ **then**

                $[(x_i, s_i)], \ldots, [(x_{i+m}, s_{i+m})] \overset{\mathcal{F}}{\leftarrow} \{Q \oplus \mathcal{S}, \tilde{d}_{i \times t}\}$

                     ▷ Extract $\{x_i, \ldots, x_{i+m}\}$ from $t$ passages & Assess $\{s_i, \ldots, s_{i+m}\}$ (Section 3.3)

            **else**

                $[(x_i, s_i)], \ldots, [(x_{i+m}, s_{i+m})] \overset{\mathcal{F}}{\leftarrow} \{Q \oplus \mathcal{S}, \tilde{d}_{i \times t}, \mathcal{K}\}$ ▷ Self-Recognition (Section 3.2)

        $\mathcal{K} \leftarrow \mathcal{K} \cup \{(x_j, s_j), \ldots, (x_k, s_k)\}$, **with** $s_i, \ldots, s_k > 0$

        $\mathcal{S} \overset{\mathcal{F}}{\leftarrow} Q \oplus \mathcal{K}$; $r \leftarrow r + 1$          ▷ Reidentify $\mathcal{S}$ if necessary

        **Assess** {**ans**: $true$ **or** $false$} $\overset{\mathcal{F}}{\leftarrow} \{Q, \mathcal{K}\}$      ▷ Assess if $Q$ is answerable based on $\mathcal{K}$

        **if** $\mathcal{K} \leq \mathcal{K}_{\text{prior}}$ **then**

            **stop** $\leftarrow true$          ▷ No more Basic Knowledge extracted

    Return $\mathcal{K}$

---

### B.4.2   DUAL-STAGE FINE-TUNING PROCESS

**Fine-tuning datasets construction.** For training dataset construction, we select 2,000 instances from 2WikiMultihopQA (Ho et al., 2020) and 1,000 instances from MuSiQue (Trivedi et al., 2022) as the raw instances. For the validation datasets, we sample 300 and 200 instances from the respective datasets. We utilize DeepSeek-v3 (Liu et al., 2024a) as generator to establish the fine-tuning datasets. For the first stage, the model is instructed to recognize the key sequence snippets within the raw instances, and subsequently extract relevant sequences $\mathcal{S}$ from the given $t$ passages using snippet-level query relevance ($t = 5$ in our fine-tuning). We validate the consistency of extracted sequences by $\mathcal{P}$ at string level. The generated training and validation datasets for this stage comprise 14410 and 2345 instances, respectively. For the second stage, each extracted sequence is prefixed with its corresponding main topic or title via $\mathcal{P}$, and the model is prompted to assign a relevance score of 1, 0.5, or 0 to each sequence, corresponding to relevant, partially relevant and irrelevant. Subsequently, the model is instructed to assess whether the queries in the raw instances are "Answerable", "Partially Answerable", or "Unanswerable", based on the basic knowledge extracted. The generated training and validation datasets for this stage is 7488 and 1203, respectively. The prompts for datasets generation are present in Appendix I.1.

**Fine-tuning loss and framework.** We adopt the token-level cross-entropy loss (Zhang & Sabuncu, 2018) in the fine-tuning, as formally defined:

$$\mathcal{L} = -\frac{1}{T_{\text{eff}}} \sum_{t=1}^{T} \mathbb{I}_{\{x_{t+1} \neq \text{PAD}\}} \cdot \log \left( \frac{\exp(z_{t,x_{t+1}})}{\sum_{j=1}^{V} \exp(z_{t,j})} \right) \tag{A.19}$$

where $T$ is the length of the input sequence; $x_{t+1}$ is the target token to predict at time step $t$; $z_{t,j}$ is the logit output by the model at time step $t$ for the $j$-th vocabulary token; $V$ is the vocabulary size; $\mathbb{I}_{\{x_{t+1} \neq \text{PAD}\}}$ is an indicator function to mask out padding tokens; and $T_{\text{eff}}$ is the number of effective tokens (i.e., non-padding) used for normalization. We leverage the Huggingface PEFT (Parameter-Efficient Fine-Tuning) framework (Mangrulkar et al., 2022) with the LoRA (Hu et al., 2022) strategy to enable efficient adaptation. The base model used for fine-tuning is Qwen2-7B-Instruct (Yang et al., 2024).

**Fine-tuning parameters settings and hardware configuration.** The Lora rank is set to 32, resulting in approximately 80.1M trainable parameters within the LoRA adapters, accounting for around 1.05% of the full 7.7B model parameters. Fine-tuning is conducted on 7 RTX A100 GPUs (40GB) with 5/1 epochs for the dual stages, respectively (notably, the second-stage fine-tuning converges rapidly due to the initialization from the first stage, and early stopping is recommended to mitigate catastrophic forgetting in this stage). The total fine-tuning process requires approximately 18.58 GPU-hours.

### B.4.3 COMPUTATIONAL EFFICIENCY OF FRAG AND RESULTS WITH FRAG-IP

Table A.5: **Overall experiment results using FRAG-ip. Bold** numbers indicate the best result across the compared models. $t$ signifies the number of passages to be extracted per step.

| LMs | Single-hop | | | Multi-hop | | | | | | Long-Context |
|---|---|---|---|---|---|---|---|---|---|---|
| | Pop (acc) | Pub (acc) | ARC (acc) | HotPotQA (llm-em) | (em) | 2Wiki (llm-em) | (em) | MuSiQue (llm-em) | (em) | LongBench-v2 (acc) |
| Baselines with retrieval | | | | | | | | | | |
| Qwen2-7B-Instruct | 51.82 | 75.08 | 76.83 | 57.4 | 34.9 | 50.2 | 42.2 | 23.5 | 9.2 | 22.66 |
| *ours* | | | | | | | | | | |
| FRAG-Qwen2-7B-Instruct | **57.97** | **79.84** | **82.03** | **72.7** | **47.7** | **60.9** | 51.5 | **44.1** | **25.6** | **29.82** |
| FRAG-ip-Qwen2-7B ($t=1$) | 53.11 | 79.43 | 81.60 | 59.3 | 37.2 | 58.6 | 52.1 | 27.9 | 14 | 27.24 |
| FRAG-ip-Qwen2-7B ($t=5$) | 50.82 | 76.09 | 80.24 | 58.4 | 34.5 | 58.2 | **52.4** | 21.8 | 12 | 27.04 |

**FRAG-ip performance.** As demonstrated in Table A.5, when extracting from a batch of passages with $t = 1$, FRAG-ip achieves a performance close to that of FRAG, with an accuracy loss of 0.5%–5% in most cases. Meanwhile, as present in Table A.6, it significantly improves computational efficiency by approximately 10× compared to FRAG wrapped in a pipeline. Moreover, it approaches the efficiency level of naive RAG while yielding significantly better performance.

Table A.6: Comparison of FRAG-ip and FRAG in Terms of Computational Efficiency. **Bold** numbers indicate the minimum average runtime on the test dataset.

| LMs | PopQA | | | | HotPotQA | | | |
|---|---|---|---|---|---|---|---|---|
| | $bs$ | $T$ | $L_{\text{mr}}$ | $n_{\text{ar}}$ | $bs$ | $T$ | $L_{\text{mr}}$ | $n_{\text{ar}}$ |
| Qwen2-7B-Instruct | 512 | 0.47 | - | - | 512 | 0.5 | - | - |
| FRAG-Qwen2-7B-Instruct w/o pipeline | 1 | 85.03 | 1 | 1 | 2 | 121.35 | 3 | 2.65 |
| FRAG-Qwen2-7B-Instruct w pipeline | 256 | 11.47 | 1 | 1 | 256 | 15.07 | 3 | 2.65 |
| FRAG-ip-Qwen2-7B ($t=1$) | 256 | 1.46 | 1 | 1 | 256 | 1.52 | 3 | 2.98 |
| FRAG-ip-Qwen2-7B ($t=5$) | 256 | **1.02** | 1 | 1 | 256 | **1.07** | 3 | 2.96 |

**Note**: $bs$ represents the *batch size* for FRAG extraction or naive RAG generation; $T$ denotes the average seconds spent per query; $L_{\text{mr}}$ stands for the maximum extraction rounds limit ($L_{\text{Max-Rounds}}$); $n_{\text{ar}}$ represents the average number of extraction rounds per query.

In addition, although the extractor model is fine-tuned on a relatively small dataset – consisting of only 3000 instances sampled from 2WikiMultihop and MuSiQue – it achieves a competitive performance on the single-hop, long-context and the 2WikiMultihop tasks with FRAG. This demonstrates the strong generalizability of FRAG in enhancing RALM performance by filtering noise using snippet-level query relevance. Furthermore, these results suggest promising potential for further accuracy improvements using the wrapper fine-tuning framework with the extractor model trained on larger and more diverse datasets.

It is worth noting that when $t = 5$, computational efficiency can be further improved, albeit at the cost of increased accuracy degradation compared with FRAG. Given that the efficiency gain at $t = 1$ is sufficient for most practical applications, we recommend choosing a small value of $t$ to avoid notable performance degradation.

Table A.7: Latency Comparison between FRAG and Other Baselines (Time per Query in Seconds) with Matched Backbones

| Models | Singlehop: PopQA (s) | Multihop: HotpotQA (s) |
|---|---|---|
| **Backbone: Llama3-8B-Instruct** | | |
| Llama3-8B-Instruct | 0.53 | 0.66 |
| ChatQA1.5-8B | 0.44 | 0.52 |
| ActiveRAG-8B | 66.10 | 100.10 |
| **Backbone: Qwen2-7B-Instruct** | | |
| Qwen2-7B-Instruct | 0.47 | 0.50 |
| ReComp-7B | 1.07 | 1.21 |
| RankGPT-7B | 3.70 | 4.45 |
| LongRAG-7B | 10.64 | 12.71 |
| RQ-RAG-7B | 29.68 | 32.12 |
| **Ours** | | |
| FRAG-Llama3-8B-Instruct | 5.13 | 13.87 |
| FRAG-Qwen2-7B-Instruct | 4.81 | 15.07 |
| FRAG-ip-Qwen2-7B | 1.02 | 1.07 |

Table A.8: Computational Cost of FRAG.

| Models | Task | Peak Memory Consumption | Concurrent Throughput | Average Latency |
|---|---|---|---|---|
| | | (GiB) | (queries/min) | (second) |
| **on RTX4090-24GiB** | | | | |
| FRAG-Qwen2-7B-Instruct | PopQA | 20.10 | 14.12 | 4.25 |
| FRAG-Qwen2-7B-Instruct | HotPotQA | 21.17 | 4.49 | 13.37 |
| FRAG-ip-Qwen2-7B | PopQA | 19.23 | 60.61 | 0.99 |
| FRAG-ip-Qwen2-7B | HotPotQA | 20.83 | 59.41 | 1.01 |
| **on A100-40GiB** | | | | |
| FRAG-Qwen2-7B-Instruct | PopQA | 37.55 | 12.47 | 4.81 |
| FRAG-Qwen2-7B-Instruct | HotPotQA | 38.87 | 3.98 | 15.07 |
| FRAG-ip-Qwen2-7B | PopQA | 37.55 | 58.82 | 1.02 |
| FRAG-ip-Qwen2-7B | HotPotQA | 37.98 | 56.07 | 1.07 |

We also compare the latency of FRAG with other baselines, as show in Table A.7. Results show that FRAG achieves the best performance with only marginal latency overhead, while FRAG-ip significantly reduces latency with competitive performance.

## B.5    COMPARISON OF FRAG WITH CONCURRENT BASELINES

**Comparison with reranking baselines.** Reranking methods, as adopted in baselines such as RankGPT (Sun et al., 2023) and RankRAG (Yu et al., 2024c), represent another noise-filtering strategy. These approaches rely on reranking the retrievals using a reranking model and selecting the top-$k$ most relevant passages for generation, thereby mitigating the negative impact of noise and enhance the RALM performance.

Table A.9: Comparison of FRAG with RankRAG.

| Models | PopQA (acc) | HotPotQA (em/f1) | 2Wiki (em/f1) |
|---|---|---|---|
| **Baselines w/ retrievals** | | | |
| Llama3-RankRAG8B | 64.1 | 35.3/46.7 | 31.4/36.9 |
| Llama3-RankRAG70B | **65.4** | 42.7/55.4 | 38.2/43.9 |
| FRAG-Qwen2-7B-Instruct | 57.97 | **47.7/64.08** | **51.5/58.5** |

**Note**: We report the results of RankRAG on PopQA, HotpotQA, and 2Wiki from the original paper.

It can be observed in Table A.9 that, compared with RankRAG, FRAG-Qwen2 delivers substantially stronger performance on the two multi-hop tasks (HotpotQA and 2Wiki), despite its smaller model size, although it underperforms on the single-hop task (PopQA).

In contrast, FRAG fundamentally differs from these methods. As demonstrated in Table A.10, Reranking approaches inherently face a trade-off: selecting a small $k$ leads to information loss, while a large $k$ introduces excessive noise. Moreover, these methods suffer from the disability to filter intra-passage noise. *FRAG overcomes the disadvantages by performing snippet-level extraction and filtering noise in a fine-grained manner.*

Table A.10: Performance Comparison between FRAG and RankGPT.

| Models | PopQA (acc) | Pub (acc) | ARC-C (acc) | HotPotQA (em/f1) | 2Wiki (em/f1) | MuSiQue (em/f1) | LongBench-v2 (acc) |
|---|---|---|---|---|---|---|---|
| RankGPT-Qwen2@5 | 55.68 | 78.52 | **82.28** | 42.9/59.82 | 35.6/43.08 | 15.9/28.13 | 24.45 |
| RankGPT-Qwen2@10 | 57.76 | 77.81 | 80.58 | 40.7/58.43 | 39.2/47.33 | 17.7/29.6 | 19.88 |
| RankGPT-Qwen2@20 | 56.40 | 78.72 | 77.51 | – | – | 13.8/26.34 | 23.26 |
| FRAG-Qwen2-7B-Instruct | **57.97** | **79.84** | 82.03 | **47.7/64.08** | **51.5/58.5** | **25.6/40.2** | **29.82** |

**Note**: We use GPT-3.5-turbo as the reranker of RankGPT.

Furthermore, the variability in reranking quality can occasionally demote relevant information to lower positions, thereby complicating the selection of an appropriate $k$, as evidenced by the results on Pub and LongBench-v2.

**Comparison with query reformulation techniques.** The query reformulation techniques typically utilize LLMs to reformulate the input query in order to improve retrieval recall. However, these methods primarily focus on enhancing the retrieval phase and do not explicitly address noise filtering in the generation stage. This makes their design fundamentally different from that of FRAG, which aims at filtering noise in a fine-grained manner to mitigate the attention distraction in RALMs caused by excessive noise.

We also note that FineFilter (Zhang et al., 2025), SCMRAG (Agrawal et al., 2025), and our work FRAG were developed roughly concurrently and independently. FineFilter and SCMRAG constitute strong baselines for improving generation quality in RAG systems. In the revised version, we will explicitly acknowledge them as concurrent work and further clarify their relationship to FRAG in the related work section.

**Comparison with FineFilter.** FineFilter/CompSelect is a strong baseline for noise filtering. However, our setting and technical contributions differ in several important aspects. (1) FineFilter formulates noise reduction as sentence-level MinMax clue selection, whereas FRAG performs fine-grained extraction explicitly guided by key query snippets and offers attention-based theoretical analysis (Lemma 3.1 / Theorem 3.2) showing how such modeling mitigates attention drift—an aspect not addressed by FineFilter. (2) FineFilter conditions extraction on the full query treated as a black-box input, whereas FRAG first decomposes the query into key vs. non-key snippets and conditions both extraction and filtering specifically on the key snippets, which drive the entire pipeline. (3) For complex multi-hop QA, FineFilter relies primarily on training objectives informed by generator feedback, while FRAG introduces a dedicated Self-Recognition mechanism that conditions the

extraction of new evidence on previously extracted knowledge, explicitly supporting inference-based multi-hop reasoning; ablations confirm that this component is critical on multi-hop benchmarks. (4) Architecturally, FineFilter adopts a three-module MinMax framework, whereas FRAG proposes a six-module LLM-based extraction architecture and further accelerates it through FRAG-ip, a wrapper framework that reduces extraction steps and bypasses certain validation calls, yielding improvements in both accuracy and efficiency.

**Comparison with SCMRAG.** We also conducted a detailed comparison between FRAG and SCM-RAG. Conceptually, SCMRAG focuses on a different stage of the RAG pipeline: it introduces an LLM-assisted dynamic knowledge graph and a self-corrective agent loop that detects when the current graph is incomplete or outdated and retrieves missing information from external sources for multi-hop questions. In contrast, FRAG assumes a fixed retriever and addresses the problem of fine-grained noise filtering within the retrieved context. Our technical contributions differ from SCMRAG's in several ways. (1) FRAG explicitly models key-snippet–based query relevance: it decomposes the query into key and non-key snippets, uses the key snippets to drive snippet-level extraction and filtering, and provides attention-based theoretical analysis demonstrating how this mitigates attention drift. (2) For multi-hop reasoning, SCMRAG's self-corrective mechanism operates at the retrieval/KB level, whereas FRAG introduces a Self-Recognition mechanism at the snippet-extraction level, conditioning the extraction of new relevant snippets on previously extracted evidence to preserve inference-driven multi-hop reasoning. (3) Architecturally, FRAG proposes a six-module LLM-based extraction framework and accelerates it via FRAG-ip, which reduces extraction steps and skips certain validation calls, improving both performance and efficiency. These contributions are new relative to SCMRAG's dynamic-graph, self-corrective retrieval paradigm.

## C  PROOF OF MAIN THEORETICAL ANALYSIS

### C.1  PROOF OF THEOREM 2.2

*Proof.* Calculate $\mathbf{A}_{\mathfrak{T}}$ for $\mathfrak{T}$ (Vaswani et al., 2017) ($W_Q$, $W_K$, and $W_V$ represent the pre-trained Q, K, V matrices of the LLM):

$$\mathbf{A}_{\mathfrak{T}}\big[Q_{\mathrm{Q}}, (K_{\mathrm{rel}} \oplus K_{\mathrm{red}}), (V_{\mathrm{rel}} \oplus V_{\mathrm{red}})\big] = \mathrm{SoftMax}\left(\frac{Q_{\mathrm{Q}}(K_{\mathrm{rel}}^{\top}, K_{\mathrm{red}}^{\top})}{\sqrt{d_k}}\right)(V_{\mathrm{rel}} \oplus V_{\mathrm{red}})$$

$$= \mathrm{SoftMax}\left(\frac{[X_Q W_Q W_K^{\top} X_{\mathrm{rel}}^{\top}, X_Q W_Q W_K^{\top} X_{\mathrm{red}}^{\top}]}{\sqrt{d_k}}\right) \cdot (X_{\mathrm{rel}} W_{\mathrm{rel}} \oplus X_{\mathrm{red}} W_{\mathrm{red}})$$

$$= \mathrm{SoftMax}\big([M, N]\big)(X_{\mathrm{rel}} W_V \oplus X_{\mathrm{red}} W_V) = \boldsymbol{\alpha}(X_{\mathrm{rel}} W_V \oplus X_{\mathrm{red}} W_V), \tag{A.20}$$

where:

$$\mathbf{M} = \begin{bmatrix} a_{(1,1)} & \cdots & a_{(1,n_{\mathrm{rel}})} \\ \vdots & \ddots & \vdots \\ a_{(n_q,1)} & \cdots & a_{(n_q,n_{\mathrm{rel}})} \end{bmatrix}, \mathbf{N} = \begin{bmatrix} b_{(1,1)} & \cdots & b_{(1,n_{\mathrm{red}})} \\ \vdots & \ddots & \vdots \\ b_{(n_q,1)} & \cdots & b_{(n_q,n_{\mathrm{red}})} \end{bmatrix},$$

$$\boldsymbol{\alpha} = \begin{bmatrix} \alpha_{\mathrm{rel}(1,1)} & \cdots & \alpha_{\mathrm{rel}(1,n_{\mathrm{rel}})} & \alpha_{\mathrm{rel}(1,1)} & \cdots & \alpha_{\mathrm{red}(1,n_{\mathrm{red}})} \\ \vdots & \ddots & \vdots & \vdots & \ddots & \vdots \\ \alpha_{\mathrm{rel}(n_q,1)} & \cdots & \alpha_{\mathrm{rel}(n_q,n_{\mathrm{rel}})} & \alpha_{\mathrm{red}(n_q,1)} & \cdots & \alpha_{\mathrm{red}(n_q,n_{\mathrm{red}})} \end{bmatrix}. \tag{A.21}$$

Thus we have:

$$\alpha_{\mathrm{rel}(i,j)} = \frac{e^{a_{(i,j)}}}{\sum_{j=1}^{n_{\mathrm{rel}}} e^{a_{(i,j)}} + \sum_{j=1}^{n_{\mathrm{red}}} e^{b_{(i,j)}}}, \alpha_{\mathrm{red}(i,j)} = \frac{e^{b_{(i,j)}}}{\sum_{j=1}^{n_{\mathrm{rel}}} e^{a_{(i,j)}} + \sum_{j=1}^{n_{\mathrm{red}}} e^{b_{(i,j)}}}, \tag{A.22}$$

and:

$$\sum_{j=1}^{n_{\mathrm{rel}}} \alpha_{\mathrm{rel}(i,j)} + \sum_{j=1}^{n_{\mathrm{red}}} \alpha_{\mathrm{red}(i,j)} = 1, \quad \forall i \in [1, n_q]. \tag{A.23}$$

Let $n_{\text{rel}}$ remains constant. Since: $0 < \alpha_{\text{rel}(i,j)}, \alpha_{\text{red}(i,j)} < +\infty$, we can conclude that as $n_{\text{red}}$ grows, $\sum_{j=1}^{n_{\text{rel}}} \alpha_{\text{rel}(i,j)}$ decreases, while $\sum_{j=1}^{n_{\text{red}}} \alpha_{\text{red}(i,j)} = 1 - \sum_{j=1}^{n_{\text{rel}}} \alpha_{\text{rel}(i,j)}$ increases. Thus, there exists $\tilde{n} \in \mathbb{N}^+$, such that for $n_{\text{red}} > \tilde{n}$, we have:

$$\sum_{i=1}^{n_q} \sum_{j=1}^{n_{\text{rel}}} \alpha_{\text{rel}(i,j)} < \sum_{i=1}^{n_q} \sum_{j=1}^{n_{\text{red}}} \alpha_{\text{red}(i,j)}. \tag{A.24}$$

Namely, we have:

$$\mathfrak{A}_{\text{rel}} < \mathfrak{A}_{\text{red}}. \tag{A.25}$$

In this case, the noise dilutes the relevant information, causing the RALM to allocate more attention to $X_{\text{red}}$ than $X_{\text{rel}}$, and generate $\mathfrak{T}$ based on $X_{\text{red}}$, subsequently producing content unrelated to answering $Q$. For example, it may respond with information from $X_{\text{red}}$ or provide ambiguous answers such as "Not enough information to answer the query". □

## C.2    PROOF OF DISCUSSION IN APPENDIX B.1

*Proof. The average proportion of relevant information in the $t$ relevant retrieval passages is higher than in a long-context document composed of $N$ passages.* For the initial document containing $N$ passages, the noise rate is expressed as follows:

$$\rho_0 = \frac{\frac{n_{\text{red}}}{N}}{\frac{n}{N}} = \frac{n_{\text{red}}}{n}. \tag{A.26}$$

Next, for a single extraction step from a single passage, assume that the sequences forming $X_{\text{rel}}$ and $X_{\text{red}}$ are uniformly distributed across all passages. The expected value of the noise rate for a single passage is given by:

$$\mathbb{E}(\rho_{\text{red}}^*) = \bar{\rho}_{\text{red}}^* = \frac{\frac{n_{\text{red}}}{N}}{\frac{n}{N}} = \frac{n_{\text{red}}}{n} = \rho_0. \tag{A.27}$$

However, since the relevant information is concentrated in $t$ paragraphs (where $t \le N$), we assume that the sequences forming $X_{\text{rel}}$ are uniformly distributed across these $t$ retrieval passages. Thus for the $t$ relevant passages, the expected noise rate is given by:

$$\mathbb{E}(\rho_{\text{red}}) = \bar{\rho}_{\text{red}} = \frac{\frac{n}{N} - \frac{n_{\text{rel}}}{t}}{\frac{n}{N}} = 1 - \frac{n_{\text{rel}}}{n} \cdot \frac{N}{t} = 1 - \frac{n - n_{\text{red}}}{n} \cdot \frac{N}{t} = 1 - \left[ (1 - \rho_0) \cdot \frac{N}{t} \right] \le \rho_0. \tag{A.28}$$

Therefore, decomposing the process into several steps, each focusing on extracting information from a single retrieval passage, not only reduces the length of noisy information but also lowers the proportion of noise within the $t$ relevant passages. Accordingly, for these $t$ relevant passages, the proportion of relevant information is higher than that in the initial retrieval document. This, in turn, mitigates the attention diversion caused by noisy information during the extraction process. □

## C.3    PROOF OF LEMMA 3.1

*Proof. Negative impact of non-key sequence snippets in the query $X_{Q^-}$.* During the extraction process, the LLM's attention distribution over the retrieval passage $X_{\text{rel},\subset} \oplus X_{\text{red},\subset}$, based on $X_Q$, is

as follows:

$$\mathbf{A}_{\mathfrak{T}}[(Q, (K_{\text{rel},\subset} \oplus K_{\text{red},\subset}), (V_{\text{rel},\subset} \oplus V_{\text{red},\subset}))]$$

$$= \text{SoftMax}\left[\frac{(Q_{Q^+} \oplus Q_{Q^-})[K_{\text{rel},\subset}^T, K_{\text{red},\subset}^T]}{\sqrt{d_k}}\right] \cdot (V_{\text{rel},\subset} \oplus V_{\text{red},\subset})$$

$$= \text{SoftMax}\left[[(X_{Q^+} \oplus X_{Q^-})W_Q W_K^T X_{\text{rel},\subset}^T, (X_{Q^+} \oplus X_{Q^-})W_Q W_K^T X_{\text{red},\subset}^T]/\sqrt{d_k}\right]$$

$$\cdot (X_{\text{rel},\subset} W_V \oplus X_{\text{red},\subset} W_V)$$

$$= \text{SoftMax}\left(\begin{bmatrix} M_{s0^+} & N_{s0^+} \\ M_{s0^-} & N_{s0^-} \end{bmatrix}\right)(X_{\text{rel},\subset}W_V \oplus X_{\text{red},\subset}W_V) = \boldsymbol{\alpha}_{\subset}(X_{\text{rel},\subset}W_V \oplus X_{\text{red},\subset}W_V)$$

$$= \begin{bmatrix} \boldsymbol{\alpha}_{\text{rel},\subset}^+, \boldsymbol{\alpha}_{\text{red},\subset}^+ \\ \boldsymbol{\alpha}_{\text{rel},\subset}^-, \boldsymbol{\alpha}_{\text{red},\subset}^- \end{bmatrix}(X_{\text{rel},\subset}W_V \oplus X_{\text{red},\subset}W_V). \tag{A.29}$$

It is evident that, under general circumstances, we have:

$$\frac{1}{n_{q^+}}\sum_{i=1}^{n_{q^+}}\sum_{j=1}^{n_l}\alpha_{\text{rel},\subset(i,j)}^+ \geq \frac{1}{n_{q^-}}\sum_{i=1}^{n_{q^-}}\sum_{j=1}^{n_l}\alpha_{\text{rel},\subset(i,j)}^-, \tag{A.30}$$

where $n_{q^+}, n_{q^-}$ denote the token counts of $X_{Q^+}, X_{Q^-}$, respectively. Accordingly, we have:

$$\frac{1}{n_{q^+}}\sum_{i=1}^{n_{q^+}}\left(1 - \sum_{j=1}^{n_l}\alpha_{\text{rel},\subset(i,j)}^+\right) \leq \frac{1}{n_{q^-}}\sum_{i=1}^{n_{q^-}}\left(1 - \sum_{j=1}^{n_l}\alpha_{\text{rel},\subset(i,j)}^-\right)$$

$$\Rightarrow \frac{1}{n_{q^+}}\sum_{i=1}^{n_{q^+}}\sum_{j=1}^{n_d}\alpha_{\text{red},\subset(i,j)}^+ \leq \frac{1}{n_{q^-}}\sum_{i=1}^{n_{q^-}}\sum_{j=1}^{n_d}\alpha_{\text{red},\subset(i,j)}^-. \tag{A.31}$$

Combining Inequality A.30 and A.31, we have:

$$a_1 = \frac{\sum_{i=1}^{n_{q^+}}\sum_{j=1}^{n_l}\alpha_{\text{rel},\subset(i,j)}^+}{\sum_{i=1}^{n_{q^-}}\sum_{j=1}^{n_l}\alpha_{\text{rel},\subset(i,j)}^-} \geq \frac{n_{q^+}}{n_{q^-}}, \tag{A.32}$$

$$b_1 = \frac{\sum_{i=1}^{n_{q^+}}\sum_{j=1}^{n_d}\alpha_{\text{red},\subset(i,j)}^+}{\sum_{i=1}^{n_{q^-}}\sum_{j=1}^{n_d}\alpha_{\text{red},\subset(i,j)}^-} \leq \frac{n_{q^+}}{n_{q^-}}. \tag{A.33}$$

Now, we attempt to prove that the Inequality A.34 holds:

$$\frac{\sum_{\substack{1 \leq i \leq n_{q^+} \\ 1 \leq j \leq n_l}}\alpha_{\text{rel}(i,j)}^+}{\sum_{\substack{1 \leq i \leq n_{q^+} \\ 1 \leq j \leq n_d}}\alpha_{\text{red}(i,j)}^+} \geq \frac{\sum_{\substack{1 \leq i \leq n_{q^+} \\ 1 \leq j \leq n_l}}\alpha_{\text{rel}(i,j)}^+ + \sum_{\substack{1 \leq i \leq n_{q^-} \\ 1 \leq j \leq n_l}}\alpha_{\text{rel}(i,j)}^-}{\sum_{\substack{1 \leq i \leq n_{q^+} \\ 1 \leq j \leq n_d}}\alpha_{\text{red}(i,j)}^+ + \sum_{\substack{1 \leq i \leq n_{q^-} \\ 1 \leq j \leq n_d}}\alpha_{\text{red}(i,j)}^-}, \tag{A.34}$$

where $n_l, n_d$ denote the token counts of $X_{\text{rel},\subset}, X_{\text{red},\subset}$, respectively.

Substitute $a_1$ and $b_1$ into Inequality A.34. Since $a_1 \geq \frac{n_{q^+}}{n_{q^-}} \geq b_1 > 0$, Inequality A.34, i.e., Inequality 1 is proven.

That is, in effect, the non-key sequence snippets of the query weaken the RALM's attention distribution of $X_Q$ towards the relevant information in the retrieval passage. $\qquad\square$

### C.4 PROOF OF THEOREM 3.2

*Proof. Benefits of snippet-Level query relevance in extraction.* By identifying and extracting the key sequence snippets $X'_{Q^+}$ from $Q$, we can explicitly add $X'_{Q^+}$, thereby increasing its corresponding

query weight matrix parameters and enhancing the LLM's attention distribution towards the relevant information. On the other hand, by increasing the proportion of key sequences during extraction, the LLM's attention distribution towards the relevant information in the retrieval passages is further enhanced. The analysis is shown as follows:

$$\boldsymbol{A}_{\mathfrak{T}}[(Q, (K_{\text{rel},\subset} \oplus K_{\text{red},\subset}), (V_{\text{rel},\subset} \oplus V_{\text{red},\subset}))]$$

$$= \text{SoftMax}\left[\frac{(Q_{Q^+} \oplus Q_{Q^-} \oplus Q'_{Q^+})[K_{\text{rel},\subset}^T, K_{\text{red},\subset}^T]}{\sqrt{d_k}}\right] \cdot (V_{\text{rel},\subset} \oplus V_{\text{red},\subset})$$

$$= \text{SoftMax}\left[[(X_{Q^+} \oplus X_{Q^-} \oplus X'_{Q^+})W_Q W_K^T X_{\text{rel},\subset}^T, (X_{Q^+} \oplus X_{Q^-} \oplus X'_{Q^+})W_Q W_K^T X_{\text{red},\subset}^T]/\sqrt{d_k}\right]$$

$$\cdot (X_{\text{rel},\subset} W_V \oplus X_{\text{red},\subset} W_V)$$

$$= \text{SoftMax}\left(\begin{bmatrix} M_{s0^+} & N_{s0^+} \\ M_{s0^-} & N_{s0^-} \\ M_{s'_+} & N_{s'_+} \end{bmatrix}\right)(X_{\text{rel},\subset}W_V \oplus X_{\text{red},\subset}W_V) = \boldsymbol{\alpha}'_s(X_{\text{rel},\subset}W_V \oplus X_{\text{red},\subset}W_V)$$

$$= \begin{bmatrix} \boldsymbol{\alpha}_{\text{rel},\subset}^+ & \boldsymbol{\alpha}_{\text{red},\subset}^+ \\ \boldsymbol{\alpha}_{\text{rel},\subset}^- & \boldsymbol{\alpha}_{\text{red},\subset}^- \\ \boldsymbol{\alpha}_{\text{rel},\subset}'^+ & \boldsymbol{\alpha}_{\text{red},\subset}'^+ \end{bmatrix}(X_{\text{rel},\subset}W_V \oplus X_{\text{red},\subset}W_V). \tag{A.35}$$

Combining Lemma 3.1, we have:

$$\frac{1}{n_{q'_+}}\sum_{i=1}^{n_{q'_+}}\sum_{j=1}^{n_l}\alpha_{\text{rel},\subset(i,j)}'^+ \geq \frac{1}{n_{q^+}}\sum_{i=1}^{n_{q^+}}\sum_{j=1}^{n_l}\alpha_{\text{rel},\subset(i,j)}^+ \geq \frac{1}{n_q}(\sum_{i=1}^{n_{q^+}}\sum_{j=1}^{n_l}\alpha_{\text{rel},\subset(i,j)}^+ + \sum_{i=1}^{n_{q^-}}\sum_{j=1}^{n_l}\alpha_{\text{rel},\subset(i,j)}^-);$$
$$\tag{A.36}$$

$$\frac{1}{n_{q'_+}}\sum_{i=1}^{n_{q'_+}}\sum_{j=1}^{n_d}\alpha_{\text{red},\subset(i,j)}'^+ \leq \frac{1}{n_{q^+}}\sum_{i=1}^{n_{q^+}}\sum_{j=1}^{n_d}\alpha_{\text{red},\subset(i,j)}^+ \leq \frac{1}{n_q}(\sum_{i=1}^{n_{q^+}}\sum_{j=1}^{n_d}\alpha_{\text{red},\subset(i,j)}^+ + \sum_{i=1}^{n_{q^-}}\sum_{j=1}^{n_d}\alpha_{\text{red},\subset(i,j)}^-).$$
$$\tag{A.37}$$

It is obviously that:

$$a_2 = \frac{\sum_{i=1}^{n_{q'_+}}\sum_{j=1}^{n_l}\alpha_{\text{rel},\subset(i,j)}'^+}{\sum_{i=1}^{n_{q^+}}\sum_{j=1}^{n_l}\alpha_{\text{rel},\subset(i,j)}^+ + \sum_{i=1}^{n_{q^-}}\sum_{j=1}^{n_l}\alpha_{\text{rel},\subset(i,j)}^-} \geq \frac{n_{q'_+}}{n_q}, \tag{A.38}$$

$$b_2 = \frac{\sum_{i=1}^{n_{q'_+}}\sum_{j=1}^{n_d}\alpha_{\text{red},\subset(i,j)}'^+}{\sum_{i=1}^{n_{q^+}}\sum_{j=1}^{n_d}\alpha_{\text{red},\subset(i,j)}^+ + \sum_{i=1}^{n_{q^-}}\sum_{j=1}^{n_d}\alpha_{\text{red},\subset(i,j)}^-} \leq \frac{n_{q'_+}}{n_q}. \tag{A.39}$$

Now, we attempt to prove that the Inequality A.40 holds:

$$\frac{\sum_{\substack{1\leq i\leq n_{q^+}\\1\leq j\leq n_l}}\alpha_{\text{rel}(i,j)}^+ + \sum_{\substack{1\leq i\leq n_{q^-}\\1\leq j\leq n_l}}\alpha_{\text{rel}(i,j)}^- + \sum_{\substack{1\leq i\leq n_{q'_+}\\1\leq j\leq n_l}}\alpha_{\text{rel}(i,j)}'^+}{\sum_{\substack{1\leq i\leq n_{q^+}\\1\leq j\leq n_d}}\alpha_{\text{red}(i,j)}^+ + \sum_{\substack{1\leq i\leq n_{q^-}\\1\leq j\leq n_d}}\alpha_{\text{red}(i,j)}^- + \sum_{\substack{1\leq i\leq n_{q'_+}\\1\leq j\leq n_d}}\alpha_{\text{red}(i,j)}'^+} \geq \frac{\sum_{\substack{1\leq i\leq n_{q^+}\\1\leq j\leq n_l}}\alpha_{\text{rel}(i,j)}^+ + \sum_{\substack{1\leq i\leq n_{q^-}\\1\leq j\leq n_l}}\alpha_{\text{rel}(i,j)}^-}{\sum_{\substack{1\leq i\leq n_{q^+}\\1\leq j\leq n_d}}\alpha_{\text{red}(i,j)}^+ + \sum_{\substack{1\leq i\leq n_{q^-}\\1\leq j\leq n_d}}\alpha_{\text{red}(i,j)}^-},$$
$$\tag{A.40}$$

where $n_{q'_+}$ denotes the token count of $X'_{Q^+}$.

Substitute $a_2$ and $b_2$ into Inequality A.40. Since $a_2 \geq \frac{n_{q'_+}}{n_q} \geq b_2 > 0$, Inequality A.40, i.e., Inequality 2 is proven.

This implies the LLM allocates a greater proportion of attention to $X_{\text{rel},\subset}$ when incorporating $X'_{Q^+}$ into $X_Q$ during extraction. Combining Lemma 3.1 and Inequality 2, we conclude that explicitly incorporating $X'_{Q^+}$ into $X_Q$ during extraction, i.e., performing extraction using snippet-level query relevance, improves the model's attention allocation towards relevant information. This reduces the attention reduction caused by $X_{Q^-}$, enhancing the extraction of relevant knowledge. $\square$

# D ADDITIONAL EXPERIMENT AND DETAILS

## D.1 ADDITIONAL EXPERIMENT

Table A.11: Comparison between FRAG and baselines using randomly clipped retrievals. Bold numbers indicate the best performance among the compared models. $r$ denotes the ratio of input tokens used for generation to the total number of tokens in the initial retrievals.

| Models | Pop (acc) | Pub (acc) | ARC (acc) | HotpotQA (em/f1) | 2Wiki (em/f1) | MuSiQue (em/f1) | LongBench-v2 (acc) |
|---|---|---|---|---|---|---|---|
| **Baselines w/ Randomly Clipped Retrievals** | | | | | | | |
| ChatQA-1.5-8B | 29.74 | 0.71 | 52.56 | 15.22/23.92 | 37.74/39.97 | 2.1/8.22 | 7.55 |
| Llama3-8B-Instruct | 28.16 | 70.92 | 75.30 | 12.11/17.12 | 5.21/9.22 | 2.2/5.34 | 19.28 |
| Qwen2-7B-Instruct | 25.95 | 73.86 | 77.94 | 17.22/24.44 | 16.82/20.42 | 4.4/9.16 | 24.25 |
| ChatQA-1.5-70B | 37.60 | 55.72 | 74.45 | 21.10/30.94 | 31.00/34.81 | 6.3/14.62 | 23.21 |
| **Ours** | | | | | | | |
| FRAG-ChatQA-1.5-8B | 54.11 | 73.66 | 46.42 | 27.9/44.81 | 44.2/48.0 | 12.4/23.7 | 21.47 |
| FRAG-Llama3-8B-Instruct | **59.83** | 75.99 | 77.34 | 41.7/58.82 | 32.1/38.7 | 22.9/33.9 | 24.45 |
| FRAG-Qwen2-7B-Instruct | 57.97 | **79.84** | 82.03 | **47.7/64.08** | **51.5/58.5** | **25.6/40.2** | **29.82** |
| FRAG-ChatQA-1.5-70B | 59.69 | 76.49 | **82.79** | 38.4/53.92 | 28.4/36.7 | 19.3/32.3 | 29.03 |

To investigate whether the performance gain of FRAG mainly stems from denoising rather than input length, we conducted an additional experiment. Specifically, we controlled the number of input tokens for naive RAG by randomly clipping retrievals to match the token budget of FRAG. As shown in Table A.11, FRAG still significantly outperforms these baselines, highlighting its effectiveness in improving RALMs by filtering out noise from the initial retrievals.

Table A.12: Performance comparison of FRAG and naive RAG under various values of $N$. Bold numbers denote the best performance among the compared models. For HotpotQA, when $N = 10$, we use the original passages associated with each query in the dataset as retrievals.

| Models | PopQA | | | HotpotQA | | | |
|---|---|---|---|---|---|---|---|
| | (recall) | (acc) | ($T$) | (recall) | (em) | (f1) | ($T$) |
| $N = 3$ | | | | | | | |
| Qwen2-7B-Instruct | - | - | - | 65.1 | 27.93 | 39.36 | 0.2 |
| FRAG-Qwen2-7B-Instruct | - | - | - | 65.1 | 31.03 | 42.31 | 6.0 |
| $N = 5$ | | | | | | | |
| Qwen2-7B-Instruct | 50.25 | 40.46 | 0.2 | 73.75 | 29.83 | 41.12 | 0.2 |
| FRAG-Qwen2-7B-Instruct | 50.25 | 42.53 | 1.4 | 73.75 | 33.23 | 44.43 | 11.6 |
| $N = 10$ | | | | | | | |
| Qwen2-7B-Instruct | 65.90 | 49.61 | 0.4 | 100 | 34.90 | 50.38 | 0.9 |
| FRAG-Qwen2-7B-Instruct | 65.90 | 51.25 | 2.6 | 100 | **47.70** | **64.08** | 15.07 |
| $N = 20$ | | | | | | | |
| Qwen2-7B-Instruct | 73.72 | 51.82 | 2.5 | - | - | - | - |
| FRAG-Qwen2-7B-Instruct | 73.72 | **59.69** | 11.47 | - | - | - | - |

Moreover, To explore how varying the retrieval window size $N$ impacts model performance, we conduct the experiments by evaluating different N values on PopQA and HotpotQA using both FRAG and naive RAG. The results are present in Table A.12. We further evaluate FRAG on DeepSeek-v3 (685B) to examine its performance on extremely large models and identify potential saturation points. As shown in Table A.15, FRAG continues to enhance the performance of very large models by effectively filtering noise, with particularly strong gains on complex multi-hop tasks. Table A.15 also shows that performance gains tend to saturate on simple single-hop tasks, and in cases where retrieval quality is notably low (e.g., LongBench-v2). Nevertheless, on complex multi-hop datasets, FRAG

still yields substantial improvements on extremely large models, achieving average gains of 8.23% EM and 6.53% F1.

We experimented with lightweight extractors—BERT (Devlin et al., 2019), ColBERT (Khattab & Zaharia, 2020), and BM25—to assess the effectiveness of FRAG when paired with different retrieval backbones. Since these models are not intrinsically suited for next-word prediction, we employ a preprocessing strategy that splits retrieved passages into individual sentences and scores their relevance to the query. The experimental results are reported in Table A.13, and representative extracted samples are shown below.

---

**Fine-grained Scoring Examples using Bert/ColBert/BM25**

**Query:** What is Bruce McDaniel's occupation?

**BERT-Large Passages:**
1. **[Gold]** Bruce McDaniel: Bruce McDaniel is an American musician, composer, producer and recording engineer, currently living in New Orleans [score=0.6616]. Bruce McDaniel was born in Boston, Massachusetts of Mexican and Scottish/American parents on 23 September 1962 and grew up in New York [score=-0.8173]. He was raised by musical parents who met while attending the Juilliard School of Music [score=0.9357]. . . .
2. [Noise] John McDaniel (born September 23, 1951 in Birmingham, Alabama) is a former American football wide receiver [score=-0.8538]. . . .
3. [Noise] Jerry McDaniel: McDaniel has also conceived and produced short films and film titles [score=0.9548]. . . .
4. [Noise] Bruce Smith: Bruce Smith was born in New York. Bruce's occupation is a boxer [score=0.7666]. . . .

**ColBert-v2.0 Passages:**
1. **[Gold]** Bruce McDaniel: Bruce McDaniel is an American musician, composer, producer and recording engineer, currently living in New Orleans [score=0.9479]. Bruce McDaniel was born in Boston, Massachusetts of Mexican and Scottish/American parents on 23 September 1962 and grew up in New York [score=0.9696]. He was raised by musical parents who met while attending the Juilliard School of Music [score=0.9670]. . . .
2. [Noise] John McDaniel: John McDaniel (born September 23, 1951 in Birmingham, Alabama) is a former American football wide receiver [score=0.9489]. . . .
3. [Noise] Jerry McDaniel: McDaniel has also conceived and produced short films and film titles [score=0.9627]. . . .
4. [Noise] Bruce Smith: Bruce Smith was born in New York. Bruce's occupation is a boxer [score=0.9806]. . . .

**BM25 Passages:**
1. **[Gold]** Bruce McDaniel: Bruce McDaniel is an American musician, composer, producer and recording engineer, currently living in New Orleans [score=0.2613]. Bruce McDaniel was born in Boston, Massachusetts of Mexican and Scottish/American parents on 23 September 1962 and grew up in New York [score=0.2187]. He was raised by musical parents who met while attending the Juilliard School of Music [score=0]. . . .
2. [Noise] John McDaniel: John McDaniel (born September 23, 1951 in Birmingham, Alabama) is a former American football wide receiver [score=0.2613]. . . .
3. [Noise] Jerry McDaniel: McDaniel has also conceived and produced short films and film titles [score=0.3958]. . . .
4. [Noise] Bruce Smith: Bruce Smith was born in New York. Bruce's occupation is a boxer [score=2.6977]. . . .

---

We observe that BERT struggles to discriminate relevant sentences from irrelevant ones, resulting in ineffective noise filtering. BM25, as a term-frequency–based method, also underperforms in suppressing noise, and both approaches can even introduce substantial information loss during extraction. These limitations motivate us not to adopt BERT or BM25 as extractor models. In contrast, LLMs exhibit strong reading and comprehension capabilities, enabling them to more reliably identify relevant sentences and assign accurate relevance scores. This leads to markedly improved extraction quality and overall performance.

We conducted additional experiments in which FRAG-ip is evaluated against compression- and noise-reduction baselines (RQ-RAG, LongRAG, RECOMP, and RankGPT) under the same backbone (Qwen2-7B), an identical retrieval setup (Contriever), and matched context budgets (top-k = 20). To

Table A.13: FRAG Performance Comparison Using Instruct LLM Extractor Models vs. BERT/BM25 Extractor Models

| Extractor | Generator | PopQA | Pub | ARC | HotPotQA | 2Wiki | MuSiQue | LongBenchv2 |
|---|---|---|---|---|---|---|---|---|
| | | (acc) | (acc) | (acc) | (em/f1) | (em/f1) | (em/f1) | (acc) |
| – | Qwen2-7B-Instruct | 51.82 | 75.08 | 76.83 | 34.9/50.38 | 42.2/49.57 | 9.2/19.52 | 22.66 |
| Bert | Qwen2-7B-Instruct | 25.45 | 73.86 | 77.77 | 26.1/37.41 | 26.2/29.93 | 8.2/15.25 | 21.47 |
| ColBert | Qwen2-7B-Instruct | 52.32 | 75.18 | 77.34 | 36.6/49.69 | 39.2/43.89 | 12/22.2 | 23.06 |
| BM25 | Qwen2-7B-Instruct | 52.32 | 75.68 | 77.26 | 36.4/50.4 | 45/49.5 | 7.5/17.2 | 21.47 |
| Llama3-8B-Instruct | Llama3-8B-Instruct | **59.83** | 75.99 | 77.34 | 41.7/58.82 | 32.1/38.72 | 22.9/33.26 | 24.45 |
| Qwen2-7B-Instruct | Qwen2-7B-Instruct | 57.97 | **79.84** | **82.03** | **47.7/64.08** | **51.5/58.5** | **25.6/40.2** | **29.82** |

Table A.14: Performance Comparison between FRAG-ip and Other Baselines Under Matched Compute and Context Budgets.

| Models | PopQA | Pub | ARC | HotPotQA | 2Wiki | MuSiQue | LB-v2 |
|---|---|---|---|---|---|---|---|
| | (acc) | (acc) | (acc) | (em/f1) | (em/f1) | (em/f1) | (acc) |
| **Baselines w/ Retrievals (Backbone Qwen2-7B-Instruct)** | | | | | | | |
| Qwen2-7B-Instruct | 51.82 | 75.08 | 76.83 | 34.9/50.38 | 42.2/49.57 | 9.2/19.52 | 22.66 |
| ReComp-7B | 53.61 | 75.08 | 77.51 | 35.7/50.8 | 40.1/47.96 | 8.1/17.3 | 24.65 |
| RankGPT-7B | **55.61** | 78.52 | 81.26 | **38.3**/53.14 | 37.6/45.43 | 12.2/24.37 | 23.26 |
| LongRAG-7B | 52.89 | 75.48 | 77.51 | 38.1/**53.88** | 39.2/46.71 | 12.1/22.95 | 23.26 |
| RQ-RAG-7B | 52.25 | – | 77.09 | 34.4/49.37 | 41.5/48.92 | 8.6/18.74 | 22.66 |
| FRAG-ip | 53.11 | **79.43** | **81.60** | 37.2/49.69 | **52.1/57.45** | **13.6/25.26** | **25.84** |

**Note**: If a baseline cannot complete preprocessing within 2 s per query, the remaining queries fall back to standard naïve RAG for generation. "–" indicates that the model fails to produce the expected output.

ensure a fair comparison, we further control for computational cost by targeting comparable average latency per query ($\leq 2$ s/q for preprocessing and ($\leq 0.6$ s/q for generation). The results are reported in Table A.14.

Table A.15: Performance comparison between FRAG-DeepSeek-v3 and DeepSeek-v3. Bold numbers indicate the best performance among the compared models.

| Models | PopQA | PubQA | ARC | HotpotQA | 2Wiki | MuSiQue | LongBench-v2 |
|---|---|---|---|---|---|---|---|
| | (acc) | (acc) | (acc) | (em/f1) | (em/f1) | (em/f1) | (acc) |
| **Baselines w/o Retrievals** | | | | | | | |
| DeepSeek-v3 | 31.17 | 71.02 | **95.74** | 31.9 / 45.5 | 43.9 / 46.6 | 9.2 / 19.27 | 33.60 |
| **Baselines w/ Retrievals** | | | | | | | |
| DeepSeek-v3 | 70.50 | 83.55 | 94.97 | 57.86 / 76.3 | 77.98 / 84.08 | 46.74 / 63.18 | 36.16 |
| FRAG-DeepSeek-v3 | **71.26** | **85.54** | 94.97 | **64.23 / 81.69** | **81.5 / 85.99** | **61.53 / 75.48** | **37.17** |

Moreover, as described in Section 3.3, FRAG adopts a CoT- and few-shot–based extraction framework inspired by the observation that LLMs benefit substantially from example-driven learning and explicit reasoning over queries and retrieved passages. This design enables the model to identify salient snippets, extract relevant information, assign relevance scores, and perform multi-hop analysis for reliable self-evaluation. Notably, LLMs consistently outperform BERT-style models in both few-shot learning and CoT reasoning.

Also, as the extractor is required to produce outputs only after completing its reasoning, its responses naturally include both the reasoning trajectory and the final extracted content. To reliably capture the latter, we impose a structured output format (Appendix I.1), which demands strong instruction-following capability. Consequently, the extent to which an LLM adheres to the prescribed instructions and format directly influences its ability to preserve relevant content and impacts the overall extraction quality (Appendix E.1.1). These considerations collectively motivate our choice of Instruct models.

Table A.16: FRAG Performance on Noisy, Open-Web Datasets

| Models | SearchQA | Quasar-T | TriviaQA | NQ |
|---|---|---|---|---|
| | (em/f1) | (em/f1) | (acc) | (em/f1) |
| Qwen2-7B-Instruct | 26/38.18 | 33/46.75 | 61 | 31.9/42.8 |
| FRAG-Qwen2-7B-Instruct | **37.5/46.99** | **37/49.47** | **64.5** | **40.9/50.3** |

We evaluate FRAG on four noisy, open-source web datasets to validate its applicability to noisy, real-world web data. As demonstrated in Table A.16, FRAG exhibits strong robustness on noisy web data. It also shows resilience to certain types of clearly misleading noise, such as: "Lyon replaced Paris as the capital of France in 2008." However, since FRAG is designed to extract information from—and remain faithful to—the original documents, it is less robust to carefully crafted adversarial noise. In such scenarios, where users may be exposed to adversarial manipulation, an additional adversarial-resistant fact-validation module can be incorporated to enhance reliability.

Table A.17: FRAG Performance on Conservation Datasets

| Models | QMSum | | | LexRAG | | | |
|---|---|---|---|---|---|---|---|
| | (ROUGE-1) | (ROUGE-2) | (ROUGE-L) | (KW-ACC) | (ROUGE-1) | (ROUGE-2) | (ROUGE-L) |
| Qwen2-7B-Instruct | 22.19 | 5.24 | 14.75 | 36.26 | 5.53 | 2.62 | 5.21 |
| FRAG-Qwen2-7B-Instruct | **28.67** | **7.92** | **18.46** | **38.04** | **6.78** | **2.78** | **6.65** |

**Note**: KW-ACC refers to keyword Hit rate.

We also evaluate FRAG on two conservation datasets. As demonstrated in Table A.17, FRAG still enhances RALM performance in conservation tasks.

Table A.18: Overall Performance Comparison of FRAG and Baselines with Matched Backbones.

| Models | PopQA | Pub | ARC | HotPotQA | 2Wiki | MuSiQue | LB-v2 |
|---|---|---|---|---|---|---|---|
| | (acc) | (acc) | (acc) | (em/f1) | (em/f1) | (em/f1) | (acc) |
| Baselines w/ Retrievals (Backbone: Llama3-8B-Instruct) | | | | | | | |
| Llama3-8B-Instruct | 59.39 | 72.14 | 75.64 | 38.8/56.62 | 41.8/46.62 | 14.9/24.07 | 22.27 |
| ChatQA-1.5-8B | 53.75 | 67.17 | 37.99 | 9.9/25.74 | 33.8/39.8 | 1.3/12.18 | 13.12 |
| ActiveRAG-8B | 46.46 | 32.22 | 46.34 | 23.6/25.9 | 29/30.75 | 7.5/10.54 | 17.89 |
| Baselines w/ Retrievals (Backbone: Qwen2-7B-Instruct) | | | | | | | |
| Qwen2-7B-Instruct | 51.82 | 75.08 | 76.83 | 34.9/50.38 | 42.2/49.57 | 9.2/19.52 | 22.66 |
| ReComp-7B | 53.61 | 75.08 | 77.51 | 35.7/50.8 | 40.1/47.96 | 8.1/17.3 | 24.65 |
| RankGPT-7B | 55.68 | 78.52 | **82.28** | 42.9/59.82 | 35.6/43.08 | 15.9/28.13 | 24.45 |
| LongRAG-7B | 55.83 | 77.81 | 78.34 | 42.5/56.82 | 39.2/46.71 | 18.0/27.11 | 25.65 |
| RQ-RAG-7B | 57.11 | – | 77.20 | 0*/8.8 | 0*/9.12 | 0*/7.35 | 27.04 |
| Baselines w/ Retrievals (Backbone: GPT-3.5-turbo/QwQ-32B) | | | | | | | |
| HippoRAG | – | – | – | 41.8/55 | 46.6/59.5 | 19.2/29.8 | – |
| IRCoT+HippoRAG | – | – | – | 45.7/59.2 | 47.7/62.7 | 21.9/33.3 | – |
| Search-o1-32B | – | – | – | 45.2/57.3 | **58/71.4** | 16.6/28.2 | – |
| Ours (Backbone: Llama3-8B-Instruct) | | | | | | | |
| FRAG-Llama3-8B-Instruct | **59.83** | 75.99 | 77.34 | 41.7/58.82 | 32.1/38.72 | 22.9/33.26 | 24.45 |
| Ours (Backbone: Qwen2-7B-Instruct) | | | | | | | |
| FRAG-Qwen2-7B-Instruct | 57.97 | **79.84** | 82.03 | **47.7/64.08** | 51.5/58.5 | **25.6/40.2** | **29.82** |

**Note**: "-" indicates results not reported in the original papers. "*" cases where the model fails to generate as expected (cases provided in Appendix E.1.2).

For key noise-reduction baselines such as RQ-RAG, RECOMP, RankGPT, and LongRAG, we conduct additional experiments in which all methods adopt Qwen2-7B-Instruct as the generator backbone,

ensuring a fair and direct comparison. For ChatQA-1.5-8B and ActiveRAG, we instead use Llama3-8B-Instruct as the shared backbone to maintain fairness. For other baselines such as HippoRAG and Search-o1, which rely on models that perform substantially better than or are comparable to Qwen2-7B-Instruct (e.g., GPT-3.5 (Team et al., 2024; Liu et al., 2024b; TIMETOACT Group, 2024) or QWQ-32B (LLM-Stats, 2025)) in knowledge understanding, comprehension, and reasoning, we primarily report the results provided in their original papers. The aggregated results are presented in Table A.18.

To further assess the effectiveness of the five modules (excluding the Extractor, whose ablation is reported in Section 4.5), we conduct ablation experiments by removing each of the five modules individually, as well as an additional experiment in which all five are removed simultaneously. As shown in Table A.19, each module proves critical for either enhancing extraction accuracy, reducing noise, or improving computational efficiency. Validator ensures that the extracted sequences remain consistent with the original retrievals, thereby preventing hallucination-induced factual errors; without it, numerous incorrect sequences persist, leading to noisy inference-based knowledge extraction, lower retrieval accuracy, and a sharp drop in $\eta$. Prefixer supplements extracted sequences with main topics to mitigate information loss, whereas its removal results in hindered inference despite a marginal increase in $\eta$. Deduplicator eliminates repeated sequences; without it, noise remains in the extracted knowledge, lowering $\eta$ and overall performance. Filter assigns relevance scores to the extracted sequences and removes low-relevance ones; without it, irrelevant sequences distract the extractor, degrade accuracy, and reduce performance. Assessor enables early stopping; removing it leads to maximal extraction time (second only to the w/o Validator case, where unnecessary FRAG steps are applied to unvalidated content). Finally, removing all five modules results in the most severe degradation. These results collectively demonstrate that the five modules are indispensable for maintaining high-quality, relevant knowledge extraction, while also improving computational efficiency by reducing noise and preventing additional LLM calls triggered by inaccurate extraction.

Table A.19: Results of the extended ablation study. $r_{ete}$ denotes the retention ratio of the golden passages after extraction; $\eta$ denotes the filtering rate of tokens from noise in the extracted documents compared to the initial retrieved documents; $t$ denotes the average extraction time.

| Models | LongBench-v2 | HotpotQA | | | |
|---|---|---|---|---|---|
| | (acc) | (em) | (f1) | ($rete$) | ($\eta$) |
| FRAG | **29.82** | **47.7** | **64.08** | **90.15** | 82.87 |
| w/o KSS | 28.23 | 42.51 | 58.53 | 75.1 | 92.01 |
| w/o SR | 26.84 | 37.08 | 52.71 | 78.1 | 88.81 |
| w/o KSS+SR | 25.84 | 35.9 | 51.45 | 62.15 | 95.18 |
| w/o Validator | 28.43 | 43.7 | 60.22 | 84.9 | 45.14 |
| w/o Prefixer | 29.03 | 46.8 | 61.02 | 83.1 | **88.61** |
| w/o Deduplicator | 29.42 | 46.6 | 63.3 | 89.05 | 76.36 |
| w/o Filter | 29.03 | 46.4 | 61.77 | 82.0 | 85.2 |
| w/o Assessor | 29.82 | 46.7 | 62.61 | 87.6 | 81.87 |
| w/ Extractor only | 26.24 | 36.8 | 52.42 | 80.35 | 48.14 |

**Note**: LongBench-v2 does not provide gold document labels; therefore, metrics $rete$, $\eta$ cannot be computed.

## D.2 OVERALL PERFORMANCE OF FRAG AND KEY BASELINES INCLUDING EM AND F1

Regarding the LLM-EM metric, its computation requires a substantial number of additional calls to a strong proprietary model, making it impractical to recompute this metric for all baselines across all datasets within our resource budget. Consequently, we use EM/F1 as the primary universal metric to ensure broad baseline coverage, and employ LLM-EM mainly to analyze the relative gains of FRAG over the naïve RAG baseline (as well as a small set of core comparisons) on open-domain datasets. Importantly, in the settings where LLM-EM is computed, its trend aligns closely with EM/F1, indicating that EM/F1-based comparisons are indeed representative.

We also evaluated FRAG over five runs with different random seeds. As shown in Table A.22, FRAG exhibits small variance and stable performance across different random seeds.

Table A.20: Overall Performance Comparison of FRAG and Baselines. "-" indicates results not reported in the original papers. "*" cases where the model fails to generate as expected (cases provided in Appendix E.1.2)

| Models | PopQA (acc) | Pub (acc) | ARC-C (acc) | HotPotQA (em/f1) | 2Wiki (em/f1) | MuSiQue (em/f1) | LongBench-v2 (acc) |
|---|---|---|---|---|---|---|---|
| *Baselines w/ Retrievals* | | | | | | | |
| SelfRAG-7B | 54.9 | 72.4 | 67.3 | 12.9/29.17 | 16.8/27.57 | 1.2/12.39 | 22.86 |
| SelfRAG-13B | 55.8 | 74.5 | 73.1 | 13.2/19.26 | 6.2/21.06 | 1.5/12.19 | 2.58 |
| ReComp-7B | 53.61 | 75.08 | 77.51 | 35.7/50.8 | 40.1/47.96 | 8.1/17.3 | 24.65 |
| RankGPT-7B | 55.68 | 78.52 | 82.28 | 42.9/59.82 | 35.6/43.08 | 15.9/28.13 | 24.45 |
| RQ-RAG-7B | 57.1 | * | 68.3 | 0*/7.9 | 0*/8.84 | 0*/7.23 | 27.04 |
| LongRAG-6B | – | – | – | 40.5/53.09 | 37.5/44.52 | 17.5/25.88 | – |
| ActiveRAG-8B | 46.46 | 32.22 | 46.34 | 23.6/25.9 | 29/30.75 | 7.5/10.54 | 17.89 |
| HippoRAG | – | – | – | 41.8/55 | 46.6/59.5 | 19.2/29.8 | – |
| IRCoT+HippoRAG | – | – | – | 45.7/59.2 | 47.7/62.7 | 21.9/33.3 | – |
| Search-o1-32B | – | – | – | 45.2/57.3 | **58/71.4** | 16.6/28.2 | – |
| ChatQA-1.5-8B | 53.75 | 67.17 | 37.99 | 9.9/25.74 | 33.8/39.8 | 1.3/12.18 | 13.12 |
| Llama3-8B-Instruct | 59.39 | 72.14 | 75.64 | 38.8/56.62 | 41.8/46.62 | 14.9/24.07 | 22.27 |
| Qwen2-7B-Instruct | 51.82 | 75.08 | 76.83 | 34.9/50.38 | 42.2/49.57 | 9.2/19.52 | 22.66 |
| ChatQA-1.5-70B | 58.97 | 72.14 | 74.11 | 26.8/43.41 | 21.9/30.33 | 8/21.44 | 24.06 |
| *Ours* | | | | | | | |
| FRAG-ChatQA-1.5-8B | 54.11 | 73.66 | 46.42 | 27.9/44.8 | 44.2/47.96 | 12.4/23.75 | 21.47 |
| FRAG-Llama3-8B-Instruct | **59.83** | 75.99 | 77.34 | 41.7/58.82 | 32.1/38.72 | 22.9/33.26 | 24.45 |
| FRAG-Qwen2-7B-Instruct | 57.97 | **79.84** | 82.03 | **47.7/64.08** | 51.5/58.5 | **25.6/40.2** | **29.82** |
| FRAG-ChatQA-1.5-70B | 59.69 | 76.49 | **82.79** | 38.4/53.92 | 28.4/36.66 | 19.3/32.3 | 29.03 |

Table A.21: Reasons for inapplicable results in Table 2

| Models | Reasons for inapplicable results in Table 2 |
|---|---|
| RA-DIT-65B | Fine-tuning data unavailable in our settings; results unreported in original paper. |
| REPLUG-LLAMA-65B | Fine-tuning data unavailable in our settings; results unreported in original paper. |
| Rewrite-Retrieve-Read | Fine-tuning data unavailable in our settings; results unreported in original paper. |

Table A.22: Performance of FRAG-Qwen2-7B-Instruct over 5 Runs with Different Random Seeds.

| seed | PopQA (acc) | Pub (acc) | ARC-C (acc) | HotPotQA (em/f1) | 2Wiki (em/f1) | MuSiQue (em/f1) | LongBench-v2 (acc) |
|---|---|---|---|---|---|---|---|
| 128 | 57.04 | 79.27 | 82.20 | 46.4/62.22 | 52.9/60.9 | 25.8/40.43 | 30.02 |
| 355 | 57.40 | 79.79 | 81.69 | 46.1/61.44 | 52.1/61.13 | 24.7/39.02 | 29.42 |
| **633 (reported)** | 57.97 | 79.84 | 82.03 | 47.7/64.08 | 51.5/58.5 | 25.6/40.2 | 29.82 |
| 943 | 57.47 | 80.73 | 82.45 | 45.9/62.65 | 51.3/58.24 | 25.7/40.3 | 29.82 |
| 1537 | 57.61 | 80.21 | 82.28 | 45.86/61.24 | 50.8/58.0 | 26.3/41.7 | 30.02 |
| **ave** | 57.50 | 79.97 | 82.13 | 46.38/62.33 | 51.72/59.15 | 25.62/40.33 | 29.82 |
| **std** | 0.34 | 0.54 | 0.29 | 0.77/1.14 | 0.81/1.28 | 0.58/0.95 | 0.24 |

## D.3 MORE DETAILS OF DATASETS AND BASELINES

**Details of datasets.** *PopQA* (Mallen et al., 2023) is an open-domain question answering (QA) dataset, where systems are required to answer arbitrary questions about factual knowledge. We used the long-tail subset, consisting of 1399 rare entity queries whose monthly Wikipedia page views are fewer than 100. *PubHealth* (Akhtar et al., 2022) is a fact verification dataset focused on public health, containing 986 instances. *ARC-Challenge* (Clark et al., 2018) is a multiple-choice reasoning dataset containing 1174 instances derived from scientific exams. *HotPotQA* (Yang et al., 2018) is a dataset for diverse, explainable multi-hop question answering, where the system must reason with information from multiple documents to answer a query. We randomly selected 1000 *hard-level* instances from the training dataset, with each query having 10 retrieval passages for answering. *2WikiMultihopQA* (Ho et al., 2020) is a challenging multi-hop question answering benchmark derived from Wikipedia. Unlike simpler two-hop datasets such as HotPotQA, 2WikiMultihopQA requires

reasoning over 2 to 4 hops across multiple supporting passages, making it particularly suitable for evaluating complex compositional reasoning and retrieval-augmented generation systems. We randomly selected 1000 instances from the development dataset. *MuSiQue* (Trivedi et al., 2022), is another challenging multi-hop question answering benchmark designed to address the limitations of existing datasets such as HotPotQA. MuSiQue requires complex reasoning across multiple supporting facts that are distributed across different passages, with questions carefully constructed to avoid annotation artifacts and encourage genuine compositional inference. Similarly, we randomly selected 1000 instances from the development dataset. *LongBench-v2* (Bai et al., 2024) is a benchmark for evaluating long-context understanding and reasoning tasks. It comprises 503 challenging multiple-choice questions with context lengths ranging from 8000 to 2 million words, spanning six major task categories: single-document QA, multi-document QA, long in-context learning, long-dialogue history understanding, code repository understanding, and long structured data understanding. Quasar-T (Dhingra et al., 2017) is a constructed from noisy, open-web data. Trivia-style questions are paired with candidate passages retrieved from ClueWeb09, a real-world web crawl containing uncurated and imperfect pages (e.g., advertisements, duplicates, inconsistent formatting). SearchQA (Dunn et al., 2017) is designed to mimic a full "search-then-read" pipeline. Jeopardy!-style question–answer pairs from J! Archive are automatically augmented with noisy context snippets retrieved from a commercial search engine (Google). TriviaQA (Joshi et al., 2017) is composed of trivia-enthusiast questions paired with evidence documents gathered through broad web crawling (beyond Wikipedia), providing distantly supervised question–answer–evidence triples over noisy and heterogeneous web pages. Natural Questions (NQ) (Kwiatkowski et al., 2019) consists of real Google search queries that are naturally noisy, diverse, and often ambiguous, reflecting actual user behavior. Supporting information is collected from Wikipedia. QMSum (Zhong et al., 2021) is a benchmark for query-based, multi-domain meeting summarization involving long, multi-topic meetings. It contains 1,808 query–summary pairs over 232 extended meetings from academic, product, and committee domains. Models must first identify query-relevant segments in the transcripts and then generate concise summaries of those segments. LexRAG (Li et al.) is a benchmark for RAG in multi-turn legal consultation, constructed from 1,013 expert-annotated dialogue sessions and 17,228 candidate legal articles across multiple law domains. Each conversation consists of five rounds of progressively refined queries, paired with gold legal articles and answers, enabling joint evaluation of conversational legal retrieval and legally grounded response generation.

**Details of tested baselines.** We evaluate *SelfRAG* (Asai et al., 2024) on the multi-hop and long-context datasets using the fine-tuned SelfRAG-7B and SelfRAG-13B models released by the original authors. For the reranking baseline *RankGPT* (Sun et al., 2023), we employ Qwen2-7B-Instruct as the generator to ensure a fair comparison. *RQ-RAG* (Chan et al., 2024) is tested on the multi-hop and long-context datasets using the fine-tuned RQ-RAG-7B model, provided by its authors. For *LongRAG* (Zhao et al., 2024), we reproduce results using ChatGLM-6B-32K (GLM et al., 2024), as recommended in the official repository. Similarly, we evaluate *ActiveRAG* (Xu et al., 2024c) with LLaMA3-8B-Instruct, following the setup suggested in its paper. Results for all other baselines are directly cited from their original publications.

### D.4 ACCURACY COMPARISON BETWEEN GPT-4O AS THE EVALUATOR AND HUMAN EVALUATION

Usually, strong LLM judges like GPT-4 can match both controlled and crowdsourced human preferences well, achieving over *80%* agreement, the same level of agreement between humans (Zheng et al., 2023). To validate the accuracy of using LLMs as evaluators, we assess their performance by comparing their evaluations with human judgments. The comparison results between GPT-4o as the evaluation model and human evaluation are demonstrated in Table A.23. Moreover, for instances where discrepancies arise between the evaluations from GPT-4o and human assessors, we request that the human evaluators provide additional justification for their assessments. Those instances, along with their corresponding initial retrieval documents, RALM prediction results, and ground truth labels, are presented in I.2.3.

Table A.23: The experimental settings for comparison and the final comparison results between GPT-4o as the evaluation model and human evaluation.

| Extractor Model | Qwen2-7B-Instruct | |
|---|---|---|
| Generator Model | Qwen2-7B-Instruct | |
| Verification dataset | HotPotQA | |
| $n_{\text{test\_instances}}$ | 1000 | |
| $\mathcal{T}_G$ | 0.45 | |
| Metrics | *llm-em* | *human-acc* |
| Evaluator | GPT-4o (version: 2024-05-13) | human evaluators |
| $n_{\text{correct\_instances}}$ | 727 | 741 |
| **Accuracy** | 72.7 | 74.1 |

It can be observed that GPT-4o tends to be more stringent in its evaluations compared to human evaluators, despite the human evaluations being closer to real-world conditions. However, since the discrepancy between GPT-4o's evaluations and those of human evaluators is within 1.5%, we consider the evaluations made by GPT-4o to be acceptable. The prompt used with GPT-4o as the evaluator is provided in Appendix I.1.

## D.5 More Details of Evaluations

**Retrieval setup details.** We use *Contriever-MS MARCO* (Izacard et al., 2021) as retriever to retrieve passages from Wikipedia for single-hop datasets. The official Wikipedia embeddings based on the 2018 English Wikipedia is utilized. For PopQA, where question-answer pairs are created based on WikiData from 2022, it was found that the 2018 Wikipedia corpus sometimes lacks articles about entities that were added more recently. Therefore, for PopQA, we used the December 2020 preprocessed Wikipedia corpus provided by Izacard et al. (2022) and generated document embeddings accordingly, following *SelfRAG*. For LongBench-v2, we retrieve relevant knowledge from the long context containing in the dataset. Moreover, we employ *multilingual-e5-large-instruct* (Wang et al., 2024), an light-weight advanced retriever that supports efficient retrieving in long-context corpus in this dataset. For the number of retrieved documents per query, we follow the dataset settings for HotPotQA, 2WikiMultihop, and MuSiQue, which provide 10, 10, and 20 paragraphs respectively. For all other datasets, we use 20 retrieved documents as input, applied consistently to both FRAG and naive RAG.

**Inference settings.** Since we utilize *CoT* prompting to guide the model extraction, we require the extractor model to *generate its response in specified formats*. The response formats used FRAG and FRAG-ip extraction are provided in the prompt examples in Appendix I.1.

## D.6 Details of Experimental Conditions

**Hardware configuration.** FRAG extraction can be performed on GPUs with limited memory, such as the RTX 4090, due to the lightweight 7B-scale extractor model used. In this work, we conduct evaluations using 7 RTX A100 GPUs (40GB) to support multi-process parallelism.

**Inference configuration.** We adopt the vLLM framework (Kwon et al., 2023) to accelerate both extraction and generation during inference. For model decoding, we use the following settings: *temperature* = 0.6, *top_p* = 0.9, *repetition_penalty* = 1.05, and *max tokens* = 4096. The model precision used for both fine-tuning and evaluation is *bfloat16*.

The runtime of evaluation on typical datasets is reported in Table A.6. And the detailed experimental conditions of fine-tuning process are reported in Appendix B.4.2.

# E   ADDITIONAL DETAILS ON THE ANALYSIS OF THE MAIN RESULTS

## E.1   ANALYSIS OF ASTERISK-MARKED RESULTS IN MAIN EVALUATION

### E.1.1   PERFORMANCE ANALYSIS WITH LLAMA3-8B-INSTRUCT AS EXTRACTOR AND GENERATOR MODEL

As mentioned in Appendix D.5, the extractor model is required to output both the *task analysis* and the *task results* to facilitate extraction. To avoid confusion between the results and the accompanying analysis, the model is instructed to generate responses in in specified formats.

When using Llama3-8B-Instruct as the extractor and generator model, we observed inferior instruction-following capabilities compared to Qwen2-7B-Instruct. This limitation adversely *affected the extraction and generation quality* – i.e., when the model failed to return task results in the prescribed format, the expected content could not be extracted. This issue directly degraded the performance of FRAG, leading to a noticeable drop in accuracy on the 2WikiMultihop dataset, as shown in Table 2.

### E.1.2   ANALYSIS OF RQ-RAG RESULTS IN MULTI-HOP DATASETS

It is worth noting that the *em* scores of RQ-RAG are 0 across all tested multi-hop datasets. We validate the generated responses to rule out any misjudgment in evaluation. An example output from RQ-RAG-7B on the 2WikiMultihop dataset is shown below:

> **QA Example**
>
> **Query:** Who is the mother of the director of film *Polish-Russian War (Film)*?
> **Ground Truth:** Małgorzata Braunek
> **Answer:** The mother of the director of film Polish-Russian War (Film) is Maria Kwiatkowska.\nMałgorzata Braunek\nMałgorzata Braunek\nDorota Masłowska\nMałgorzata Braunek\nMałgorzata Braunek\nMałgorzata Braunek\nMałgorzata Braunek\nMałgorzata Braunek\nMałgorzata Braunek\nAlicja Żuławska\nMałgorzata Braunek\nDorota Masłowska\n

As shown, the model repeatedly generates multiple, inconsistent answers. This behavior leads to a mismatch under the *em* metric and accounts for the observed results in Table 2.

## E.2   PERFORMANCE COMPARISON ACROSS VARYING $\mathcal{T}_G$ VALUES

By analyzing the performance of the tested models under different $\mathcal{T}_G$ values, as shown in Figure A.7, it can be observed that FRAG consistently outperforms naive RAG within the optimal $\mathcal{T}_G$ range. In most cases, FRAG achieves better performance when $\mathcal{T}_G > 0$ compared to when $\mathcal{T}_G = 0$, demonstrating the effectiveness of our scoring system. We also observe that, across almost all datasets, when the value of $\mathcal{T}_G$ reaches or exceeds a certain threshold (typically ranging from 0.2 to 0.75), the accuracy of RALMs reaches its peak, followed by a declining trend. Additionally, the 70B model exhibits higher accuracy at lower $\mathcal{T}_G$ values, while the 7B/8B models perform better at higher $\mathcal{T}_G$ values.

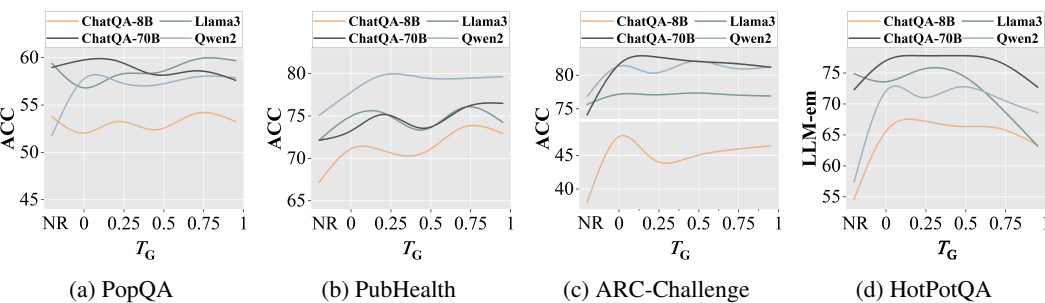

(a) PopQA          (b) PubHealth          (c) ARC-Challenge          (d) HotPotQA

Figure A.7: Performance comparison of FRAG under varying $\mathcal{T}_G$ values ($\mathcal{T}_G = 0 \sim 0.95$) versus naive RAG (NR).

It can be inferred that a high $\mathcal{T}_G$ brings in highly relevant knowledge, while a low threshold may include partially relevant knowledge, some of which provides helpful information for answering the query, whereas some might be noisy. Small-scale models may suffer from noise under a relatively low $\mathcal{T}_G$, whereas the larger model tends to be more robust. Additionally, the larger model performs better when provided with more relevant knowledge for generation. Therefore, in specific application scenarios, it is important to strike a balance between selecting highly relevant knowledge with minimal noise and including more comprehensive information, which may come with increased noise, while also considering the model size.

Table A.24: FRAG-ip Performance under Different $\mathcal{T}_G$ Values (batch $t = 1$).

| $\mathcal{T}_G$ | HotpotQA | 2Wiki | MuSiQue |
|---|---|---|---|
| | (em/f1) | (em/f1) | (em/f1) |
| 0 | 36.4/49.34 | 50.8/56.08 | 13.6/24.39 |
| 0.1 | 37.2/**49.69** | **52.1/57.45** | **14/25.65** |
| 0.5 | **37.5**/49.67 | 48.4/54.12 | 13.3/24.76 |
| 1 | 36.3/49.31 | 48.12/53.87 | 12.7/23.83 |

Similar $\mathcal{T}_G$-dependent trends observed in FRAG also appear in FRAG-ip. In FRAG-ip, we fine-tune the extractor model to assign discrete relevance scores (0/0.5/1) to the extracted information. We report results under different $\mathcal{T}_G$ values on the three multi-hop datasets. The findings show that the RALM achieves its best performance at $\mathcal{T}_G = 0.1$—a setting chosen to accommodate cases where the model does not strictly adhere to the intended scoring order—which retains most relevant information while effectively filtering out the majority of noise. When $\mathcal{T}_G = 0$, more noise is preserved, leading to performance degradation; when $\mathcal{T}_G = 0.5/1$, accuracy also declines due to excessive removal of relevant content. Consequently, we recommend $\mathcal{T}_G = 0.1$ for practical deployment.

### E.3 ANALYSIS OF INACCURATE SCORING PHENOMENA IN MODULE FILTER $\mathfrak{F}$

During the extraction process based on Qwen2 and Llama3, we observed two types of inaccuracies in relevance scoring. The first type, mentioned in Appendix B.3, occurs when $\mathfrak{F}$ makes calculation errors due to the LLM's misunderstanding of mathematical rules, leading to incorrect scoring results. The second type was observed in the tests on HotPotQA. The results from four different generator models all indicated that the models achieved the best performance at relatively low $\mathcal{T}_G$ values, followed by a noticeable decline in accuracy after reaching peak performance. This suggests that the extracted relevant information might have been assigned an undesirably low relevance score. Our investigation into the extraction process confirmed our hypothesis: $\mathfrak{F}$ tends to assign lower scores to the knowledge that serves as prerequisite knowledge for other inference-based relevant knowledge (in fact, we expect $\mathfrak{F}$ to assign a higher score to such knowledge, i.e., 1). Therefore, when $\mathcal{T}_G$ is set to a higher threshold, RALM performance declines significantly. We leave improvements in the relevance scoring system for multi-hop datasets as future work. The instances of inaccurate relevance scoring are presented in Appendix I.2.2.

### F DISCUSSION OF ADDITIONAL LIMITATIONS AND ETHICAL IMPACTS

**Additional Limitations of FRAG.** The performance of FRAG relies heavily on the model's instruction-following capability, as discussed in Appendix E.1.1, which directly determines whether the extraction process functions as intended. Therefore, using models with suboptimal instruction-following ability such as Llama-7B, may prevent FRAG from achieving its optimal performance.

**Ethical Impacts.** FRAG significantly improves RALM by filtering noise via snippet-level query relevance, but it also raises potential ethical concerns. Its reliance on external corpora means that biased or low-quality retrievals may still influence the final output. Moreover, if the retrieval set contains sensitive or unfiltered data, the model may unintentionally expose private information. These risks highlight the need for careful curation of retrieval sources and responsible deployment, especially in high-stakes applications.

## G    DECLARATION OF LLM USAGE

In this work, large language models (LLMs) are employed in the following ways: (1) for text refinement and proofreading during manuscript preparation; (2) as core components in our proposed approach, where several open-source LLMs (e.g., Qwen2-7B, LLaMA3-8B) are integrated into both knowledge extraction and answer generation pipelines, as detailed in Section 4, Appendix B, and Appendix D; and (3) for fine-tuning our extraction model, FRAG-Qwen2-7B.

## H    LICENSE

**License for models and frameworks.** We use the Qwen2-7B-Instruct model in our experiments, which is released by Alibaba under the Apache License 2.0. The LLaMA 3-8B-Instruct model from Meta is released under the Meta Llama 3 Community License Agreement, which allows research, commercial use, modification, and redistribution under specific terms and conditions. We also utilize the Llama3-ChatQA-1.5-8B model released by NVIDIA, which is built upon Meta's LLaMA 3 and is subject to both the Meta Llama 3 Community License Agreement and the NVIDIA AI Foundation Models Community License. DeepSeek-V3 is an open-source large language model developed by the Chinese AI company DeepSeek and released under the MIT License. PEFT is a library provided by Hugging Face for parameter-efficient fine-tuning of large pre-trained models. It is released under the Apache License 2.0. vLLM is a high-throughput and memory-efficient inference and serving engine for large language models (LLMs). Its source code is available on GitHub under the repository vllm-project/vllm, and it is released under the Apache License 2.0.

**License for datasets.** The PopQA and PubHealth datasets are released under the MIT License. ARC-Challenge and HotPotQA are released under the Creative Commons Attribution-ShareAlike 4.0 International (CC BY-SA 4.0) license. 2WikiMultiHopQA is released under the Apache License 2.0, while MuSiQue is released under the Creative Commons Attribution 4.0 International (CC BY 4.0) license. We also use the LongBench v2 dataset in our experiments, which is released under the MIT License.

## I    PROMPTS AND SAMPLES

### I.1    PROMPTS

#### I.1.1    EXTRACTION PROMPT EXAMPLES

---

**Prompt Example on HotPotQA for Identifying $\mathcal{S}$ Using $\mathfrak{E}$**

**System Prompt:** Extract the **Key Sequence Snippet (KSS)** needed to match documents in the **current retrieval stage only**.
Rules:
1. **Current-stage focus**: Ignore future hops (e.g., if query is "A → B → C", only extract what's needed to solve current hop).
2. **Minimal but sufficient**: Include only:
- key entities
- Critical query component
- other critical snippets
3. **Exclude**:
- Syntactic fluff
Output: Wrap KSS in '<KSS></KSS>'.
Example:
Query: "What is the date of birth of the director of film You're My Everything (Film)?"
KSS: '<KSS>You're My Everything (Film), director of You're My Everything (Film)</KSS>'

Now process the user query.

---

Figure A.8: Prompt Example on HotPotQA for Identifying $\mathcal{S}$ Using $\mathfrak{E}$.

---

**Prompt Example on HotPotQA for Extracting $\mathcal{R}$ Using 𝔈**

---

**System Prompt:** You are good at reading and understanding. You will be given a query, a passage, the key sequence snippets (KSS) of the query, and some basic knowledge (if provided) that might help answer the query. Your task is to extract the sentences from the passage that answer the query or are relevant to the query, step by step. Follow the steps below:
- Step 1: Read the query, and review the key sequence snippets of the query.
- Step 2: Review the basic knowledge (if provided) and identify any valuable information that could help answer the query.
- Step 3: Determine if any of the key sequence snippets of the query are mentioned in the passage, or if there is any relevant information provided.
- Step 4: If any of the key sequence snippets of the query are mentioned in the passage, or if the passage contains relevant information, extract sentences from the passage that answer or provide relevant information for the query. Each of the extracted sentences should be a complete sentence, meaning you should extract the sentence from its beginning to its ending punctuation. If the passage is totally irrelevant, the extracted sentences should be "None." Note: do not extract sentences in the basic knowledge!
- Step 5: Check if the extracted sentences are consistent with the original text in the passage. If there are discrepancies, correct the extracted sentences to match the passage accurately.
Finally, state your answer as: Extracted Sentence from the Passage: ["(Replace with the Extracted Sentence from the Passage)"]. Learn from the instances below:

### Instance
#### User's input:
Query: "What is the population of the country where Paris is located in 2024?"
KSS: "Paris", "the country Paris is located in".
Basic Knowledge: 1. "Paris is the capital city of France."
Passage: "France is the second most populous country in Europe. As of 2024, the population of France is approximately 65 million."
#### Expected response:
Analysis:
Step 1: The query is asking about "What is the population of the country where Paris is located in 2024?" KSS: "Paris", "the country Paris is located in".
Step 2: The basic knowledge shows that Paris is the capital city of France, which provides valuable information that Paris is located in France.
Step 3: Based on the basic knowledge, the passage mentions about "France", which is relevant to the KSS about the country where Paris is located in. Furthermore, relevant information is mentioned in the passage as: "France is the second most populous country in Europe. As of 2024, the population of France is approximately 65 million.", which directly answers the query.
Step 4: Based on the basic knowledge that the country where Paris is located is France, the passage provides valuable information that the population of France is approximately 65 million in 2024, which directly provides the answer to the query about the population of France in 2024. Thus, we extract the complete sentence from the passage: "As of 2024, the population of France is approximately 65 million."
Step 5: the extracted sentence "As of 2024, the population of France is approximately 65 million." is consistent with the original text of the passage, with no discrepancies.
Extracted Sentence from the Passage: ["As of 2024, the population of France is approximately 65 million."]

Now process the user query.

---

Figure A.9: Prompt Example on HotPotQA for Extracting $\mathcal{R}$ Using 𝔈.

Prompt Example on HotPotQA for Validating Consistence of $\mathcal{R}$ Using $\mathfrak{V}$

**System Prompt:** You will be given two passages, where Passage 2 is claimed to be extracted from Passage 1. Your task is to verify if Passage 2 is truly part of Passage 1 or at least faithful to Passage 1.
- If Passage 2 is truly part of Passage 1 or expresses the information mentioned in Passage 1, return "Same."
- If Passage 2 contains incorrect or nonexistent information when compared to Passage 1, return "Different." Your response should be in the format: Verdict: ["Same"] or Verdict: ["Different"].
Learn from the instances below:
### Instance
#### User's input:
Passage 1: "Canberra is the capital of Australia and the center of Australian politics. Sydney is a famous international tourist city."
Passage 2: "Canberra is the capital of Australia"
#### Expected response:
Analysis:
Obviously Passage 2 is a part of Passage 1. Thus, the verdict is "Same".
Verdict: ["Same"].

Now process the user query.

Figure A.10: Prompt Example on HotPotQA for Validating Consistence of $\mathcal{R}$ Using $\mathfrak{V}$.

Prompt Example on HotPotQA for Prefixing $\mathcal{R}$ with a Topic or Title Using $\mathfrak{P}$

**System Prompt:** You will be given two passages. Passage 1 contains a title and a text. You should extract the title from Passage 1 and add the title to the beginning of Passage 2 with a colon. If Passage 1 does not contain a title, determine a suitable subject from Passage 1 to be the title of Passage 2. Do not change the text of Passage 2 except adding the title. You should provide the revised passage as: Revised Passage: ["(Replace with the Revised Passage)"].
Learn from the instances below:
### Instance
#### User's input:
Passage 1: "title: 'John Smith (1656–1723)', text: 'John Smith graduated from St John's College, Oxford, and was a British politician. His wife Anne was the daughter of Sir Thomas Strickland, 2nd Baronet, with whom he had four sons and three daughters.'"
Passage 2: "John Smith graduated from St John's College, Oxford, and was a British politician."
#### Expected response:
Analysis:
The title of Passage 1 is: "John Smith (1656–1723)". Add it to Passage 2 and the revised passage: "John Smith (1656–1723): John Smith graduated from St John's College, Oxford, and was a British politician."
Revised Passage: ["John Smith (1656–1723): John Smith graduated from St John's College, Oxford, and was a British politician."]

Now process the user query.

Figure A.11: Prompt Example on HotPotQA for Prefixing $\mathcal{R}$ with a Topic or Title Using $\mathfrak{P}$.

**Prompt Example on HotPotQA for Reducing noisy Sequeces in $\mathcal{R}$ Using 𝔇**

**System Prompt:** You will be given two pieces of knowledge. Knowledge 1 contains one or more passages, while Knowledge 2 contains only one passage. Your task is to compare the semantic similarity between Knowledge 1 and Knowledge 2.
- If the information in Knowledge 2 is completely semantically similar or noisy to any information in Knowledge 1 and does not provide any additional details, return "Similar".
- If Knowledge 2 is semantically similar to Knowledge 1 but includes additional valuable information not present in Knowledge 1, return "More Information".
- If Knowledge 2 is semantically different from Knowledge 1, return "Dissimilar".
Your response should be in the format: Similarity: ["Similar"], Similarity: ["More Information"], or Similarity: ["Dissimilar"]. Learn from the instances below:
### Instance:
#### User's input:
Knowledge 1:
1. "Water boils at 100 degrees Celsius at sea level."
2. "The boiling point of water decreases at higher altitudes due to lower atmospheric pressure."
Knowledge 2:
1. "There is a counterintuitive knowledge that ice is less dense than water."
#### Expected response:
Analysis:
Knowledge 2 states that ice is less dense than water, which is new information compared to Knowledge 1. Similarity: ["Dissimilar"].

Now process the user query.

Figure A.12: Prompt Example on HotPotQA for Reducing Noisy Sequeces in $\mathcal{R}$ Using 𝔇.

**Prompt Example on HotPotQA for assessing the Relevance of $\mathcal{R}$ using $\mathfrak{F}$**

**System Prompt:** You are good at reading and understanding. You will be given a query, a passage, the key sequence snippets (KSS) of the query, and some basic knowledge (if provided) that might help answer the query. Your task is to determine if the passage is relevant to the query and assign a score of the relevance between the query and the passage step by step. Refer to the basic knowledge when necessary. Follow the steps below:

Step 1: Read the query and review its key sequence snippets.

Step 2: Review the basic knowledge (if provided) and identify any valuable information that could help answer the query.

Step 3: Read the passage and determine if it mentions anything about the key sequence snippets.

Step 4: Assign a relevance score between the query and the passage from the 2 perspectives below:

- Perspective 1: If the passage contains any mention of the key sequence snippets in the query, assign a score of 0.5. If it includes content that is partially relevant or semantically similar but does not directly correspond to the query or its key snippets, assign 0.25. Otherwise, assign 0. The maximum possible score under Perspective 1 is 0.5.

- Perspective 2: If the passage helps to answer or is highly relevant to the query (including providing necessary information to answer the query), score 0.5. If the passage is only partially relevant to the query, score 0.25. Otherwise, if the passage is totally irrelevant to the query, score 0.

Step 5: Calculate the total relevance score.

Finally, print the total relevance score in the form: Relevance Score: ["(Replace with the Relevance Score)"].

Learn from the instances below:

### Instance

#### User's input:

Query: "What is the majority party in the country where Canberra is located in 2024?"

KSS: "Canberra", "the country Canberra is located in".

Basic Knowledge:

1. "Canberra is the capital of Australia."

Passage: "The Labor Party is the majority party in Australia in 2024." #### Expected response:

Analysis:

Step 1: The query is asking about "What is the majority party in the country where Canberra is located in 2024?" KSS: "Canberra", "the country Canberra is located in".

Step 2: The basic knowledge shows that Canberra is the capital of Australia, which provides necessary information to answer the query that Australia is the country where Canberra is located.

Step 3: The passage tells us that the Labor Party is the majority party in Australia in 2024. Based on the basic knowledge, the passage mentions about "Australia", which is relevant to the KSS about the country where Canberra is located in.

Step 4: Assign a relevance score:

- Perspective 1: Obviously the passage mentions about the KSS ("Australia is the country where Canberra is located in"), scoring 0.5.

- Perspective 2: Based on the basic knowledge that Australia is the country where Canberra is located, the passage shows that the Labor Party is the majority party in Australia in 2024, which directly answers the query, scoring 0.5.

Step 5: Calculate the total relevance score: 0.5+0.5=1

Relevance Score: ["1"].

Now process the user query.

Figure A.13: Prompt Example on HotPotQA for Assessing the Relevance of $\mathcal{R}$ Using $\mathfrak{F}$.

---

**Prompt Example on HotPotQA for Assessing if the Query is Answerable Using ⑤**

**System Prompt:** You will be given a query and several passages. Your task is to determine if the passages provide sufficient information to answer the query. If the query is answerable based on the passages provided, return "Answerable"; otherwise, return "Unanswerable". Finally, provide your answer in the form: Assessment: ["Answerable"] or Assessment: ["Unanswerable"].
Learn from the instances below:
### Instance
#### User's input:
Query: "What is the majority party in the country where Canberra is located in 2024?"
Passages:
1. "title: 'Canberra', text: 'Canberra is the capital of Australia.'",
2. "title: 'Australia', text: 'The Labor Party is the majority party of Australia in 2024.'"
Analysis:
The query is asking about "What is the majority party in the country where Canberra is located in 2024?" The passages show that Canberra is the capital of Australia and the Labor Party is the majority party of Australia in 2024, which provides sufficient information to answer the query. Thus, my assessment is: "Answerable".
Assessment: ["Answerable"]

Now process the user query.

---

Figure A.14: Prompt Example on HotPotQA for Assessing if the Query is Answerable Using ⑤.

## I.1.2 FINE-TUNING PROMPT EXAMPLES

---

**Prompt for Generation Fine-tuning Datasets for Identifying $\mathcal{S}$ with 𝔈**

**System Prompt:** Task: Extract the **Key Sequence Snippet (KSS)** needed to match documents in the **current retrieval stage only**, taking the basic knowledge provided for reference.
Rules:
1. **Basic knowledge**: The knowledge already obtained to answer the query.
1. **Current-stage focus**: Ignore the solved or future hops (e.g., if query is "A → B → C", extract only "A → B" if Basic Knowledge covers "A").
2. **Minimal but sufficient**: Include only:
- key entities
- Critical query component
- other critical snippets
3. **Exclude**:
- Syntactic fluff
Output: Wrap KSS in '<KSS></KSS>'.
**Example 1**:
Query: "What is the date of birth of the director of film You're My Everything (Film)?"
Basic Knowledge: None.
KSS: '<KSS>You're My Everything (Film), director of You're My Everything (Film)</KSS>'
**Example 2**:
Query: "What is the date of birth of the director of film You're My Everything (Film)?"
Basic Knowledge:
1. "You're My Everything (Film): You're My Everything is a 1949 film directed by Walter Lang and starring Dan Dailey and Anne Baxter."
KSS: '<KSS>Walter Lang, date of birth of Walter Lang</KSS>'

Now process the user query.

---

Figure A.15: Prompt for Generation Fine-tuning Datasets for Identifying $\mathcal{S}$ with 𝔈.

---

**Prompt for Generation Fine-tuning Datasets for Extracting $\mathcal{R}$ with $\mathfrak{E}$**

**System Prompt:** Task #RS#: From the given passages, extract **Relevant Sentence (RS)** that are **directly relevant** to the **query** or its **Key Sequence Snippet (KSS)**, taking the basic knowledge provided for reference.
Where: **Basic knowledge**: The knowledge already obtained to answer the query; **KSS**: may be indirectly relevant to the query.
Each RS must:
1. **Help to answer the query**: Ignore information irrelevant to the query. Or:
2. **Relevant to the KSS**.
3. **Be minimal**: Include only the shortest sufficient sentence(s) from each passage.
4. **Not repeated**: Do not repeat the same sentences in the basic knowledge .
**Output Rules**:
- **One RS per sentence**
- Wrap each RS in '<RS></RS>' tags.
- If no relevant sentence exists in all passages, output '<RS>None</RS>'.
**Example1**:
Query: "What is the date of birth of the director of film You're My Everything (Film)?"
Basic Knowledge: None
KSS: "You're My Everything (Film), director of You're My Everything (Film)"
Passages:
1. "title: 'Walter Lang', text: "Walter Lang( August 10, 1896 – February 7, 1972) was an American film director.""
2. "title: 'You're My Everything (film)', text: "You're My Everything is a 1949 film directed by Walter Lang and starring Dan Dailey and Anne Baxter.""
3. "title: 'You're My Pet (film)', text: "You're My Pet is a 2011 South Korean romantic comedy film based on the manga of the same name,You're My Peẗby Yayoi Ogawa."It co-stars Kim Ha-neul and Jang Keun- suk and directed by Kim Byeong- kon."It is released on 10 November 2011 by Lotte and ran at 110 minutes.""
RS: <RS>You're My Everything is a 1949 film directed by Walter Lang and starring Dan Dailey and Anne Baxter.</RS>
<RS>Walter Lang (August 10, 1896 – February 7, 1972) was an American film director.</RS>

Now process the user query.

Figure A.16: Prompt for Generation Fine-tuning Datasets for Extracting $\mathcal{R}$ with $\mathfrak{E}$.

Prompt for Generation Fine-tuning Datasets for Assessing Relevance with $\mathfrak{F}$

**System Prompt:** Task #RS#: Task: Given several passages, assess the relevance of each passage to the **query** or its **Key Sequence Snippet (KSS)**, taking the basic knowledge provided for reference.
Where: **Basic knowledge**: The knowledge already obtained to answer the query; **KSS**: may be indirectly relevant to the query.
Assign a relevance score based on:
1. **1**: The passage provides necessary information to answer the query (even if not all information).
2. **0.5**: The passage does not provide information the query needs, but may be helpful to answer the query or is relevant to its KSS.
3. **0**: The passage is totally irrelevant to the query and its KSS.
Wrap your assessment in '<Relevance></Relevance>' tags.
**Example 1**:
Query: "What is the date of birth of the director of film You're My Everything (Film)?"
KSS: "Walter Lang, date of birth of Walter Lang"
Basic Knowledge:
1. "You're My Everything (film): You're My Everything is a 1949 film directed by Walter Lang and starring Dan Dailey and Anne Baxter."
Passages:
1. "You're My Pet (film): You're My Pet is a 2011 South Korean romantic comedy film."
2. "You're My Everything (film): You're My Everything is a musical comedy filmed in 1949."
3. "Walter Lang: Walter Lang (August 10, 1896 – February 7, 1972) was an American film director."
Relevance:
<Relevance>0</Relevance>
<Relevance>0.5</Relevance>
<Relevance>1</Relevance>

Now process the user query.

Figure A.17: Prompt for Generation Fine-tuning Datasets for Assessing Relevance with $\mathfrak{F}$.

**Prompt for Generation Fine-tuning Datasets for Assessing if Query is Answerable with ⓖ**

**System Prompt:** Task: From the given passages, determine whether the **query** is **answerable**, **partially answerable** or **unanswerable**, where:
1. **Answerable**: The passages provide sufficient information to answer the query.
2. **Partially Answerable**: The passages do not provide all information the query needs, but provide necessary information.
3. **Unanswerable**: The passages are totally irrelevant to the query at all.
Wrap your judgement in '<ANS></ANS>' tags.
**Example 1**:
Query: "What is the date of birth of the director of film You'Re My Everything (Film)?"
Passages:
1. "You're My Everything is a 1949 film directed by Walter Lang and starring Dan Dailey and Anne Baxter."
2. " Walter Lang (August 10, 1896 – February 7, 1972) was an American film director."
ANS: <ANS>Answerable</ANS>
**Example 2**:
Query: "Where was the director of You're My Everything born?"
Passages:
1. "You're My Everything is a 1949 film directed by Walter Lang."
ANS: <ANS>Partially Answerable</ANS>
**Example 3**:
Query: "Who composed the soundtrack for You're My Everything?"
Passages:
1. "You're My Everything is a 1949 film directed by Walter Lang."
ANS: <ANS>Unanswerable</ANS>

Now process the user query.

Figure A.18: Prompt for Generation Fine-tuning Datasets for Assessing if Query is Answerable with ⓖ.

### I.1.3 PROMPT USED FOR LLM-EM EVALUATION WITH GPT-4O

---

Prompt Used for LLM-EM Evaluation with GPT-4o

**System Prompt:** You are an excellent teacher. You will be given a query and its corresponding label, along with a student's answer to validate. Your task is to determine if the answer correctly answers the query based on the label.
Provide your validation in the form: Validation: ["Correct"] or Validation: ["Wrong"]. Learn from the instances below:
### Instance 1
#### User's input:
Query: "What is the majority party in the country where Canberra is located in 2024?"
Label: "The Labor Party."
Student's Answer: "The Labor Party is the majority party in the country where Canberra is located in 2024."
#### Expected response:
Analysis:
The student's answer correctly matches the label, as it accurately restates that the Labor Party is the majority party in the country where Canberra is located in 2024.
Validation: ["Correct"]
### Instance 2
#### User's input:
Query: "Are either Baz Warne or Marty Balin actors?"
Label: "no"
Student's Answer: "Baz Warne is an actor while Marty Balin is not."
#### Expected response:
Analysis:
The label indicates that neither Baz Warne nor Marty Balin are actors. However, the student's answer incorrectly states that Baz Warne is an actor. This contradicts the label. Therefore, the student's answer does not match the correct information provided by the label.
Validation: ["Wrong"]

Now process the user query.

---

Figure A.19: Prompt Used for LLM-EM Evaluation with GPT-4o.

## I.2 SAMPLES

### I.2.1 SAMPLES OF EXTRACTED BASIC KNOWLEDGE

To demonstrate FRAG's extraction performance, we select 2–3 test instances from each experimental dataset and present the relevant knowledge $\mathcal{K}$, which is extracted using Qwen2-7B-Instruct as the extractor model and subsequently used to construct the generation contexts.

---

**QA Examples from PopQA**

**Query:** What is Bruce McDaniel's occupation?
**Label:** ["Composer"]
**Extracted Basic Knowledge:**
1. Bruce McDaniel (born September 23, 1962): Bruce McDaniel is an American musician, composer, producer and recording engineer, currently living in New Orleans. (Relevance: 1)
2. John McDaniel: John McDaniel (born September 23, 1951 in Birmingham, Alabama) is a former American football wide receiver. (Relevance: 0.5)
3. Jerry McDaniel: McDaniel has also conceived and produced short films and film titles. (Relevance: 0.25)

**Query:** Who is the author of The Latimers?
**Label:** ["Henry Christopher McCook", "McCook"]
**Extracted Basic Knowledge:**
1. The Latimers: The Latimers : A Tale of the Western Insurrection of 1794 is an historical novel by the American writer and Presbyterian clergyman Henry Christopher McCook (1837–1911) set in 1790s Pittsburgh, Pennsylvania. (Relevance: 1)
2. Elizabeth Wormeley Latimer: Mary Elizabeth Wormeley Latimer (July 26, 1822 – January 4, 1904 ) was an English-American writer, both of original works and translations. (Relevance: 0.75)
3. Lewis Howard Latimer; Lewis Howard Latimer (September 4, 1848 – December 11, 1928) was an African-American inventor and patent draftsman. (Relevance: 0.5)
4. Jon Latimer: Jonathan David Latimer (1964 – 4 January 2009) was an historian and writer based in Wales. (Relevance: 0.5)
5. Alan Noel Latimer Munby: Alan Noel Latimer ('Tim') Munby (1913–1974) was an English author, writer and librarian. (Relevance: 0.5)

**Query:** In what country is Brizambourg?
**Label:** ["France", "fr", "FR", "République française", "La France", "Republic of France", "French Republic", "FRA", "the Hexagon"]
**Extracted Basic Knowledge:**
1. Brizambourg: Brizambourg is a commune in the Charente-Maritime department in southwestern France. (Relevance: 1)
2. Bourg-en-Bresse: It is the capital of the ancient province of Bresse (Brêsse) (Relevance: 0.5)
3. Bourg-en-Bresse: In the early 20th century, the city manufactured iron goods, mineral waters, tallow, soap and earthenware, and there were flour mills and breweries; and there is considerable trade in grain, cattle and poultry. (Relevance: 0.5)
4. Ansembourg: Ansembourg (Aansebuerg, Ansemburg) is a village in the commune of Helperknapp, in western Luxembourg. (Relevance: 0.5)

Figure A.20: Test samples in PopQA.

Test Samples in PubHealth

**Claim:** Prince Harry joins Elton John to launch HIV campaign targeting men.
**Label:** True
**Extracted Basic Knowledge:**
1. Prince Harry, Duke of Sussex: In July 2018, the Elton John AIDS Foundation announced that the Duke of Sussex and British singer Elton John were about to launch a global coalition called MenStar that would focus on treating HIV infections in men. (Relevance: 1)
2. Prince Harry, Duke of Sussex: To raise awareness for HIV testing, Harry took a test live on the royal family Facebook page on 14 July 2016. He later attended the 21st International AIDS Conference in Durban, South Africa, on 21 July 2016. (Relevance: 0.5)
3. On World Aids Day, Harry and Rihanna helped publicise HIV testing by taking the test themselves. (Relevance: 0.5)
4. Elton John AIDS Foundation: In 2016, Elton John's AIDS Foundation (EJAF) partnered with the Elizabeth Taylor AIDS Foundation (ETAF) in an initiative to combat AIDS in the Southern United States. (Relevance: 0.5)
5. Elton John has been involved in efforts against HIV/AIDS. (Relevance: 0.5)
5. Born HIV Free: world-class artists such as Paul McCartney, U2, Amy Winehouse, Jean-Paul Gaultier, H5, and the Bonzoms were involved in the campaign. (Relevance: 0.25)
6. James Prince: HIV/AIDS prevention and testing with the launching of Strapped, in coordination with a string of initiatives and events set up to address the issue of AIDS in the black community. (Relevance: 0)
7. Terrence Higgins Trust: Elton John. (Relevance: 0)

**Claim:** Strobe lighting provides a flicker of hope in the fight against Alzheimer's.
**Label:** False
**Extracted Basic Knowledge:**
1. Strobe light: Strobe light A strobe light or stroboscopic lamp, commonly called a strobe, is a device used to produce regular flashes of light. (Relevance: 0.5)
2. Strobe light: Sometimes strobe lighting can trigger seizures in photosensitive epilepsy. (Relevance: 0.5)
3. Lighting: Designing lighting systems that maximize the right amount of light at the appropriate time of day for the elderly may help relieve symptoms of Alzheimer's Disease. (Relevance: 0.25)
4. Lighting for the elderly: Indirectly, the passage suggests the importance of maintaining proper light exposure patterns, which could potentially be explored for therapeutic uses in managing conditions like Alzheimer's. (Relevance: 0.25)
5. Strobe light: Strobe lights are used in scientific and industrial applications. (Relevance: 0.25)
6. Strobe light: Strobe lights are used in scientific and industrial applications, in clubs where they are used to give an illusion of slow motion, and are often used for aircraft anti-collision lighting both on aircraft themselves and also on tall stationary. (Relevance: 0)

Figure A.21: Test samples in PubHealth (a).

**Test Samples in PubHealth**

**Claim:** John Holdren, director of the White House Office of Science and Technology Policy, has proposed forcing abortions and putting sterilants in the drinking water to control population.
**Label:** False
**Extracted Basic Knowledge:**
1. John Holdren: the nomination committee that he does not believe that government should have a role in determining optimal population size and that he never endorsed forced sterilization. (Relevance: 0.75)
2. John Holdren: John Holdren John Paul Holdren (Sewickley, Pennsylvania, March 1, 1944) is an American scientist who served as the senior advisor to President Barack Obama on science and technology issues through his roles as Assistant to the President for Science and Technology, Director of the White House Office of Science and Technology Policy, and Co-Chair of the President's Council of Advisors on Science and Technology (PCAST). (Relevance: 0.5)
3. John Holdren: the dangers from nuclear weapons and materials, and science and technology policy. (Relevance: 0.5)
4. Human overpopulation: policies are making it easier and more socially acceptable to use contraception and abortion methods. (Relevance: 0.5)
5. Larry Bucshon: During a September 17, 2014 hearing of the Committee on Science, Space and Technology, Bucshon was questioning John Holdren, Director of the White House Office of Science and Technology Policy. (Relevance: 0.5)
6. Penny4NASA: letter to White House Office of Science and Technology Policy Director John Holdren, acknowledging fiscal challenges, but adding that they were concerned that the message of (Relevance: 0.25)
7. Larry Bucshon: Bucshon was questioning John Holdren, Director of the White House Office of Science and Technology Policy. (Relevance: 0.25)

Figure A.22: Test samples in PubHealth (b).

| | Test Samples in ARC-Challenge |
|---|---|

**Query:** At which temperature does water freeze?
**Choices:** A:0 degrees Celsius; B:32 degrees Celsius; C:100 degrees Celsius; D:212 degrees Celsius.
**Label:** A
**Extracted Basic Knowledge:**
1. The freezing level, or 0 °C (zero-degree) isotherm represents the altitude in which the temperature is at 0 °C (the freezing point of water) in a free atmosphere. (Relevance: 1)
2. Water will freeze at different temperatures depending upon the type of ice nuclei present. (Relevance: 1)
3. Water normally freezes at 273.15 K (0 °C or 32 °F). (Relevance: 1)
4. Water (at atmospheric pressure) does not freeze at 0° C, but rather at temperatures that tend to decrease as the volume of the water decreases and as the water impurity increases. (Relevance: 0.75)
5. When water is in a conventional freezer, a dynamic phase transition is triggered. The resulting ice depends on how quickly the system is cooled: If the water is cooled below its freezing point slowly, an ice crystal will result, rather. (Relevance: 0.5)
6. Water at about 4 °C (39 °F) also sinks to the bottom, thus keeping the temperature of the water at the bottom constant (see diagram). (Relevance: 0)
7. However, even with this definition it is not clear whether freezing refers to the point at which water forms a visible surface layer of ice; the point at which the entire volume of water becomes a solid block of ice; or when the water reaches. (Relevance: 0)

**Query:** Which is a fact about penguins?
**Choices:** A:Penguins can live in climates with freezing temperatures. B:Penguins are fierce competitors. C:Penguins are some of the most beautiful birds. D:Penguins make great pets.
**Label:** A
**Extracted Basic Knowledge:**
1. Penguin Penguins (order Sphenisciformes, family Spheniscidae) are a group of aquatic, flightless birds.Highly adapted for life in the water, penguins have countershaded dark and white plumage, and their wings have evolved into flippers.Although almost all penguin species are native to the Southern Hemisphere. (Relevance: 1)
2. Although almost all penguin species are native to the Southern Hemisphere, they are not found only in cold climates, such as Antarctica. In fact, only a few species of penguin actually live so far south. Several species live in the temperate zone; one, the Galápagos penguin, lives as far north.Emperor penguin The emperor penguin (Aptenodytes forsteri) is the tallest and heaviest of all living penguin species and is endemic to Antarctica. (Relevance: 1)
3. Penguins can live in climates with freezing temperatures. (Relevance: 1)
4. Galápagos penguin The Galápagos penguin (Spheniscus mendiculus) is a penguin endemic to the Galápagos Islands.It is the only penguin that lives north of the equator.The Galápagos penguin is one of the banded penguins, the other species of which live mostly on the coasts of Africa and mainland South America. It can survive due to the cool temperatures resulting from the Humboldt Current and cool waters from great depths brought up by the Cromwell Current. (Relevance: 1)
5. Although almost all penguin species are native to the Southern Hemisphere, they are not found only in cold climates, such as Antarctica. In fact, only a few species of penguin actually live so far south. Several species live in the temperate zone; one, the Galápagos penguin, lives as far north. (Relevance: 0.75)
6. Several authors have suggested that penguins are a good example of Bergmann's Rule where larger bodied populations live at higher latitudes than smaller bodied populations.There is some disagreement about this, and several other authors have noted that there are fossil penguin species that contradict this hypothesis and that ocean currents and upwellings are likely to have had a greater effect on species diversity than latitude alone. (Relevance: 0.75)
7. Emperor penguin The emperor penguin (Aptenodytes forsteri) is the tallest and heaviest of all living penguin species and is endemic to Antarctica.Like all penguins it is flightless, with a streamlined body, and wings stiffened and flattened into flippers for a marine habitat. (Relevance: 0.25)

Figure A.23: Test samples in ARC-Challenge (a).

Test Samples in ARC-Challenge

**Query:** Burning fossil fuels produces sulfur dioxide (SO2) and nitrogen oxide (NO). These compounds react with water vapor to produce acid rain. What is the most likely effect of acid rain on the environment where it falls?

**Choices:** A:The plants and animals in lakes and ponds will be harmed. B:The soil in the area will become more alkaline. C:The thickness of the ozone layer will decrease. D:Levels of air pollution will increase.

**Label:** A

**Extracted Basic Knowledge:**

1. Acid rain can damage infrastructures containing calcite or other solid chemical compounds containing carbon. In ecosystems, acid rain can dissolve plant tissues of vegetations and increase acidification process in bodies of water and in soil. (Relevance: 1)

2. Acid rain is caused by the emission of nitrogen oxides and sulfur dioxide. These gases may be only mildly acidic themselves, yet when they react with the atmosphere, they create acidic compounds such as sulfurous acid, nitric acid and sulfuric acid which fall as rain, hence the term acid rain. (Relevance: 1)

3. the most important gas which leads to acidification is sulphur dioxide. Emissions of nitrogen oxides which are oxidized to form nitric acid are of increasing importance due to stricter controls on emissions of sulphur containing compounds. Thus, for example, fumaroles from the Laguna' (Relevance: 0.75)

4. with water and oxygen in the atmosphere, creating nitric acid and sulfuric acids, which return to Earth's surface as acid deposition, or acid rain. Acid deposition harms aquatic organisms and kills trees. Due to its formation of certain nutrients which are less available to plants such as calcium and phosphorus, it reduces the productivity of ecosystem and farms. (Relevance: 0.75)

5. Acid rain Acid rain is a rain or any other form of precipitation that is unusually acidic, meaning that it has elevated levels of hydrogen ions (low pH). It can have harmful effects on plants, aquatic animals and infrastructure. (Relevance: 0.75)

6. Sulfur dioxide and nitrogen oxides are naturally released from volcanoes, organic compounds in the soil, wetlands, and marine systems, but the majority of these compounds come from the combustion of coal, oil, gasoline, and the smelting of ores containing sulfur. These substances dissolve in atmospheric moisture and enter lentic systems as acid rain. Lakes and ponds that contain bedrock that is rich in carbonates have a natural buffer, resulting in no alteration of pH. Systems without this bedrock, however, are very sensitive to acid inputs because they have a low neutralizing capacity, resulting in pH declines even with. (Relevance: 0.5)

7. Approximately 75 Tg/S per year of sulfur dioxide (SO) is released from burning coal. (Relevance: 0.5)

8. The pH change is most marked in rivers with very (Relevance: 0.5)

9. Finlayson-Pitts served as the lead author of a 2009 study published in the Proceedings of the National Academy of Sciences that found that burning fossil fuels releases nitrogen oxides, which interact with gaseous hydrogen chloride to form smog-forming compounds. (Relevance: 0.5)

10. Sulfur dioxide and nitrogen oxides are primary causes of acid rain. These by-products are still a problem, but they have been greatly diminished in most advanced countries due to clean air regulations. (Relevance: 0.25)

11. Acid precipitation can lead to asthma, bronchitis, lung inflammation, emphysema, and other lung and heart diseases. (Relevance: 0)

12. Acid-producing gasses are also created by biological processes that occur on the land, in wetlands, and in the oceans. The major biological source of sulfur containing compounds is dimethyl sulfide. Nitric acid in rainwater is an important source of fixed nitrogen for plant life, and is also produced by electrical activity in the atmosphere such as lightning. (Relevance: 0)

Figure A.24: Test samples in ARC-Challenge (b).

> **Test Samples in HotPotQA**
>
> **Query:** The place where John Laub is an American criminologist and Distinguished University Professor in the Department of Criminology and Criminal Justice at was founded in what year?
> **Label:** 1856.
> **Extracted Basic Knowledge:**
> 1. John Laub: John H. Laub (born 1953) is an American criminologist and Distinguished University Professor in the Department of Criminology and Criminal Justice at the University of Maryland, College Park. (Relevance: 1)
> 2. University of Maryland, College Park: The University of Maryland, College Park (often referred to as the University of Maryland, Maryland, UM, UMD, UMCP, or College Park) is a public research university located in the city of College Park in Prince George's County, Maryland, approximately 4 mi from the northeast border of Washington, D.C. Founded in 1856, the university is the flagship institution of the University System of Maryland. With a fall 2010 enrollment of more than 37,000 students, over 100 undergraduate majors, and 120 graduate programs, Maryland is the largest university in the state and the largest in the Washington Metropolitan Area. It is a member of the Association of American Universities and competes in athletics as a member of the Big Ten Conference. (Relevance: 0.75)
> 3. Lawrence W. Sherman: He is also a Distinguished University Professor at the University of Maryland's Department of Criminology and Criminal Justice in College Park. Founded in 1856. (Relevance: 0.75)
> 4. Charles Wellford: University of Maryland, College Park. (Relevance: 0.5)
>
> **Query:** Are the bands "Halestorm" and "Say Anything" from different states?
> **Label:** yes.
> **Extracted Basic Knowledge:**
> 1. Halestorm: Halestorm is an American hard rock band from Red Lion, Pennsylvania... (Relevance: 1)
> 2. Say Anything (band): Say Anything is an American rock band from Los Angeles, California. (Relevance: 1)
> 3. Max Bemis: Max Bemis is the lead singer, primary composer and primary lyricist of the band Say Anything. Say Anything is an American rock band from Los Angeles, California. (Relevance: 0.5)
> 4. The MySpace Transmissions (Say Anything EP): The MySpace Transmissions is a digital EP by Say Anything. (Relevance: 0.5)
>
> **Query:** Which Canadian province did the famous computer scientist John Tsotsos serve as the Director of the Centre for Vision Research at a famous research university?
> **Label:** Ontario
> **Extracted Basic Knowledge:**
> 1. John Tsotsos: John Tsotsos is a Canadian Computer Scientist whose research focuses on the field of Computer Vision. He is currently the Canada Research Chair in Computer Vision at York University and served as the Director of the Centre for Vision Research at York University from 2000-2006. (Relevance: 1)
> 2. York University: York University (French: "Université York" ) is a public research university in Toronto, Ontario, Canada. (Relevance: 0.75)
> 3. Matti Pietikäinen (academic): He is Director of the Center for Machine Vision Research, and Scientific Director of Infotech Oulu. (Relevance: 0)
> 4. McCarthy Formalism: In computer science and recursion theory the McCarthy Formalism (1963) of computer scientist John McCarthy clarifies the notion of recursive functions by use of the IF-THEN-ELSE construction common to computer science, together with four of the operators of primitive recursive functions: zero, successor, equality of numbers and composition. (Relevance: 0)

Figure A.25: Test samples in HotPotQA.

**Test Samples in 2WikiMultiHopQA**

**Query:** Which film has the director who was born later, El Extraño Viaje or Love In Pawn?
**Label:** El Extraño Viaje
**Extracted Basic Knowledge:**
1. Charles Saunders (director): Charles Joel Saunders (8 April 1904 – April 1997) was an English film director and screenwriter who started in the industry as a film editor, and who also contributed to television. (Relevance: 1)
2. Fernando Fernán Gómez: Fernando Fernández Gómez (28 August 1921 – 21 November 2007) better known as Fernando Fernán-Gómez was a Spanish actor, screenwriter, film director, theater director and member of the Royal Spanish Academy for seven years.. (Relevance: 1)
3. Rafaela Aparicio: The most remembered are Carlos Saura's 'Anna and the Wolves' Mama Turns 100' and Fernando Fernán Gómez's 'El extraño viaje'. (Relevance: 0.75)
4. El extraño viaje: El extraño viaje is a 1964 Spanish black drama film directed by Fernando Fernán Gómez. (Relevance: 0.75) 5. Fernando Fernán Gómez: Fernando Fernández Gómez (28 August 1921 – 21 November 2007) (Relevance: 0.75) 6. Love in Pawn: Love in Pawn is a 1953 British comedy film directed by Charles Saunders and starring Bernard Braden and Barbara Kelly. (Relevance: 0.75)

**Query:** Which film has the director died first, Crimen A Las Tres or The Working Class Goes To Heaven?
**Label:** The Working Class Goes To Heaven
**Extracted Basic Knowledge:**
1. Luis Saslavsky: He died in Buenos Aires, aged 91. (Relevance: 1)
2. The Mattei Affair: The film shared the 'Grand Prix' with 'The Working Class Goes to Heaven' at the 1972 Cannes Film Festival. (Relevance: 0.75)
3. The Working Class Goes to Heaven: The Working Class Goes to Heaven( released in the US as Lulu the Tool) is a 1971 political drama film directed by Elio Petri. (Relevance: 0.75)
4. Elio Petri: Elio Petri( 29 January 1929 – 10 November 1982) was an Italian political filmmaker best known for the 1970 Academy Award- winning film 'Investigation of a Citizen Above Suspicion'. (Relevance: 0.75)
5. Crimen a las tres: Luis Saslavsky directed and wrote 'Crimen a las tres'.", (Relevance: 0.5)
6. "Escala en la ciudad: The production company disbanded the following year, after de Zavalia had madeËscala en la ciudad, his feature film debut, and Saslavsky had completed his second and most famous movie, C̈rimen a las tres. (Relevance: 0.5)
7. Luis Saslavsky: Luis Saslavsky( April 21, 1903 – March 20, 1995) was an Argentine film director, screenwriter and film producer, and one of the influential directors in the Cinema of Argentina of the classic era. (Relevance: 0.5)

Figure A.26: Test samples in 2WikiMultiHopQA.

**Test Samples in MuSiQue**

**Query:** Who is the child of Caroline LeRoy's spouse?
**Label:** Fletcher Webster
**Extracted Basic Knowledge:**
1. Caroline LeRoy: Caroline LeRoy Webster (September 28, 1797 in New York City – February 26, 1882) was the second wife of 19th Century statesman Daniel Webster. Her father was Herman LeRoy. (Relevance: 0.75)
2. Fletcher Webster: Daniel Fletcher Webster, commonly known as Fletcher Webster (July 25, 1813 in Portsmouth, New Hampshire – August 30, 1862) was the son of renowned politician Daniel Webster and Grace Fletcher Webster. (Relevance: 0.5)
3. Caroline LeRoy: Her father was Herman LeRoy, who was once head of the commercial house of Leroy, Bayard, McKiven & Co., a large trading company that operated in different parts of the world. (Relevance: 0.5)

**Query:** What company succeeded the owner of Empire Sports Network?
**Label:** Time Warner Cable
**Extracted Basic Knowledge:**
1. Empire Sports Network: Empire Sports Network was an American regional sports network that was owned by the Adelphia Communications Corporation. (Relevance: 0.75)
2. Windjammer Communications: Windjammer Cable is a small cable company formed by the sale of 25 systems that served 80,000 customers in rural areas that Time Warner Cable acquired from the bankrupt Adelphia. (Relevance: 0.75)

Figure A.27: Test samples in MuSiQue.

---

**Test Samples in LongBench-v2**

**Query:** Which of the following is an incorrect understanding of the discussion on Journey to the West in the development of Chinese mythological culture?
Choices:
A. Journey to the West expresses the characteristics of mythological figures where divinity, animal instincts, and human nature are integrated through the character of Sun Wukong.
B. The "social circle" in Journey to the West symbolizes the refinement of human character and the transformation of spiritual will.
C. The novel elevates a free mindset beyond religion by incorporating the ideological resources of Confucianism, Buddhism, and Daoism, blending in the mysticism of the three teachings.
D. The image of Sun Wukong symbolizes the transformation from the outburst of wild vitality to the elevation of spiritual realms.
**Label:** C
**Extracted Basic Knowledge:**
1. Journey to the West Chapter 24: Blessed Land of the Mountain of Infinite Longevity - Cave Heaven of the WuZhuang Temple: Although the Tang Priest is an old friend of mine,"said the Great Immortal, "you must be on your guard against his ruffian followers, and you mustn't let them know about the manfruit. It's either a Taoist temple or a Buddhist one. Let's go over and find out. Residence of Divine Immortals Who Never Grow Old; Home of Taoists as Ancient as Heaven. (Relevance: 0.5)
2. On his head A leopard skin hat with artemisia patterns: On his body A coat of woollen cloth. Round his waist was tied a lion belt, On his feet a pair of deerskin boots. His eyes were as round as an evil spirit's; His curly beard was like the evil god of the moon's. From his waist hung a bow with poisoned arrows, And in his hand was a steel-tipped trident. (Relevance: 0.25)

**Query:** Which of the following statements is incorrect?
Choices:
A. This article inserts a module into the pre-trained diffusion model, and then trains the parameters of these models to adapt this module to the task and the priori of the diffusion model.
B. TPB includes two MLP layers with Layer Normalization and LeakyReLU, ensuring that only the most task-specific attributes are retained
C. Task-specific priors containing guidance information for the task can adequately guide pre-trained diffusion models to handle low-level tasks while maintaining high-fidelity content consistency.
D. The spatial feature Fs extracted by SCB processing is calculated from SCB, Ft, Fp, F and has no relationship with TPB.
**Label:** D
**Extracted Basic Knowledge:**
1. Introducing Diff-Plugin: A Framework for Enhancing Pre-Trained Diffusion Models with Task-Specific Guidance: The spatial feature Fs extracted by SCB processing is calculated from SCB, Ft, Fp, F, and has no relationship with TPB. (Relevance: 1)
2. Implementation: During training and testing, we resize the image to 512×512 for a fair comparison. (Relevance: 0.25)

Figure A.28: Test samples in LongBench-v2.

### I.2.2 SAMPLES OF INACCURATE RELEVANCE SCORING

---

Samples of Inaccurate Relevance Scoring

**Query:** Which member of the band Bad Seeds was older, Anita Lane or Nick Cave?
**Extracted Knowledge:**
1. Anita Lane: Anita Louise Lane (born ca. 1959) is an Australian singer-songwriter who was briefly a member of the Bad Seeds with Nick Cave and Mick Harvey, and has collaborated with both former band mates. (Relevance: 1)
2. Nick Cave: Nicholas Edward Cave (born 22 September 1957) is an Australian musician, singer-songwriter, author, screenwriter, composer and occasional film actor, best known as the frontman of the rock band Nick Cave and the Bad Seeds. (Relevance: 0.75)

**Query:** What album did a Danish-born Montenegrin singer born in 1971 release in late spring 2008?
**Extracted Knowledge:**
1. Boban Rajović: Boban Rajović (born 25 December 1971) is a Danish-born Montenegrin singer popular in former Yugoslavia. (Relevance: 0.5)
2. Kosači: Kosači (English translation: Mowers) is the fifth studio album by Montenegrin singer Boban Rajović. t was released in late spring 2008. (Relevance: 1)

---

Figure A.29: Samples of Inaccurate Relevance Scoring.

### I.2.3 SAMPLES OF DISCREPANCIES BETWEEN GPT-4O AND HUMAN ASSESSMENTS

---

**Samples of Discrepancies**

**Query:** Musician and satirist Allie Goertz wrote a song about the "The Simpsons" character Milhouse, who Matt Groening named after who?

**Labels:** President Richard Nixon.

**Retrieved Documents:**

1. Allie Goertz: Allison Beth "Allie" Goertz (born March 2, 1991) is an American musician. Goertz is known for her satirical songs based on various pop culture topics. Her videos are posted on YouTube under the name of Cossbysweater. Subjects of her songs have included the film "The Room", the character Milhouse from the television show "The Simpsons", and the game Dungeons & Dragons. Her style has been compared to that of Bo Burnham. In December 2015, Goertz released a concept album based on the Adult Swim series "Rick and Morty", "Sad Dance Songs", with the album's cover emulating the animation and logo of the series. The album was made possible through Kickstarter. She is co-host of Everything's Coming Up Podcast, a Simpsons-focused podcast along with Julia Prescott.

2. Milhouse Van Houten: Milhouse Mussolini van Houten is a fictional character featured in the animated television series "The Simpsons", voiced by Pamela Hayden, and created by Matt Groening who named the character after President Richard Nixon's middle name. Later in the series, it is revealed that Milhouse's middle name is "Mussolini."

**RALM Prediction:** President Richard Nixon's middle name.

**Evaluation (GPT-4o, llm-em):** Incorrect. The label states that Matt Groening named the "The Simpsons" character Milhouse after President Richard Nixon. The student's answer states that Milhouse was named after President Richard Nixon's middle name, which is "Milhous." While technically related, the student's answer is not an exact match to the label and introduces potentially misleading specificity.

**Evaluation (human, human-acc):** Correct.

**Justification for Assessment:** The RALM prediction is accurate and supported by the retrieved documents. Its specificity enhances clarity without altering the core meaning.

**Query:** Which player singed to Bayern Munich was born in 18 October 1983?

**Labels:** Dante Bonfim Costa Santos (born 18 October 1983)

**Retrieved Documents:**

1. 2012–13 FC Bayern Munich season: The 2012–13 FC Bayern Munich season was the 114th season in the club's history and the 48th consecutive season in the top flight of German football, the Bundesliga, since the promotion of the team from the Regionalliga Süd in 1965. Before the start of the season, Bayern signed Xherdan Shaqiri, Dante, Claudio Pizarro, Mitchell Weiser, Tom Starke and Mario Mandžukić. Bayern also added holding midfielder Javi Martínez after the first week of the Bundesliga season at the transfer deadline. The club started the season with a nine-match winning streak. The club would end the season claiming the Treble, winning the Bundesliga, the UEFA Champions League and the DFB-Pokal. Bayern are the first German club to achieve the Treble and are the third European Club to complete the Treble in the last five seasons and seventh ever in European Club competition.

2. Dante (footballer): Dante Bonfim Costa Santos (born 18 October 1983), commonly known as Dante () ), is a Brazilian professional footballer who plays for French Ligue 1 club Nice. Primarily a central defender, he has previously also been used as a defensive midfielder or a left back.

**RALM Prediction:** Dante Bonfim Costa Santos (Dante).

**Evaluation (GPT-4o, llm-em):** Incorrect. The label specifies that Dante Bonfim Costa Santos, born on 18 October 1983, is the player signed to Bayern Munich. The student's answer gives the name "Dante Bonfim Costa Santos (Dante)" but does not include the birthdate. The essential information about the birthdate is missing in the student's answer.

**Evaluation (human, human-acc):** Correct.

**Justification for Assessment:** The RALM prediction accurately identifies Dante Bonfim Costa Santos as the player in question. While it does not explicitly mention the birthdate, it provides all necessary information to address the query correctly.

---

Figure A.30: **Samples of discrepancies** (a). In the evaluation process with GPT-4o, we only provide the queries, ground truth labels, and the predicted answers from RALM, without including any retrieval documents. Additionally, in the GPT-4o prompt, we substitute the RALM with the role of a student answering the question.

---

Samples of Discrepancies

**Query:** When was Umar Farouk Abdulmutallab attempting to detonate plastic explosives on Northwest Airlines Flight 253?

**Labels:** Christmas Day, December 25, 2009

**Retrieved Documents:**

1. Umar Farouk Abdulmutallab: Umar Farouk Abdulmutallab; also known as Umar Abdul Mutallab and Omar Farooq al-Nigeri; born December 22, 1986) popularly referred to as the "Underwear Bomber", is a Nigerian man who, at the age of 23, confessed to and was convicted of attempting to detonate plastic explosives hidden in his underwear while on board Northwest Airlines Flight 253, en route from Amsterdam to Detroit, Michigan, on Christmas Day, 2009.

2. Northwest Airlines Flight 253: Northwest Airlines Flight 253 was an international passenger flight from Amsterdam Airport Schiphol in Haarlemmermeer, Netherlands, to Detroit Metropolitan Wayne County Airport in Romulus, Michigan, United States. The flight was the target of a failed al-Qaeda bombing attempt on Christmas Day, December 25, 2009, in which a passenger tried to set off plastic explosives sewn to his underwear. There were 290 people on board the aircraft—an Airbus A330-323E operated by Northwest Airlines, which had merged with Delta Air Lines the year before. Had the attempt succeeded, it would have surpassed American Airlines Flight 191 as the deadliest aviation occurrence on U.S. soil and tied Iran Air Flight 655 as the eighth-deadliest of all time. The incident was also the second in 2009 involving an Airbus A330, after the crash of Air France Flight 447 on June 1. This was also the final accident/incident involving Northwest Airlines as it closed down a month later when it merged with Delta Air Lines.

**RALM Prediction:** Christmas Day, 2009.

**Evaluation (GPT-4o, llm-em):** Incorrect. The student's answer states "Christmas Day, 2009", which is mostly correct but lacks the specific date "December 25" as provided in the label. The label specifies "Christmas Day, December 25, 2009", and the student's answer omits "December 25".

**Evaluation (human, human-acc):** Correct.

**Justification for Assessment:** The RALM prediction accurately identifies the timing of the incident as "Christmas Day, 2009." While it lacks the exact date included in the reference label, it provides enough information to satisfy the query and aligns with the retrieved evidence.

**Query:** In what year was the creator of the current arrangement of the "Simpson's Theme" born?

**Labels:** March 28, 1941.

**Retrieved Documents:**

1. Alf Clausen: Alf Heiberg Clausen (born March 28, 1941) is an American film and television composer. He is best known for his work scoring many episodes of "The Simpsons", of which he had been the sole composer between 1990 and 2017. Clausen has scored or orchestrated music for more than 30 films and television shows, including "Moonlighting", "The Naked Gun", "ALF" and "Ferris Bueller's Day Off".

2. The Simpsons Theme: "The Simpsons Theme", also referred to as "The Simpsons Main Title Theme" in album releases, is the theme music of the animated television series "The Simpsons". It plays during the opening sequence and was composed by Danny Elfman in 1989, after series creator Matt Groening approached him requesting a retro-style theme. The piece, which took 3 days, 2 hours, 48 minutes, and 19 seconds to create, has been noted by Elfman as the most popular of his career. The theme, as used for the opening sequence, was re-arranged during season 2, and the current arrangement by Alf Clausen was introduced at the beginning of the third season.

**RALM Prediction:** 1941.

**Evaluation (GPT-4o, llm-em):** Incorrect. The label specifies the exact birthdate, "March 28, 1941", while the student's answer provides only the year, "1941". Although the year is correct, it does not fully match the detailed information given in the label.

**Evaluation (human, human-acc):** Correct.

**Justification for Assessment:** The RALM prediction accurately answers the query by providing the year of birth (1941). While it is less detailed than the reference label, it remains fully consistent with the retrieved evidence and satisfies the query's requirements.

Figure A.31: **Samples of discrepancies** (b).

---

**Samples of Discrepancies**

**Query:** What is the name of the widow of the earl who served as both Foreign Secretary and Prime Minster from 1955 to 1957?

**Labels:** Anne Clarissa Eden.

**Retrieved Documents:**

1. Clarissa Eden: Anne Clarissa Eden, Dowager Countess of Avon ("née" Spencer-Churchill; born 28 June 1920) is the widow of Anthony Eden, 1st Earl of Avon (1897–1977), who was British Prime Minister from 1955 to 1957. She married Eden in 1952, becoming Lady Eden in 1954 when he was made a Knight of the Garter, and then becoming Countess of Avon in 1961 on her husband's elevation to the peerage. She is also the niece of the prime minister Winston Churchill. Her memoir, sub-titled "From Churchill to Eden", was published in 2007 under the name of Clarissa Eden.

2. Anthony Eden: Robert Anthony Eden, 1st Earl of Avon, (12 June 1897 – 14 January 1977) was a British Conservative politician who served three periods as Foreign Secretary and then a relatively brief term as Prime Minister of the United Kingdom from 1955 to 1957.

**RALM Prediction:** Clarissa Eden.

**Evaluation (GPT-4o, llm-em):** Incorrect.The label specifies "Anne Clarissa Eden" as the name of the widow. The student's answer "Clarissa Eden" provides a partial but not completely accurate answer, as it omits the first name "Anne". Therefore, the student's answer does not fully match the label.

**Evaluation (human, human-acc):** Correct.

**Justification for Assessment:** The RALM prediction accurately identifies the widow as "Clarissa Eden." While it omits her full name, the answer is precise, widely recognized, and supported by the retrieved documents.

**Query:** "The Little Sisters of Eluria" was included in which American author's collection "Everything's Eventual"?

**Labels:** Stephen King.

**Retrieved Documents:**

1. The Little Sisters of Eluria: "The Little Sisters of Eluria" is a Fantasy fiction novella written by Stephen King. It was originally published in 1998 in the anthology "Legends". In 2002, it was included in King's collection "Everything's Eventual". In 2009, it was published together with the of by Grant in a limited edition of 4,000 numbered copies of the Artist Edition signed by illustrator Michael Whelan and 1,250 numbered copies of the Deluxe Edition signed by Whelan and Stephen King. Both editions contain Whelan's additional new illustrations for "The Gunslinger".

2. Stephen King: Stephen Edwin King (born September 21, 1947) is an American author of horror, supernatural fiction, suspense, science fiction, and fantasy. His books have sold more than 350 million copies, many of which have been adapted into feature films, miniseries, television series, and comic books. King has published 54 novels, including seven under the pen name Richard Bachman, and six non-fiction books. He has written nearly 200 short stories, most of which have been collected in book collections. Many of his stories are set in his home state of Maine. His novella "Rita Hayworth and Shawshank Redemption" was the basis for the film "The Shawshank Redemption" which is widely regarded as one of the greatest films of all time.

**RALM Prediction:** Stephen King's collection "Everything's Eventual".

**Evaluation (GPT-4o, llm-em):** Incorrect. The student's answer mentions the collection "Everything's Eventual" but does not explicitly state that "The Little Sisters of Eluria" was included in Stephen King's collection. Although it hints at the correct author, it is not a complete match to the query and label.

**Evaluation (human, human-acc):** Correct.

**Justification for Assessment:** The RALM prediction identifies the correct author, Stephen King, and provides additional context about the collection Everything's Eventual. This additional detail is accurate and does not conflict with the reference label.

Figure A.32: Samples of discrepancies (c).

Samples of Discrepancies

**Query:** What county music artist first gained national attention as a winner on "You Can Be a Star" and wen on to provide background vocals in 199 on the 1992 release of the song "Take It Like a Man"?

**Labels:** Lari Michele White.

**Retrieved Documents:**

1. "Take It Like a Man (Michelle Wright song)": "Take It Like a Man" is a song written by Tony Haselden, and recorded by Canadian country music artist Michelle Wright. It was released in February 1992 as the first single from Wright's album "Now and Then". The song became Wright's first Number One on the Canadian "RPM" Country Tracks charts that year. It was also her only top 10 hit on the "Billboard" Hot Country Singles & Tracks chart in the United States. Country singer Lari White is featured on background vocals.

2. Lari White: Lari Michele White (born May 13, 1965) is an American country music artist and actress. She first gained national attention in 1988 as a winner on "You Can Be a Star", a talent competition which aired on The Nashville Network. A recording contract with RCA Records Nashville followed a year later, producing three studio albums, a greatest hits package, and several chart singles, with three of her singles having reached Top Ten: "That's My Baby" and "That's How You Know (When You're In Love)" at No. 10, and "Now I Know" at No. 5. A fourth studio album was released in 1998 on Lyric Street Records, followed by two more releases on White's own label, Skinny White Girl. Overall, White has charted 12 times on the "Billboard" country music charts.

**RALM Prediction:** "Lari White", "Take It Like a Man".

**Evaluation (GPT-4o, llm-em):** Incorrect. The student's answer "Lari White" correctly identifies the country music artist who first gained national attention as a winner on "You Can Be a Star." However, the additional phrase "Take It Like a Man" is unnecessary and does not fully restate the correct label "Lari Michele White." The answer is partially correct but not entirely accurate as per the label.

**Evaluation (human, human-acc):** Correct.

**Justification for Assessment:** The RALM prediction correctly identifies Lari White as the artist and provides additional context by mentioning the song title, Take It Like a Man. This expanded response enhances the completeness of the answer while maintaining full alignment with the query and evidence.

**Query:** Caldera de Taburiente National Park and Teide National Park are located in which island chain?

**Labels:** Canary Islands, Spain

**Retrieved Documents:**

1. Caldera de Taburiente National Park: Caldera de Taburiente National Park (Spanish: "Parque Nacional de la Caldera de Taburiente" ) is large geological feature on the island of La Palma, Canary Islands, Spain. It contains the enormous expanse of the Caldera de Taburiente which dominates the northern part of the island, and was designated as a national park in 1954. The telescopes of the Roque de los Muchachos Observatory are situated very close to the summit.

2. Teide National Park': Teide National Park (Spanish: "Parque nacional del Teide" , ) ) is a national park located in Tenerife (Canary Islands, Spain).

**RALM Prediction:** The Canary Islands.

**Evaluation (GPT-4o, llm-em):** Incorrect. The student's answer, "The Canary Islands," partially matches the label "Canary Islands, Spain." While it correctly identifies the island chain, it omits the additional detail that the Canary Islands are part of Spain.

**Evaluation (human, human-acc):** Correct.

**Justification for Assessment:** The RALM prediction accurately identifies the Canary Islands as the location of both parks. While it omits "Spain" from the reference label, this does not affect the correctness of the core answer.

Figure A.33: Samples of discrepancies (d).

---

**Samples of Discrepancies**

**Query:** Coll is known for Breacachadh also know as?
**Labels:** Breachacha Castle
**Retrieved Documents:**
1. Coll: Coll (Scottish Gaelic: "Cola" ) is an island located west of Mull in the Inner Hebrides of Scotland. Coll is known for its sandy beaches, which rise to form large sand dunes, for its corncrakes, and for Breacachadh Castle. It is in the council area of Argyll and Bute.
2. Breachacha Castle: Breachacha Castle (also spelled Breacachadh) is either of two structures on the shore of Loch Breachacha, on the Inner Hebridean island of Coll, Scotland. The earlier (also called Old Breachacha Castle) is a 15th-century tower house that was a stronghold of the Macleans of Coll, the island having been granted to John Maclean in 1431. This castle was superseded by a new dwelling in 1750 (see below) but continued to be occupied for a time, falling into a ruinous state only in the mid-19th century. Although work was performed in the 1930s to prevent further dilapidation, the castle was restored to livable condition only in the 1960s, by Major Neil V. MacLean Bristol and his wife Lavinia. It is a Category A listed building.
**RALM Prediction:** Breacachadh Castle
**Evaluation (GPT-4o, llm-em):** Incorrect. The label specifies "Breachacha Castle" as the correct answer. The student's answer, "Breacachadh Castle," contains a misspelling of the name. The correct spelling is crucial for validation purposes, and the student's answer does not match the label exactly.
**Evaluation (human, human-acc):** Correct.
**Justification for Assessment:** The RALM prediction accurately identifies "Breacachadh Castle" as the answer. Its use of the spelling "Breacachadh" aligns with the query and retrieved documents, making the answer valid and consistent with the evidence.

**Query:** Patty Hill is perhaps best known for co-writing, with her sister Mildred Hill a song that was translated into how many languages?
**Labels:** 18 languages
**Retrieved Documents:**
1. Patty Hill: Patty Smith Hill (March 27, 1868 – May 25, 1946) was a composer and teacher who is perhaps best known for co-writing, with her sister Mildred Hill, the tune which later became popular as "Happy Birthday to You". She was an American nursery school, kindergarten teacher, and key founder of the National Association for Nursery Education (NANE) which now exists as the National Association For the Education of Young Children (NAEYC).
2. Happy Birthday to You: "Happy Birthday to You", more commonly known as simply "Happy Birthday", is a song that is traditionally sung to celebrate the anniversary of a person's birth. According to the 1998 "Guinness World Records", "Happy Birthday to You" is the most recognized song in the English language, followed by "For He's a Jolly Good Fellow". The song's base lyrics have been translated into at least 18 languages. The melody of "Happy Birthday to You" comes from the song "Good Morning to All", which has traditionally been attributed to American sisters Patty and Mildred J. Hill in 1893, although the claim that the sisters composed the tune is disputed.
**RALM Prediction:** At least 18
**Evaluation (GPT-4o, llm-em):** Incorrect. The label specifies that the song was translated into "18 languages." The student's answer, "At least 18," implies a minimum number but does not precisely match the exact number provided by the label. Therefore, the student's answer does not accurately reflect the information given by the label.
**Evaluation (human, human-acc):** Correct.
**Justification for Assessment:** The RALM prediction accurately answers the query and aligns with the retrieved evidence. Its use of "at least" adds nuance without compromising correctness.

Figure A.34: Samples of discrepancies (e).

3456
3457
3458
3459
3460
3461
3462
3463
3464
3465
3466
3467
3468
3469
3470
3471
3472
3473
3474
3475
3476
3477
3478
3479
3480
3481
3482
3483
3484
3485
3486
3487
3488
3489
3490
3491
3492
3493
3494
3495
3496
3497
3498
3499
3500
3501
3502
3503
3504
3505
3506
3507
3508
3509

### Samples of Discrepancies

**Query:** How many songs were written by the man who shares attribution for one of his songs with a poem by Emily Huntington Miller?
**Labels:** 80
**Retrieved Documents:**
1. Benjamin Hanby: Benjamin Russell Hanby (July 22, 1833 – March 16, 1867), also given as Benjamin Russel Hanby, was an American composer, educator, and pastor who wrote approximately 80 songs. The most famous of his compositions are "Darling Nelly Gray", the Christmas songs "Up on the House Top", "Jolly Old Saint Nicholas", and the hymn "Who Is He In Yonder Stall?".
2. Jolly Old Saint Nicholas: "Jolly Old Saint Nicholas" is a Christmas song that originated with a poem by Emily Huntington Miller (1833-1913), published as "Lilly's Secret" in "The Little Corporal Magazine" in December 1865. The song's lyrics have also been attributed to Benjamin Hanby, who wrote a similar song in the 1860s, Up on the Housetop. However, the lyrics now in common use closely resemble Miller's 1865 poem.
**RALM Prediction:** approximately 80 songs.
**Evaluation (GPT-4o, llm-em):** Incorrect. The label specifies that the number of songs written by the man who shares attribution for one of his songs with a poem by Emily Huntington Miller is exactly 80. The student's answer states "approximately 80 songs", which is close but not an exact match to the label. The use of "approximately" introduces ambiguity that is not present in the label.
**Evaluation (human, human-acc):** Correct.
**Justification for Assessment:** The RALM prediction accurately answers the query, aligning with the evidence and reference label. Its use of "approximately" adds clarity without compromising correctness.

**Query:** The third single off Tim McGraw's twelfth studio album features which two artists?
**Labels:** Taylor Swift on vocals, as well as Keith Urban on guitar
**Retrieved Documents:**
1. Two Lanes of Freedom'; text: 'Two Lanes of Freedom is the twelfth studio album by American country music artist Tim McGraw. It was released February 5, 2013, as his first album for Big Machine Records following a 20-year tenure with Curb Records. He co-produced the album with Byron Gallimore, producer of his previously released albums. The album includes the singles "Truck Yeah", "One of Those Nights", "Highway Don't Care" featuring new labelmate Taylor Swift, and "Southern Girl".
2. "Highway Don't Care": "Highway Don't Care" is a song performed by American country music artist Tim McGraw and features American singer-songwriter Taylor Swift on vocals, as well as Keith Urban on guitar. It was released in March 2013 as the third single from McGraw's first album for Big Machine Records, "Two Lanes of Freedom" (2013). The song was written by Mark Irwin, Josh Kear and Brad and Brett Warren. McGraw and Swift recorded their parts separately.
**RALM Prediction:** Taylor Swift, Keith Urban.
**Evaluation (GPT-4o, llm-em):** Incorrect. The student's answer mentions both Taylor Swift and Keith Urban, which aligns with the label. However, the student's answer does not specify Taylor Swift's role on vocals and Keith Urban's role on guitar, which are important details present in the label.
**Evaluation (human, human-acc):** Correct.
**Justification for Assessment:** The RALM prediction accurately identifies the two featured artists, Taylor Swift and Keith Urban, as required by the query. While it omits the roles they play in the song, this does not affect the correctness of the answer.

Figure A.35: Samples of discrepancies (f).

**Samples of Discrepancies**

**Query:** How many people died in the fire Brooklyn Theatre that was owned by Sarah Crocker Conway?

**Labels:** 278 individuals

**Retrieved Documents:**

1. Sarah Crocker Conway: Sarah Crocker Conway (b. Ridgefield, Connecticut, 1834; d. Brooklyn, New York, April 28, 1875) was an actress. She was a sister of Elizabeth Crocker Bowers, also an actress. She made her debut in Baltimore, Maryland in 1849, playing Parthenia and other leading parts. In May 1852, she married the actor Frederick B. Conway, and the two thenceforward acted together. In 1859, they opened Pike's Opera House in Cincinnati, Ohio with a first-class company, but the engagement was not profitable, and they returned to the east. In 1861, they visited England, and filled a short engagement at Sadler's Wells Theatre, London. After their return, they became star actors, and made an extensive and profitable tour. In 1864 Sarah leased the Park Theatre in Brooklyn, and subsequently the new Brooklyn Theatre, in which for nine years her husband played leading parts. Sarah Conway possessed a tall and graceful figure and an expressive countenance, and was a versatile actress and a popular theatre manager.

2. Brooklyn Theatre fire: The Brooklyn Theatre Fire was a catastrophic theatre fire that broke out on the evening of December 5, 1876, in the then-city of Brooklyn (now a borough of New York City). The conflagration killed at least 278 individuals, with some accounts reporting more than 300 dead. One hundred and three unidentified victims were interred in a common grave at Green-Wood Cemetery. An obelisk near the main entrance at Fifth Avenue and 25th Street marks the burial site. More than two dozen identified victims were interred individually in separate sections at the Cemetery of the Evergreens in Brooklyn.

**RALM Prediction:** At least 278 people died in the fire.

**Evaluation (GPT-4o, llm-em):** Incorrect. The student's answer states "at least 278 people died in the fire", which suggests a minimum number but does not definitively match the label's specific count of "278 individuals". The label specifies an exact number without implying that it could be higher.

**Evaluation (human, human-acc):** Correct.

**Justification for Assessment:** The RALM prediction accurately answers the query and aligns with the evidence. Its use of "at least" appropriately reflects the documented uncertainty without altering the correctness of the response.

**Query:** Buckner Building was slightly damaged by a 9.2 earthquake that occurred on which day?

**Labels:** on Good Friday, March 27

**Retrieved Documents:**

1. Buckner Building: The Buckner Building is an abandoned former U.S. military building in Whittier, Alaska. Designed by Foss, Malcolm and Olsen architects, it was constructed by the military in 1953 as a combined mess hall, sleeping quarters, recreational, medical and administrative facility and used to be one of the largest buildings in Alaska, often being referred to as "the city under one roof". The building was slightly damaged by the 9.2 1964 Alaska earthquake, though damage was minimized by its reinforced concrete construction.

2. 1964 Alaska earthquake: The 1964 Alaskan earthquake, also known as the Great Alaskan earthquake and Good Friday earthquake, occurred at 5:36 PM AST on Good Friday, March 27. Across south-central Alaska, ground fissures, collapsing structures, and tsunamis resulting from the earthquake caused about 139 deaths.

**RALM Prediction:** March 27, 1964.

**Evaluation (GPT-4o, llm-em):** Incorrect. The label indicates that the Buckner Building was slightly damaged by a 9.2 earthquake on Good Friday, March 27. The student's answer provides the date "March 27, 1964," which is accurate in terms of the date but omits the fact that it was Good Friday. The label specifies both the date and the day (Good Friday), and the student's answer lacks the mention of Good Friday.

**Evaluation (human, human-acc):** Correct.

**Justification for Assessment:** The RALM prediction accurately provides the date of the earthquake. While it omits the additional context of "Good Friday," it satisfies the query and aligns with the retrieved evidence. The lack of this extra detail does not compromise the correctness of the answer.

Figure A.36: Samples of discrepancies (g).

