# OpenReview forum: "FRAG: Filtering Noise Using Snippet-Level Query Relevance"
_ICLR.cc/2026/Conference — Submitted to ICLR 2026_

### Official Review · Reviewer_z6uA · 2025-10-24

**Soundness:** 2
**Presentation:** 3
**Contribution:** 2
**Rating:** 4
**Confidence:** 3

**Summary:**

This paper introduces Fine-Grained RAG (FRAG), an extraction framework designed to solve the "noise" problem in Retrieval-Augmented Generation (RAG). Standard RAG performance often degrades when retrieving large amounts of information, as irrelevant content distracts the model. FRAG addresses this by operating at the snippet-level, identifying and extracting only the most relevant information from retrieved documents while filtering out noise. To handle complex, multi-hop queries, FRAG uses a component called "Self-Recognition." This module leverages historically extracted knowledge as context to preserve crucial inference-based information that might otherwise be filtered out. While FRAG significantly boosts accuracy, it introduces latency. To alleviate this, it proposes FRAG-ip, a fine-tuned framework that accelerates FRAG by approximately 10x.

**Strengths:**

This paper has the following strengths:

- It effectively establishes its theoretical grounding by demonstrating the negative impact of retrieval noise on RAG performance. This clearly motivates its core objective: filtering noise and extracting relevant information to enhance RAG accuracy.
- The core contribution is FRAG, a novel framework that introduces two key innovations: 1) snippet-level query relevance to effectively filter noise, and 2) Self-Recognition to preserve inference-based knowledge for complex, multi-hop tasks. It practically addresses the method's computational cost by proposing FRAG-ip, a fine-tuned framework that improves efficiency by approximately 10x.
- It provides extensive experimental results demonstrating that FRAG markedly boosts RAG performance, particularly on complex tasks (+13.44% gain). The results convincingly support its claims of achieving state-of-the-art performance.

**Weaknesses:**

This paper has the following weaknesses:

- The central performance claims in Table 2 are confounded by the use of different base LLMs across methods. As model capability is a significant variable, this makes it difficult to isolate the true performance gains attributable to the FRAG framework itself. A more direct comparison, where FRAG and all baselines utilize the same base LLM, would provide a much fairer and clearer assessment of its advantages.

- The paper reports strong accuracy improvements, but it lacks an analysis of statistical significance. Reporting the mean and variance over multiple experimental runs would be necessary to confirm that the observed gains are consistent and not an artifact of run-to-run variability.

- The ablation experiments, while insightful, are conducted on a single benchmark. To strengthen the paper's conclusions, it would be beneficial to replicate this study on a different benchmark (such as LongBench-v2). This would help confirm if the contributions of individual components, particularly Self-Recognition, generalize across different datasets and task structures.

- The paper introduces FRAG-ip to address the latency of FRAG, but it fails to provide a comparative cost analysis (e.g., runtime, inference latency) against the baseline methods. Without this data, it is difficult for a reader to assess the practical cost-benefit trade-off of FRAG's accuracy gains relative to standard RAG and other baselines.

- The discussion of related work appears to overlook recent and relevant studies on noise filtering and robustness in RAG (e.g., [1][2]). Situating FRAG in relation to these contemporary methods would provide a more complete picture of its novel contributions to the field.

[1] Wang, Fei, et al. "Astute rag: Overcoming imperfect retrieval augmentation and knowledge conflicts for large language models." arXiv preprint arXiv:2410.07176 (2024).

[2] Xiang, Chong, et al. "Certifiably robust rag against retrieval corruption." arXiv preprint arXiv:2405.15556 (2024).

**Questions:**

- Could the authors provide a comparison where FRAG and the key baselines are evaluated using the same base LLM to ensure a fair and direct comparison?

- To demonstrate the robustness of the results, could the authors provide the mean and standard deviation of the key performance metrics over multiple runs?

- To strengthen the claims about the framework's design, could the authors provide results from an ablation study on a different benchmark (such as LongBench-v2)? This would help confirm that the conclusions about each component's contribution generalize across different datasets.

- Could the authors provide data on the runtime and/or latency of both FRAG and FRAG-ip relative to the baselines?

---

> ### Author Response · Authors · 2025-11-28
>
> Thanks sincerely for your valuable review!
>
> > Q1: Could the authors provide a comparison where FRAG and the key baselines are evaluated using the same base LLM to ensure a fair and direct comparison?
>
> Thank you for raising this point. For key noise-reduction baselines such as *RQ-RAG, RECOMP, RankGPT, and LongRAG,* we provide additional experiments where all methods use *Qwen2-7B-Instruct* as the generator backbone to ensure a fair and direct comparison. For *ChatQA-1.5-8B and ActiveRAG*, we use *Llama3-8B-Instruct* as the shared backbone for fair comparison.
>
> For other baselines such as *HippoRAG and Search-o1*, which use models that *perform substantially better than or comparable to Qwen2-7B-Instruct (e.g., GPT-3.5 [1–3] or QWQ-32B [4])* on knowledge understanding, comprehension, and reasoning tasks, we primarily report the original results from their respective papers.
> Here are the results:
>
> ### Table 1: Overall Performance Comparison of FRAG and Baselines with Matched Backbones
> |**Models**|**PopQA**|**Pub**|**ARC**|**HotPotQA**|**2Wiki**|**MuSiQue**|**LB-v2**|
> |-|-|-|-|-|-|-|-|
> ||**ACC**|**ACC**|**ACC**|**EM/F1**|**EM/F1**|**EM/F1**|**ACC**|
> |**Baselines w/ Retrievals (BackboneLlama3-8B-Instruct)**||||||||
> |Llama3-8B-Instruct|59.39|72.14|75.64|38.8/56.62|41.8/46.62|14.9/24.07|22.27|
> |ChatQA-1.5-8B|53.75|67.17|37.99|9.9/25.74|33.8/39.8|1.3/12.18|13.12|
> |ActiveRAG-8B|46.46|32.22|46.34|23.6/25.9|29/30.75|7.5/10.54|17.89|
> |**Baselines w/ Retrievals (BackboneQwen2-7B-Instruct)**||||||||
> |Qwen2-7B-Instruct|51.82|75.08|76.83|34.9/50.38|42.2/49.57|9.2/19.52|22.66|
> |ReComp-7B|53.61|75.08|77.51|35.7/50.8|40.1/47.96|8.1/17.3|24.65|
> |RankGPT-7B|55.68|78.52|**82.28**|42.9/59.82|35.6/43.08|15.9/28.13|24.45|
> |LongRAG-7B|55.83|77.81|78.34|42.5/56.82|39.2/46.71|18.0/27.11|25.65|
> |RQ-RAG-7B|57.11|-|77.20|0*/8.8|0*/9.12|0*/7.35|27.04|
> |**Baselines w/ Retrievals (Backbone:GPT-3.5-turbo/QwQ-32B)**||||||||
> |HippoRAG|-|-|-|41.8/55|46.6/59.5|19.2/29.8|-|
> |IRCoT+HippoRAG|-|-|-|45.7/59.2|47.7/62.7|21.9/33.3|-|
> |Search-o1-32B|-|-|-|45.2/57.3|**58/71.4**|16.6/28.2|-|
> |**Ours (BackboneLlama3-8B-Instruct)**||||||||
> |FRAG-Llama3-8B-Instruct|**59.83**|75.99|77.34|41.7/58.82|32.1/38.72|22.9/33.26|24.45|
> |**Ours (Backbone: Qwen2-7B-Instruct)**||||||||
> |FRAG-Qwen2-7B-Instruct|57.97|**79.84**|82.03|**47.7/64.08**|51.5/58.5|**25.6/40.2**|**29.82**|
>
> *Note:* “-” indicates results not reported in the original papers. “*” cases where the model fails to generate as expected (cases provided in Appendix E.1.2).
>
>
> For several methods such as *SelfRAG, REPLUG, and RA-DIT,* applying them in our setting requires *additional fine-tuning with supervision signals (e.g., model-specific training data or interaction logs) that are not available for our evaluation corpora*. Consequently, we primarily report the original results from their papers and, where official checkpoints are released, we provide *partial reproductions under our retrieval setup and dataset configuration*.
>
> `[1] Qwen2 Technical Report,`
>
> `[2] Xiang L, etc.: LONGGENBENCH: Long-context Generation Benchmark`
>
> `[3] LLM Benchmarks | November 2024`
>
> `[4] Qwen2 7B Instruct vs QwQ-32B: Comprehensive side-by-side LLM comparison`
>
> > Q2: To demonstrate the robustness of the results, could the authors provide the mean and standard deviation of the key performance metrics over multiple runs?
>
> Yes, and we sincerely appreciate you for pointing out this weakness. We have evaluated FRAG over 5 runs with different random seeds:
>
> ### Table 2: Performance of FRAG-Qwen2-7B-Instruct over 5 Runs with Different Random Seeds
> |**seed**|**PopQA**|**Pub**|**ARC-C**|**HotPotQA**|**2Wiki**|**MuSiQue**|**LongBench-v2**|
> |-|-|-|-|-|-|-|-|
> ||**ACC**|**ACC**|**ACC**|**EM/F1**|**EM/F1**|**EM/F1**|**ACC**|
> |128|57.04|79.27|82.20|46.4/62.22|52.9/60.9|25.8/40.43|30.02|
> |355|57.4|79.79|81.69|46.1/61.44|52.1/61.13|24.7/39.02|29.42|
> |**633 (reported)**|57.97|79.84|82.03|47.7/64.08|51.5/58.5|25.6/40.2|29.82|
> |943|57.47|80.73|82.45|45.9/62.65|51.3/58.24|25.7/40.3|29.82|
> |1537|57.61|80.21|82.28|45.86/61.24|50.8/58.0|26.3/41.7|30.02|
> |**ave**|57.50|79.97|82.13|46.38/62.33|51.72/59.15|25.62/40.33|29.82|
> |**std**|0.34|0.54|0.29|0.77/1.14|0.81/1.28|0.58/0.95|0.24|

---

> > ### Author Response · Authors · 2025-11-28
> >
> > > Q3: To strengthen the claims about the framework's design, could the authors provide results from an ablation study on a different benchmark (such as LongBench-v2)? This would help confirm that the conclusions about each component's contribution generalize across different datasets.
> >
> > Thank you for the suggestion. We have supplemented the ablation results on LongBench-v2:
> > ### Table 3: Ablation Study of FRAG-Qwen2-7B-Instruct on LongBench-v2
> > |**Models**|**LongBench-v2**|**HotPotQA**|
> > |-|-|-|
> > ||**ACC**|**EM/F1**|
> > |Qwen2-7B-Instruct|22.66|34.9/50.38|
> > |FRAG-Qwen2-7B-Instruct|**29.82**|**47.7/64.08**|
> > |w/o KSS|28.23|42.51/58.53|
> > |w/o SR|26.84|37.08/52.71|
> > |w/o KSS+SR|25.84|35.9/51.45|
> > |w/o Validator|28.43|43.7/60.22|
> > |w/o Prefixer|29.03|46.8/61.02|
> > |w/o Deduplicator|29.42|46.6/63.3|
> > |w/o Filter|29.03|46.4/61.77|
> > |w/o Assessor|29.82|46.7/62.61|
> > |w/ Extractor only|26.24|36.8/52.42|
> >
> > *Note:* LongBench-v2 does not provide gold document labels; therefore, metrics such as noise filtering rate cannot be computed.
> >
> > These results demonstrate that *snippet-level query relevance extraction and self-recognition, along with the effectiveness of the overall FRAG modules, work similarly on LongBench-v2*.
> >
> > > Q4: Could the authors provide data on the runtime and/or latency of both FRAG and FRAG-ip relative to the baselines?
> >
> > Yes, of course:
> > ### Table 4: Latency Comparison between FRAG and Other Baselines (Time per Query in Seconds) with Matched Backbones
> > |**Models**|**T**|**T**|
> > |-|-|-|
> > ||**Singlehop: PopQA(s)**|**Multihop:HotpotQA(s)**|
> > |**Backbone: Llama3-8B-Instruct**|||
> > |Llama3-8B-Instruct|0.53|0.66|
> > |ChatQA1.5-8B|0.44|0.52|
> > |ActiveRAG-8B|66.1|100.1|
> > |**Backbone: Qwen2-7B-Instruct**|||
> > |Qwen2-7B-Instruct|0.47|0.50|
> > |ReComp-7B|1.07|1.21|
> > |RankGPT-7B|3.7|4.45|
> > |LongRAG-7B|10.64|12.71|
> > |RQ-RAG-7B|29.68|32.12|
> > |**Ours**|||
> > |FRAG-Llama3-8B-Instruct|5.13|13.87|
> > |FRAG-Qwen2-7B-Instruct|4.81|15.07|
> > |FRAG-ip-Qwen2-7B|1.02|1.07|
> >
> > If you have any further questions, please feel free to ask, and we will do our best to address them.

---

### Official Review · Reviewer_wkBi · 2025-10-28

**Soundness:** 3
**Presentation:** 3
**Contribution:** 2
**Rating:** 4
**Confidence:** 3

**Summary:**

FRAG addresses noise in RAG by doing snippet-level extraction guided by key query fragments, plus a “Self-Recognition” step that reuses previously extracted facts for multi-hop retrieval. An efficiency variant (FRAG-ip) folds several steps into the extractor to cut latency. Empirically, FRAG improves over naïve RAG and several contemporary systems on single- and multi-hop benchmarks while reducing the generation context.

**Strengths:**

The manuscript is clearly written, with a comprehensive problem decomposition supported by careful analysis and experiments.

The proposed method delivers substantial performance gains.

FRAG-ip further offers a practical route to deployment.

**Weaknesses:**

- System Complexity

The six-stage pipeline and multiple LLM invocations make the overall system heavyweight. Its multi-stage, decision-driven design makes the full FRAG pipeline conceptually closer to an agentic-style system rather than a conventional RAG pipeline. The resulting complexity and latency poses significant challenges for real-world deployment.

Although the introduction of FRAG-ip and a pipelined execution design alleviates some of the delay, the serving setup still co-hosts at least an extractor and a generator, resulting in potential inflated VRAM usage and additional orchestration overhead. However, the discussion of FRAG-ip appears mainly in the appendix and remains incomplete—there is no evaluation of peak memory consumption, concurrent throughput, or tail-latency (P90/P99) under realistic serving conditions. The primary comparison of FRAG-ip is also limited to naïve RAG; a compute- and context-matched evaluation against other RAG noise-reduction baselines would better illustrate the system’s trade-offs.

- Limited Baseline Coverage

The main results table labels many baselines as “inapplicable” without sufficient justification, which undermines the empirical validity of the reported improvements. Providing clearer reasoning or partial reproductions would strengthen the credibility of the experimental claims.

- Reproducibility

No open-source code or implementation details are provided. Given the multi-module pipeline and the separate FRAG-ip training stage, reproducing the results would be difficult without access to the exact prompts, data-processing scripts, evaluation harness, retriever configurations, and corpus versions used in the experiments.

**Questions:**

Please refer Weaknesses. Moreover, in the FRAG-ip experiments, what TG values were used? Do the same TG-dependent trends observed in FRAG also appear in FRAG-ip?

---

> ### Author Response · Authors · 2025-11-28
>
> Thanks sincerely for your valuable review!
>
> > W1: System Complexity: The six-stage pipeline ... The primary comparison of FRAG-ip is also limited to naïve RAG; a compute- and context-matched evaluation against other RAG noise-reduction baselines would better illustrate the system’s trade-offs..
>
> - (1) We acknowledge that additional latency is the main limitation of FRAG (Section 3.3), which is precisely why we propose **FRAG-ip** to accelerate the pipeline for practical use. For real-world deployment, as demonstrated in our paper, we can use *the same model as both the extractor and the generator*—for example, Qwen2-7B-Instruct. Even when using different models, since we *only load one model into VRAM at a time to process a batch of 500/1000 queries for extraction or generation* (as done in our experiments), the VRAM load remains manageable.
>
> - (2) We fully agree—and sincerely appreciate the suggestion—that we should report metrics such as peak memory consumption, concurrent throughput, and latency (Noting that we use vLLM for inference and *vLLM pre-allocates the GPU memory* specified by hyperparameter *gpu_memory_utilization* at startup):
>
> ### Table 1: Computational Cost of FRAG
> |**Models**|**Task**|**Peak Memory Consumption**|**Concurrent Throughput**|**Average Latency**|
> |-|-|-|-|-|
> || |**GiB**|**queries/min**|**Second**|
> |**on RTX4090-24GiB**| | | | |
> |FRAG-Qwen2-7B-Instruct|PopQA|20.10|14.12|4.25|
> | |HotPotQA|21.17|4.49|13.37|
> |FRAG-ip-Qwen2-7B|PopQA|19.23|60.61|0.99|
> | |HotPotQA|20.83|59.41|1.01|
> |**on A100-40GiB**| | | | |
> |FRAG-Qwen2-7B-Instruct|PopQA|37.55|12.47|4.81|
> | |HotPotQA|38.87|3.98|15.07|
> |FRAG-ip-Qwen2-7B|PopQA|37.55|58.82|1.02|
> | |HotPotQA|37.98|56.07|1.07|
>
> *Note:* The *extraction batch size* is 500, and we set *vllm_gpu_memory_utilization=0.95*; Due to vLLM’s dynamic batching mechanism, individual queries are processed in overlapping batches, making it impractical to measure precise per-query latency. Thus, we report average latency instead.
>
> - (3) Moreover, thank you for highlighting the need for *compute- and context-matched comparisons*. We have conducted additional experiments in which *FRAG-ip is compared against compression- and noise-reduction baselines (RQ-RAG, LongRAG, RECOMP, and RankGPT) under the same backbone (Qwen2-7B)*, identical retrieval setup (Contriever), and matched context budgets (top-k = 20). We further control for compute by *targeting similar average latency per query* (≤ 2s/q for preprocessing and ≤ 0.6s/q for generation).
> ### Table 2: Performance Comparison between FRAG-ip and Other Baselines Under Matched Compute and Context Budgets
> |**Models**|**PopQA**|**Pub**|**ARC**|**HotPotQA**|**2Wiki**|**MuSiQue**|**LB-v2**|
> |-|-|-|-|-|-|-|-|
> ||**ACC**|**ACC**|**ACC**|**EM/F1**|**EM/F1**|**EM/F1|ACC**|
> |**Baselines w/ Retrievals (Backbone Qwen2-7B-Instruct)**||||||||
> |Qwen2-7B-Instruct|51.82|75.08|76.83|34.9/50.38|42.2/49.57|9.2/19.52|22.66|
> |ReComp-7B|53.61|75.08|77.51|35.7/50.8|40.1/47.96|8.1/17.3|24.65|
> |RankGPT-7B|**55.61**|78.52|81.26|**38.3**/53.14|37.6/45.43|12.2/24.37|23.26|
> |LongRAG-7B|52.89|75.48|77.51|38.1/**53.88**|39.2/46.71|12.1/22.95|23.26|
> |RQ-RAG-7B|52.25|-|77.09|34.4/49.37|41.5/48.92|8.6/18.74|22.66|
> |FRAG-ip|53.11|**79.43**|**81.60**|37.2/49.69|**52.1/57.45**|**13.6/25.26**|**25.84**|
>
> *Note:* If a baseline cannot finish preprocessing within 2s/q, *the remaining queries fall back to naïve RAG for generation.* “–” indicates that the model fails to generate as expected.
>
> It can be demonstrated that *FRAG-ip consistently delivers higher accuracy in most cases under comparable latency*, clarifying its accuracy–efficiency trade-offs. We will integrate these new results into the revised version.

---

> ### Author Response · Authors · 2025-11-28
>
> > W2: Limited Baseline Coverage: The main results table labels many baselines as “inapplicable” without sufficient justification, which undermines the empirical validity of the reported improvements. Providing clearer reasoning or partial reproductions would strengthen the credibility of the experimental claims.
>
> Thank you for raising this point. For several methods such as *SelfRAG and RA-DIT*, *applying them in our setting requires additional fine-tuning with supervision signals (e.g., model-specific training data or interaction logs) that are not available on our evaluation corpora*. Consequently, we primarily report the original numbers from their papers and, where official checkpoints are released, we provide *partial reproductions under our retrieval and dataset configuration*.
>
> Regarding metric *LLM-EM*, its computation requires substantial additional calls to a strong proprietary model, which makes recomputing this metric for all baselines across all datasets prohibitive under our resource budget. We therefore use **EM/F1** as the universal primary metric for broad baseline coverage, and employ LLM-EM mainly to analyze the relative gain of FRAG over the key baseline of naïve RAG (and a small set of core comparisons) on open-domain datasets. Importantly, *in the settings where we do compute LLM-EM, its trend is consistent with EM/F1, suggesting that the EM/F1-based comparisons are representative.*
>
> ### Table 3: Overall Performance Comparison of FRAG and Baselines
> |**Models**|**PopQA**|**Pub**|**ARC-C**|**HotPotQA**|**2Wiki**|**MuSiQue**|**LongBench-v2**|
> |-|-|-|-|-|-|-|-|
> ||**ACC**|**ACC**|**ACC**|**EM/F1**|**EM/F1**|**EM/F1**|**ACC**|
> |**Baselines w/ Retrievals**||||||||
> |SelfRAG-7B|54.9|72.4|67.3|12.9/29.17|16.8/27.57|1.2/12.39|22.86|
> |SelfRAG-13B|55.8|74.5|73.1|13.2/19.26|6.2/21.06|1.5/12.19|2.58|
> |ReComp-7B|53.61|75.08|77.51|35.7/50.8|40.1/47.96|8.1/17.3|24.65|
> |RankGPT-7B|55.68|78.52|82.28|42.9/59.82|35.6/43.08|15.9/28.13|24.45|
> |RQ-RAG-7B|57.1|*|68.3|0*/7.9|0*/8.84|0*/7.23|27.04|
> |LongRAG-6B|-|-|-|40.5/53.09|37.5/44.52|17.5/25.88|-|
> |ActiveRAG-8B|46.46|32.22|46.34|23.6/25.9|29/30.75|7.5/10.54|17.89|
> |HippoRAG|-|-|-|41.8/55|46.6/59.5|19.2/29.8|-|
> |IRCoT+HippoRAG|-|-|-|45.7/59.2|47.7/62.7|21.9/33.3|-|
> |Search-o1-32B|-|-|-|45.2/57.3|**58/71.4**|16.6/28.2|-|
> |ChatQA-1.5-8B|53.75|67.17|37.99|9.9/25.74|33.8/39.8|1.3/12.18|13.12|
> |Llama3-8B-Instruct|59.39|72.14|75.64|38.8/56.62|41.8/46.62|14.9/24.07|22.27|
> |Qwen2-7B-Instruct|51.82|75.08|76.83|34.9/50.38|42.2/49.57|9.2/19.52|22.66|
> |ChatQA-1.5-70B|58.97|72.14|74.11|26.8/43.41|21.9/30.33|8/21.44|24.06|
> |**Ours**||||||||
> |FRAG-ChatQA-1.5-8B|54.11|73.66|46.42|27.9/44.8|44.2/47.96|12.4/23.75|21.47|
> |FRAG-Llama3-8B-Instruct|**59.83**|75.99|77.34|41.7/58.82|32.1/38.72|22.9/33.26|24.45|
> |FRAG-Qwen2-7B-Instruct|57.97|**79.84**|82.03|**47.7/64.08**|51.5/58.5|**25.6/40.2**|**29.82**|
> |FRAG-ChatQA-1.5-70B|59.69|76.49|**82.79**|38.4/53.92|28.4/36.66|19.3/32.3|29.03|
>
> *Note:* “-” indicates results not reported in the original papers. “*” cases where the model fails to generate as expected (cases provided in Appendix E.1.2).
>
> We agree, however, that *labeling some methods as “inapplicable” in the main table without sufficient explanation was unclear*. In the revised version, we:
> - (i) *add a detailed table* in the appendix that, for each such baseline, specifies the concrete reason why it is not reported in our experimental setup; and
>
> - (ii) *supplement EM/F1 results comparing FRAG against the most relevant compression / noise-reduction baselines* wherever a reasonable reproduction is possible. We believe these additions clarify our experimental scope and strengthen the empirical credibility of our claims.
>
>
> > W3: Reproducibility: No open-source code or implementation details are provided. Given the multi-module pipeline and the separate FRAG-ip training stage, reproducing the results would be difficult without access to the exact prompts, data-processing scripts, evaluation harness, retriever configurations, and corpus versions used in the experiments.
>
> Thanks for the reminder! We have released our code on GitHub. Please refer to the public comment for link.

---

> > ### Author Response · Authors · 2025-11-28
> >
> > > Q: Moreover, in the FRAG-ip experiments, what TG values were used? Do the same TG-dependent trends observed in FRAG also appear in FRAG-ip?
> >
> > Yes. *Similar TG-dependent trends observed in FRAG also appear in FRAG-ip*.
> > In FRAG-ip, we fine-tune the extractor model to simply score the extracted relevant information with 0/0.5/1. We report the performance under different TG values on the three multi-hop datasets.
> > ### Table 4: FRAG-ip Performance under Different TG Values
> > |**Models**|**HotpotQA**|**2Wiki**|**Musique**|
> > |-|-|-|-|
> > |**FRAG-ip (batch t=1)**|**EM/F1**|**EM/F1**|**EM/F1**|
> > |TG=0|36.4/49.34|50.8/56.08|13.6/24.39|
> > |TG=0.1|37.2/**49.69**|**52.1/57.45**|**14/25.65**|
> > |TG=0.5|**37.5**/49.67|48.4/54.12|13.3/24.76|
> > |TG=1|36.3/49.31|48.12/53.87|12.7/23.83|
> >
> > The results show that the RALM achieves its best performance at TG = 0.1 (we set this value to accommodate cases where the model does not strictly follow the intended scoring order), which *preserves most of the relevant information while filtering out the majority of noise*. When TG = 0, more noise remains and accuracy drops. When TG = 0.5/1, accuracy also decreases due to excessive loss of relevant information.
> > Therefore, we recommend TG = 0.1 for practical use.
> >
> > If you have any further questions, please feel free to ask, and we will do our best to address them.

---

### Official Review · Reviewer_Bx9J · 2025-10-31

**Soundness:** 4
**Presentation:** 3
**Contribution:** 2
**Rating:** 6
**Confidence:** 4

**Summary:**

This paper talks about FRAG, a method for making Retrieval-Augmented Generation (RAG) less noisy and more accurate. When you make the retrieval window bigger, you get more relevant stuff, but you also get more junk (noise), which really throws off large language models. FRAG tries to fix this by picking out key snippets from queries and filtering everything at a super fine-grained (snippet-level) scale. For harder queries that need multi-hop logic, the paper adds Self-Recognition, which uses what’s already been extracted to keep inference going. There’s also a faster FRAG-ip framework to make things run quicker. In experiments, FRAG improves average accuracy 5-13 points over naive RAG and does a solid job on a bunch of standard datasets

The theoretical motivation matches what many have found in RAG research: more retrieval isn't always better, since noise quickly drowns out the good stuff. The snippet-level filter is well-motivated, and the experiments provide solid evidence. there are clear comparisons against standard and reranked RAG baselines. The handling of multi-hop/inference-style queries via Self-Recognition is clever and nicely shown in ablation studies. The results and methodology seem robust, though the paper's math is a bit dense.

This work addresses a very active problem in RAG for LLMs, getting high relevance with low noise. The move to snippet-level sorting  seems like a natural next step, and integrating Self-Recognition for multi-hop queries is a clever touch. The FRAG-ip speedup shows some attention to practical deployment. These results are important for anyone working on grounded QA, multi-turn dialogue, or scientific/medical data retrieval.

**Strengths:**

Snippet-level relevance filtering is a meaningful advance over existing sentenct level approaches.Empirical results demonstrate strong gains in both standard and challenging retrieval nd QA tasks.Handles multi-hop and complex queries better than other approaches using the Self-Recognition.Provides clear ablation studies and baseline comparisons.Addresses both accuracy and speed in RAG settings (via FRAG-ip).

**Weaknesses:**

Despite FRAG-ip, snippet-extraction adds LLM calls that may still slow things down for very large, complex queries.  Paper is packed with jargon and math, which may hurt accessibility for wider audiences.Most benchmarks are synthetic or curated; I'd like to see more real-world open domain or industry use cases.Relies on strong instruction-following LLM, uncertain results with simpler models. I think the core concept (snippet-level filtering in RAG and multi-hop self-awareness) isn’t fully original. major ideas are also present in FineFilter, SCMRAG, SIM-RAG, and related literature. Authors need to cite these works, clarify their unique contributions, and openly discuss overlap. That said, the specific pipeline (Self-Recognition + FRAG-ip) and some implementation details add value beyond simple reproduction of those other concepts.

**Questions:**

Can FRAG be extended to noisy, open-source web data (not just synthetic/curated datasets)? Any ides on lighter-weight snippet relevance scoring to reduce inference cost? How resilient is FRAG to misleading/snippet-level adversarial noise? Have you tested FRAG in a conversational (not just QA) setting, e.g., for dialogue or summarization? i can see it is bit different but it would be good if you could explicitly explain whats actually new in your approach versus FineFilter and SCMRAG?

---

> ### Author Response · Authors · 2025-11-28
>
> Thanks sincerely for your valuable review!
>
> > W1: Despite FRAG-ip, snippet-extraction adds LLM calls that may still slow things down for very large, complex queries.
>
> Yes, similar to many other RAG variants, FRAG/FRAG-ip introduce additional latency to enhance RALM performance, and may slow generation on very large or complex tasks. However, we believe that:
>
> - On one hand, FRAG has unique value for tasks requiring higher accuracy.
>
> - On the other hand, as LLM capabilities continue to improve, we can further reduce latency by decreasing the number of extraction steps and skipping validation modules such as the Verifier and Deduplicator.
>
> Moreover, we are experimenting with an improved variant that uses reinforcement learning to enable the extractor model to identify relevant information more effectively within a limited number of extraction steps. Please see the details in our response to Q2.
>
> > Q1: Can FRAG be extended to noisy, open-source web data (not just synthetic/curated datasets)?
>
> Yes, of course. We evaluate FRAG in *four noisy, open-source web datasets*:
>
> - **(1) Quasar-T:** built from noisy, open-web data. Trivia-style questions are paired with candidate passages retrieved from ClueWeb09, a real-world web crawl full of uncurated, imperfect pages (ads, duplicates, inconsistent formatting, etc.).
>
> `Bhuwan D, etc.: Quasar: Datasets for Question Answering by Search and Reading.`
>
> - **(2) SearchQA:** built to mimic a full “search then read” pipeline: Jeopardy! -style question–answer pairs from J!  Archive are automatically augmented with noisy context snippets retrieved from a commercial web search engine (Google).
>
> `Matthew D, etc.: SearchQA: A New Q&A Dataset Augmented with Context from a Search Engine.`
>
> - **(3) TriviaQA:** trivia-enthusiast questions are paired with evidence documents gathered by crawling the broader web (beyond Wikipedia), providing distantly supervised question–answer–evidence triples over noisy, heterogeneous web pages.
>
> `Mandar J, etc.: TriviaQA: A Large Scale Distantly Supervised Challenge Dataset for Reading Comprehension.`
>
> - **(4) Natural Questions (NQ):** real Google search queries which are highly noisy, natural, and often ambiguous, reflecting real user behavior. Information are collected from Wikipedia.
>
> `Tom K, etc.: Natural Questions: A Benchmark for Question Answering Research.`
>
> Here are the results:
> ### Table 1: FRAG Performance on Noisy, Open-Web Datasets
> |**Models**|**SearchQA**|**Quasar-T**|**TriviaQA**|**NQ**|
> |-|-|-|-|-|
> ||**EM/F1**|**EM/F1**|**ACC**|**EM/F1**|
> |Qwen2-7B-Instruct|26/38.18|33/46.75|61|31.9/42.8|
> |FRAG-Qwen2-7B-Instruct|**37.5/46.99**|**37/49.47**|**64.5**|**40.9/50.3**|
>
> It can be demonstrated that *FRAG extends well to noisy web datasets—whether the noise appears in the query or in the corpus—by effectively identifying useful information and filtering out noise*.
>
>
> > Q2: Any ides on lighter-weight snippet relevance scoring to reduce inference cost?
>
> Yes, we are developing a new variant of FRAG that *leverages reinforcement learning to encourage the model to extract relevant information and accurately score snippet relevance within a limited number of extraction steps. The model is rewarded for responses that capture as many relevant sentences as possible using fewer tokens and fewer steps, while also correctly ranking these sentences*. We believe this approach will significantly reduce the number of LLM calls and the overall inference cost. Please look forward to our results!
>
>
> >Q3: How resilient is FRAG to misleading/snippet-level adversarial noise?
>
> From the results provided in the response to Q1, it can be seen that FRAG demonstrates robustness on noisy web data. FRAG also shows resilience to certain types of obvious misleading noise, such as:
> ```text
> “Lyon replaced Paris as the capital of France at 2008.”
> ```
> However, since FRAG is designed to extract information from and remain faithful to the original documents, it may not be highly robust against carefully crafted adversarial noise.
>
> Given this, an additional adversarial-resistant fact-validation module can be introduced in scenarios where users may be at risk of adversarial attacks.

---

> > ### Author Response · Authors · 2025-11-28
> >
> > >Q4: Have you tested FRAG in a conversational (not just QA) setting, e.g., for dialogue or summarization?
> >
> > No. However, we additionally evaluate FRAG on *two conversational settings*:
> >
> > - **(1) QMSum:** a benchmark for *query-based, multi-domain meeting summarization* for handling long, multi-topic meetings. It contains 1808 query–summary pairs over 232 long meetings from academic, product, and committee settings, where models must first locate query-relevant spans in the transcript and then generate concise summaries of those spans.
> >
> > `Ming Z, etc.: QMSum: A New Benchmark for Query-based Multi-domain Meeting Summarization.`
> >
> > - **(2) LexRAG:** a benchmark for RAG in *multi-turn legal consultation*, built from 1013 expert-annotated dialogue sessions and 17228 candidate legal articles across multiple law domains.  Each conversation contains five rounds of progressively refined questions, with associated gold legal articles and answers, enabling joint evaluation of conversational legal retrieval and legally grounded response generation.
> >
> > `Haitao L, etc.: LexRAG: Benchmarking Retrieval-Augmented Generation in Multi-Turn Legal Consultation Conversation.`
> >
> > Here are the results:
> > ### Table 2: FRAG Performance on Conservation Datasets
> > |**Models**|**QMSum**| | |**LexRAG**|| | | |
> > |-|-|-|-|-|-|-|-|-|
> > ||**ROUGE-1**|**ROUGE-2**|**ROUGE-L**|**KW-ACC**|**ROUGE-1**|**ROUGE-2**|**ROUGE-L**|
> > |Qwen2-7B-Instruct|22.19|5.24|14.75|36.26|5.53|2.62|5.21|
> > |FRAG-Qwen2-7B-Instruct|**28.67**|**7.92**|**18.46**|**38.04**|**6.78**|**2.78**|**6.65**|
> >
> > *Note:* KW-ACC refers to keyword Hit rate.
> >
> > > Q5: I can see it is bit different but it would be good if you could explicitly explain whats actually new in your approach versus FineFilter and SCMRAG?
> >
> > We also note that *FineFilter, SCMRAG,* and our work FRAG were developed roughly concurrently and independently. FineFilter and SCMRAG are strong baselines for enhancing generation quality in RAG, and in the revised version we will explicitly mark them as concurrent work and further clarify the relationship between these methods and FRAG in the related work section.
> >
> > ### New w/ FineFilter:
> >
> > We appreciate FineFilter/CompSelect as a strong fine-grained noise-filtering baseline and agree that it targets a similar high-level goal. Technically, however, our setting and contributions are different in several ways.
> > - (1) FineFilter formulates noise reduction as sentence-level MinMax clue selection, whereas FRAG performs fine-grained extraction explicitly guided by the query’s key snippets and provides attention-based theoretical analysis (Lemma 3.1 / Theorem 3.2) showing how such key-query-snippet modeling mitigates attention drift — an aspect not modeled in FineFilter.
> >
> > - (2) FineFilter uses *the full query* as a black-box condition, while FRAG first decomposes the query into key vs. non-key snippets and then conditions extraction and *filtering on the key snippets*, which drives the whole pipeline.
> >
> > - (3) For complex, multi-hop QA, FineFilter mainly relies on *training objectives* using generator feedback, whereas FRAG introduces a dedicated *Self-Recognition mechanism* that conditions the extraction of new relevant information on previously extracted knowledge, explicitly targeting inference-based, multi-hop reasoning; ablations show this component is crucial on multi-hop benchmarks.
> >
> > - (4) Architecturally, FineFilter adopts a three-module MinMax framework, while FRAG proposes a six-module LLM-based extraction framework and further accelerate it by proposing a wrapper framework (FRAG-ip) that reduce extraction steps and skipping validating LLM calls, yielding both accuracy and efficiency gains.

---

> > > ### Author Response · Authors · 2025-11-28
> > >
> > > ### New w/ SCMRAG:
> > >
> > > We also carefully compared FRAG to SCMRAG. Conceptually, SCMRAG targets a different part of the RAG pipeline: it introduces *an LLM-assisted dynamic knowledge graph and a self-corrective agent loop* that detects when the current graph is incomplete or outdated and then fetches missing information from external sources for multi-hop questions.
> > >
> > > In contrast, FRAG assumes a given retriever and focuses on *the fine-grained noise filtering problem inside the retrieved context*. Technically, our contributions differ in three aspects.
> > > - (1) FRAG explicitly models *key-snippet–based query relevance*: we first decompose the query into key vs. non-key snippets, then use the key snippets to drive *snippet-level extraction and filtering*, and provide attention-based theoretical analysis showing how this query-snippet modeling mitigates attention drift.
> > >
> > > - (2) For multi-hop reasoning, SCMRAG’s self-corrective mechanism operates at the retrieval/KB level (updating the graph), whereas FRAG introduces a *Self-Recognition method* at the snippet-extraction level that *conditions new snippet extraction on historically extracted knowledge to preserve inference-based multi-hop evidence*.
> > >
> > > - (3) Architecturally, FRAG proposes a six-module LLM-based extraction framework and further accelerates it via FRAG-ip, a wrapper framework that reduces extraction steps and skips validating LLM calls, yielding both accuracy and efficiency gains. These contributions are thus new from SCMRAG’s dynamic-graph, self-corrective retrieval paradigm.
> > >
> > >
> > > If you have any further questions, please feel free to ask, and we will do our best to address them.

---

### Official Review · Reviewer_K9XM · 2025-11-01

**Soundness:** 1
**Presentation:** 2
**Contribution:** 2
**Rating:** 2
**Confidence:** 5

**Summary:**

This work addresses the fine-grained (snippet-level) associations between the query and the relevant documents in RAG (retrieval-augmented generation). Different from conventional RAG, the proposed FRAG framework extracts key sequential snippets S from the query, and then extracts relevant knowledge K from relevant documents according to S. This approach greatly condenses the context of RAG. In experiments, FRAG consistently improves the RAG performances on four LLMs (i.e., ChatQA-1.5-8B, Llama-3-8B-Instruct, Qwen-2-7B-Instruct, and ChatQA-1.5-70B) on 10 datasets.

**Strengths:**

* Experimental results show consistent improvements contributed by the proposed method on four LLMs on 10 datasets.

* The method greatly reduces the length of context information during RAG.

**Weaknesses:**

* While the proposed FRAG method improves the RAG performances of LLMs in vanilla RAG, the comparison with other context-condensing approaches should be comprehensively analyzed in addition to the statements in B5. In this work, no empirical results with RankRAG are shown, and the results of RankGPT were based on a different LLM (old Llama-7B). Without the comparison with these real baseline models, the benefit of the proposed method is still unclear.

* This method adds computational cost during the inference stage. The overhead could be measured and compared with other methods.

**Questions:**

* B.5 compares FRAG with reranking methods conceptionally. Can you provide empirical comparisons with those reranking methods?

---

> ### Author Response · Authors · 2025-11-28
>
> Thanks sincerely for your valuable review! However, we would like to clarify a misunderstanding in the comments:
>
> > W1: ... In this work, no empirical results with RankRAG are shown, and the results of RankGPT were based on a different LLM (old Llama-7B). Without the comparison with these real baseline models, the benefit of the proposed method is still unclear.
>
> - **Comparison with RankRAG:** It appears that RankRAG has not yet open-sourced any models or code on HuggingFace or GitHub. For comparison, we report the results on PopQA, HotpotQA and 2Wiki directly from their paper:
>
> ### Table 1: Comparison on of FRAG with RankRAG
>
> |Models|PopQA|HotPotQA|2Wiki|
> |-|-|-|-|
> ||**ACC**|**EM/F1**|**EM/F1**|
> |**Baselinesw/retrievals**||||
> |Llama3-RankRAG8B|64.1|35.3/46.7|31.4/36.9|
> |Llama3-RankRAG70B|**65.4**|42.7/55.4|38.2/43.9|
> |FRAG-Qwen2-7B-Instruct|57.97|**47.7/64.08**|**51.5/58.5**|
>
> It can be seen that FRAG-Qwen2 achieves clearly stronger performance on the two multi-hop tasks (HotpotQA and 2Wiki), even with a smaller model size, although it performs worse on the single-hop task (PopQA).
>
> - **Comparison with RankGPT:** As clearly stated in Appendix D.2: “For the reranking baseline RankGPT (Sun et al., 2023), we employ Qwen2-7B-Instruct as the generator to ensure a fair comparison.” Please kindly refer to this section. In fact, we use *the same generator (Qwen2-7B-Instruct) for both FRAG and RankGPT*, rather than using Llama-7B for RankGPT.
>
> - We also compare FRAG with other representative baselines, such as *Recomp-20B*, and FRAG still demonstrates clear advantages.
>
> > W2: This method adds computational cost during the inference stage. The overhead could be measured and compared with other methods.
>
> *We have provided computational cost with other baselines in Appendix B.4, Table A.7.* Please kindly refer to this section. Moreover, we have tested the computational cost of more baselines, here are the results:
>
> ### Table 2: Latency Comparison between FRAG and Other Baselines (Time per Query in Seconds) with Matched Backbones as Generators
> |**Models**|**T**|**T**|
> |-|-|-|
> ||**Singlehop: PopQA(s)**|**Multihop:HotpotQA(s)**|
> |**Backbone: Llama3-8B-Instruct**|||
> |Llama3-8B-Instruct|0.53|0.66|
> |ChatQA1.5-8B|0.44|0.52|
> |ActiveRAG-8B|66.1|100.1|
> |**Backbone: Qwen2-7B-Instruct**|||
> |Qwen2-7B-Instruct|0.47|0.50|
> |ReComp-7B|1.07|1.21|
> |RankGPT-7B|3.7|4.45|
> |LongRAG-7B|10.64|12.71|
> |RQ-RAG-7B|29.68|32.12|
> |**Ours**|||
> |FRAG-Llama3-8B-Instruct|5.13|13.87|
> |FRAG-Qwen2-7B-Instruct|4.81|15.07|
> |FRAG-ip-Qwen2-7B|1.02|1.07|
>
> > Q: B.5 compares FRAG with reranking methods conceptionally. Can you provide empirical comparisons with those reranking methods?
>
> Reranking is a practical approach for ordering retrieved passages so that more relevant ones appear at the top. However, its limitations are evident:
>
> - It cannot address intra-passage noise or extract only the relevant portions.
> - It is difficult to choose a hyperparameter k for selecting the top relevant passages for generation—especially for multi-hop tasks. Selecting *a small k leads to information loss*, while
> *a large k introduces excessive noise*, making the model more vulnerable to noise. The comparison below illustrates this issue:
>
> ### Table 3: Performance Comparison between FRAG and RankGPT
> |**Models**|**PopQA**|**Pub**|**ARC-C**|**HotPotQA**|**2Wiki**|**MuSiQue**|**LongBench-v2**|
> |-|-|-|-|-|-|-|-|
> ||**ACC**|**ACC**|**ACC**|**EM/F1**|**EM/F1**|**EM/F1**|**ACC**|
> |RankGPT-Qwen2@5|55.68|78.52|**82.28**|42.9/59.82|35.6/43.08|15.9/28.13|24.45|
> |RankGPT-Qwen2@10|57.76|77.81|80.58|40.7/58.43|39.2/47.33|17.7/29.6|19.88|
> |RankGPT-Qwen2@20|56.4|78.72|77.51|-|-|13.8/26.34|23.26|
> |FRAG-Qwen2-7B-Instruct|**57.97**|**79.84**|82.03|**47.7/64.08**|**51.5/58.5**|**25.6/40.2**|**29.82**|
>
> *Note*: we use GPT-3.5-turbo as the reranker of RankGPT.
>
> Moreover, the unstable reranking quality of the reranker may even *push relevant information to lower positions, making it more difficult to select an appropriate k*, as demonstrated by the results on Pub and LongBench-v2.
>
> If you have any further questions, please feel free to ask, and we will do our best to address them.

---

### Official Review · Reviewer_dXwT · 2025-11-01

**Soundness:** 3
**Presentation:** 3
**Contribution:** 3
**Rating:** 6
**Confidence:** 3

**Summary:**

In the Retrieval-Augmented Language Model (RALM), expanding the retrieval window can enhance performance by including more relevant knowledge. However, it also introduces noise, which may degrade RALM performance. To mitigate the problem, the authors propose Fine-Grained RAG (FRAG), which performs fine-grained extraction of relevant knowledge and filtering noise from initial retrievals. Furthermore, to alleviate the attention distraction caused by noise, the authors decompose the extraction into some steps. Moreover, to deal with omitted prerequisite knowledge caused by the decomposition, the authors propose Self-Recognition, which leverages
historically extracted knowledge as a reference in extraction to restore missing logical relationships. In addition to the above performance-specific proposals, the authors also propose FRAG-ip, a wrapper framework that employs dual-stage fine-tuning and accelerates FRAG. In the experiments, FRAG actually improves RALM performance on both simple and complex tasks.

**Strengths:**

- This work aims to solve the issues in the fundamental field of the current large language models (LLMs), the Retrieval-Augmented Language Model (RALM).
- This work theoretically demonstrates the negative impact of noise on RALM.
- The proposed method, FRAG, can deal with the noise issue in RALM.
- In addition to FRAG, the authors propose FRAG-ip, an efficient framework for running FRAG.
- The experiments cover various kinds of LLMs, and the settings with and without RAG baselines are also covered.
- As well as using single-hop QA, the conducted experiments employ multi-hop QA, which is considered a complex task.
- The experimental results show the promising performance gains of FARG in multiple tasks.
- The ablation study actually shows the importance of each component.

**Weaknesses:**

- The used extractor models are restricted to instruction-based models. The commonly used extractors, like BERT and its variants, and BM25-based methods are not covered.

**Questions:**

- Why did you choose Llama3-8B-Instruct and Qwen2-7B-Instruct for the extractor models in the experiments? Is there any reason for not using commonly used extractors such as BERT, BM25, etc.?

---

> ### Author Response · Authors · 2025-11-28
>
> Thanks sincerely for your valuable review!
>
> > Q: Why did you choose Llama3-8B-Instruct and Qwen2-7B-Instruct for the extractor models in the experiments? Is there any reason for not using commonly used extractors such as BERT, BM25, etc.?
>
> It is important to clarify why we choose *Instruct* models for extracting relevant information and reducing noise.
>
> ## Purpose of FRAG
>
> The goal of FRAG is to *reduce noise* while *retaining as much relevant information as possible*—both are essential for improving the correctness of RALM. In other words, if noise is not filtered effectively, or if the filtering process removes too much relevant information, the performance of RALM will degrade.
>
> ## (1) BERT/BM25 do not fit noise-filtering / relevant-information-extraction tasks well
>
> We experimented with **BERT/ColBert/BM25** as extractors. However, because both approaches are naturally unsuited to next-word-prediction tasks, we adopted a strategy that *splits retrieved passages into individual sentences* and *scores the relevance between the query and each sentence*. The extracted samples and experimental results are as follows:
>
> ```text
> Query: "What is Bruce McDaniel’s occupation?"
>
> ### Bert-Large:
> Passages:
> 1. “**[Gold ]Bruce McDaniel: Bruce McDaniel is an American musician, composer, producer and recording engineer, currently living in New Orleans [s=0.6616].** Bruce McDaniel was born in Boston, Massachusetts of Mexican and Scottish/American parents on 23 September 1962 and grew up in New York [s=-0.8173]. He was raised by musical parents who met while attending the Juilliard School of Music [s=0.9357]. ...”
> 2. [Noise] “John McDaniel (born September 23, 1951 in Birmingham, Alabama) is a former American football wide receiver [s=-0.8538]. ...”
> 3. [Noise] “Jerry McDaniel: McDaniel has also conceived and produced short films and film titles [s=0.9548]. ...”
> 4. [Noise] “Bruce Smith: Bruce Smith was born in New York. Bruce's occupation is a boxer [s=0.7666]. ...”
>
> ### ColBert-v2.0:
> Passages:
> 1. “**[Gold ]Bruce McDaniel: Bruce McDaniel is an American musician, composer, producer and recording engineer, currently living in New Orleans [s=0.9479]. **Bruce McDaniel was born in Boston, Massachusetts of Mexican and Scottish/American parents on 23 September 1962 and grew up in New York [s=0.9696]. He was raised by musical parents who met while attending the Juilliard School of Music [s=0.9670]. ...”
> 2. [Noise] “John McDaniel: John McDaniel (born September 23, 1951 in Birmingham, Alabama) is a former American football wide receiver [s=0.9489]. ...”
> 3. [Noise] “Jerry McDaniel: McDaniel has also conceived and produced short films and film titles [s=0.9627]. ...”
> 4. [Noise] “Bruce Smith: Bruce Smith was born in New York. Bruce's occupation is a boxer [s=0.9806]. ...”
>
> ### BM25:
> Passages:
> 1. “**[Gold ]Bruce McDaniel: Bruce McDaniel is an American musician, composer, producer and recording engineer, currently living in New Orleans [s=0.2613]. **Bruce McDaniel was born in Boston, Massachusetts of Mexican and Scottish/American parents on 23 September 1962 and grew up in New York [s=0.2187]. He was raised by musical parents who met while attending the Juilliard School of Music [s=0]. ...”
> 2. [Noise] “John McDaniel: John McDaniel (born September 23, 1951 in Birmingham, Alabama) is a former American football wide receiver [s=0.2613]. ...”
> 3. [Noise] “Jerry McDaniel: McDaniel has also conceived and produced short films and film titles [s=0.3958]. ...”
> 4. [Noise] “Bruce Smith: Bruce Smith was born in New York. Bruce's occupation is a boxer [s=2.6977]. ...”
> ```
> *Note:* \**[Gold] ...\** refers to the gold evidence of the query.
>
> ### Table: FRAG Performance Comparison Using Instruct LLM Extractor Models vs. BERT/BM25 Extractor Models
> |**Extractor**|**Generator**|**PopQA**|**Pub**|**ARC**|**HotPotQA**|**2Wiki**|**MuSiQue**|**LongBenchv2**|
> |-|-|-|-|-|-|-|-|-|
> |||**ACC**|**ACC**|**ACC**|**EM/F1**|**EM/F1**|**EM/F1**|**ACC**|
> |–|Qwen2-7B-Instruct|51.82|75.08|76.83|34.9/50.38|42.2/49.57|9.2/19.52|22.66|
> |Bert|Qwen2-7B-Instruct|25.45|73.86|77.77|26.1/37.41|26.2/29.93|8.2/15.25|21.47|
> |ColBert|Qwen2-7B-Instruct|52.32|75.18|77.34|36.6/49.69|39.2/43.89|12/22.2|23.06|
> |BM25|Qwen2-7B-Instruct|52.32|75.68|77.26|36.4/50.4|45/49.5|7.5/17.2|21.47|
> |Llama3-8B-Instruct|Llama3-8B-Instruct|**59.83**|75.99|77.34|41.7/58.82|32.1/38.72|22.9/33.26|24.45|
> |Qwen2-7B-Instruct|Qwen2-7B-Instruct|57.97|**79.84**|**82.03**|**47.7/64.08**|**51.5/58.5**|**25.6/40.2**|**29.82**|
>
> *Note:* For BERT/ColBERT/BM25 used as extractor models, we set TG≥0.6 for generation.

---

> > ### Author Response · Authors · 2025-11-28
> >
> > It can be seen that *BERT fails to accurately distinguish relevant sentences from irrelevant ones*, making it difficult to filter noise effectively. BM25, being a term-frequency–based method, also *performs poorly in noise filtering*. Both methods may even cause *greater information loss* during extraction. This is why we do not adopt BERT/BM25 as extractor models.
> >
> > Instead, we leverage the *strong reading and comprehension abilities of LLMs* to extract relevant sentences and score the relevance, which yields significantly better performance.
> >
> > ## (2) The Cot and few-shot-based extraction framework of FRAG:
> > FRAG employs a *CoT- and few-shot–based* extraction framework based on the observation that LLMs perform better when they can learn from provided examples and reason over the query and retrieved passages. This enables them to identify key snippets, extract relevant information, score relevance, and conduct multi-hop analysis for self-recognition. *LLMs also outperform BERT-like models in both few-shot learning and CoT reasoning.*
> >
> > ## (3) FRAG requires the strong Instruction-following capability:
> > Furthermore, since we require the extractor to generate outputs after completing its reasoning, the output naturally contains both the reasoning process and the final extracted content. To reliably capture the final extraction results, we enforce a specific output format (provided in Appendix I.1), which requires strong instruction-following capability. In other words, the better an LLM adheres to the given instructions and output format, the more likely it is to preserve the expected relevant information and the more effective the extraction becomes (details in Appendix E.1.1). These factors explain why we choose Instruct models.
> >
> > If you have any further questions, please feel free to ask, and we will do our best to address them.

---

### Author Response · Authors · 2025-11-28
**Anonymous Code Repository Release**

Dear AC and Reviewers,

Thank you for your time and valuable feedback.

To facilitate reproducibility of our main results, we have released an anonymous code repository at:

https://anonymous.4open.science/r/_FRAG-B0F8/

Sincerely,
Authors of Submission 12973

---

### Author Response · Authors · 2025-12-02
**Author Summary for FRAG (Submission #12973)**

## Major contributions of our paper

We study the central problem that retrieval noise harms RAG performance and formally characterize its negative impact. Building on this, we propose FRAG, a snippet-level RAG framework with a Self-Recognition mechanism for complex multi-hop tasks, together with an efficient variant FRAG-ip. Across multiple LLMs and diverse QA benchmarks, FRAG consistently improves over naïve RAG and achieves state-of-the-art results in our settings, with up to ~5% / 13% average gains on simple / complex tasks.

## Advantages supported by positive points from the reviews
### **1. Motivation & Methods**
- (1) Addressing a central problem in RAG — performance degradation due to retrieval noise (Reviewer dXwT, Bx9J, z6uA).
- (2) Solid theoretical grounding for impact of noise and benefits of fine-grained filtering; clear theory - method connection (Reviewer dXwT, z6uA).
- (3) Introducing a snippet-level relevance framework (Reviewer K9XM) and a Self-Recognition mechanism for handling multi-hop/inference-style queries (Reviewer Bx9J, z6uA).
- (4) Proposing FRAG-ip as an efficient and practical variant for more real-world deployment (Reviewer dXwT, Bx9J, wkBi, z6uA).
### **2. Experiments & Performance**
- (1) Broad and careful experiments across multiple LLMs, with and without RAG baselines, on both simple and complex tasks (Reviewer dXwT, K9XM, Bx9J, wkBi, z6uA).
- (2) Consistent and substantial performance gains across many datasets, with particularly strong improvements on complex tasks and better handling of challenging retrieval/QA settings (Reviewer K9XM, Bx9J, wkBi, z6uA).
- (3) Greatly reduces context length while improving accuracy, confirming its effectiveness as both a noise filter and context condenser (Reviewer K9XM).
- (4) Meaningful ablation studies showing the contribution of each component, including Self-Recognition (Reviewer dXwT, Bx9J, z6uA).
### **3. Writing & Structure**
- (1) Clearly written, with a comprehensive problem decomposition supported by analysis and experiments (Reviewer wkBi).
- (2) “Good” presentation (Reviewers dXwT, Bx9J, z6uA).

## Acknowledged Limitations and Main Concerns from the Reviews

- **System complexity and latency:** FRAG introduces a multi-stage pipeline and extra LLM calls.

We fully agree that this is a key trade-off. In response, we (i) propose FRAG-ip as a more efficient variant, and (ii) provide detailed latency and memory comparisons against strong baselines under matched backbones and context budgets.

- **Baseline coverage and fairness:** Clarification of inapplicable results and fair comparison with advanced RAG baselines.

We acknowledge the need for clearer clarification and fairer comparisons. To address this, we (i) clarify these constraints in the appendix with a per-baseline table, (ii) provide EM/F1 results for most compared baselines in the appendix, and (iii) conduct additional experiments under matched backbones, with compute- and context-matched comparisons for most baselines.

- **Scope of tasks and models:** Extension to noisy web data, conversational settings, and simpler BERT/BM25-based extractors.

We (i) add new experiments on noisy web and conversational datasets, which further demonstrate the effectiveness of FRAG, and (ii) evaluate FRAG with BERT/BM25 as extractor models, which supports the rationale for choosing LLM as backbone.

---

### Author Response · Authors · 2025-12-03
**Revised Paper for Submission #12973 Uploaded**

Dear AC and Reviewers,

Thank you for your time and valuable feedback.

### We have uploaded the revised version of the paper and highlighted all modifications in blue.

Sincerely,

Authors of Submission #12973

---

### Meta-Review · Area_Chair_LuXu · 2025-12-06

**Summary:**

This paper describes FRAG, which is a way to improve the performance of retrieval augmented generation. While some of the reviewers had positive feedback (and some didn't), I remain concerned about the response to one of the reviewer, with the presentation of the results of computation time and performance, and which do not quite align. Can we know exactly how much better performance we get for the significant inference cost (if I understand correctly, even in the case of FRAG-ip)? Is it worth the computation cost, and are the differences significantly noticeable compared to something more efficient?

**Reviewer Concerns:**

See above

**Reviewer Scores:**

I cannot predict such a thing.

---

### Decision · Program_Chairs · 2026-01-26

Reject